# CP-Agent: Context-Aware Multimodal Reasoning for Cellular Morphological Profiling under Chemical Perturbations

**Yuxin Zhang[1,\*], Yiyao Li[2,\*], Ping Shu Ho[4], Simon See[4], Zhenqin Wu[2,†], Kevin Tsia[1,3,5,†]**

[1]Department of Electrical and Computer Engineering, The University of Hong Kong

[2]School of Computing and Data Science, The University of Hong Kong

[3]School of Biomedical Engineering, The University of Hong Kong

[4]Nvidia AI Technology Center

[5]Advanced Biomedical Instrumentation Centre

## Abstract

Cell Painting combines multiplexed fluorescent staining, high-content imaging, and quantitative analysis to generate high-dimensional phenotypic readouts to support diverse downstream tasks such as mechanism-of-action (MoA) inference, toxicity prediction, and construction of drug–disease atlases. However, existing workflows are slow, costly and difficult to interpret. Approaches for drug screening modeling predominantly focus on molecular representation learning, while neglecting actual experimental context (e.g., cell line, dosing schedule, etc.), limiting generalization and MoA resolution. We introduce CP-Agent, an agentic multimodal large language model (MLLM) capable of generating mechanism-relevant, human-interpretable rationales for cell morphological changes under drug perturbations. At its core, CP-Agent leverages a context-aware alignment module, CP-CLIP, that jointly embeds high-content images and experimental metadata to enable robust treatment and MoA discrimination (achieving a maximum F1-score of 0.896). By integrating CP-CLIP outputs with agentic tool usage and reasoning, CP-Agent compiles rationales into a structured report to guide experimental design and hypothesis refinement. These capabilities highlight CP-Agent's potential to accelerate drug discovery by enabling more interpretable, scalable, and context-aware phenotypic screening – streamlining iterative cycles of hypothesis generation in drug discovery.

## 1 Introduction

High-content imaging with Cell Painting has become a workhorse for scalable phenotypic drug discovery. This technique, integrating advanced microscopy, multiplexed fluorescent staining and quantitative image analysis, allows us to establish high-dimensional morphological cell profiles that capture rich multiscale cellular responses to chemical perturbations. These profiles have been proven valuable in supporting mechanism-of-action (MoA) inference (Tian et al., 2023), toxicity prediction (Ewald et al., 2025), hit triage (Vincent et al., 2020), and drug repurposing (Fredin Haslum et al., 2024), while also enabling the construction of reference atlases and improved target deconvolution (Moffat et al., 2017).

In Cell Painting workflows, cells are perturbed under diverse conditions and the experimental context is not a nuisance to control but a signal to model. For instance, dose and time define trajectories; cellular background modulates pathway readouts (Appendix B.2). The resulting profiles guide follow-up experiments and can advance phenotype-driven drug discovery. However, Cell Painting-based drug discovery remains limited by several challenges: (i) complex intermediate dependencies: Morphological responses are highly context-dependent. For example, concentration-dependent profiles show low correlations across dose levels (Pearson r = 0.21-0.26) (Trapotsi et al., 2022), and MoA prediction is sensitive to cell line context (Seal et al., 2024). Ignoring these structures conflates

---

*Equal contribution. †Corresponding authors. Project page: https://github.com/letitia-zhang/CP-Agent

biology with acquisition artifacts and wastes the valuable metadata; (ii) convergent morphologies: Compounds with distinct mechanisms may induce morphological readouts convergence, reducing MoA resolution, thereby complicating the extraction of standardized, interpretable descriptors. (iii) Lack of semantic grounding: Representing image embeddings as unstructured feature vectors restricts their capacity for semantic reasoning and downstream biological inference.

Recently, various AI methods have been introduced to Cell Painting datasets, such as generative approaches to synthesize images under perturbations (Navidi et al., 2024; Cross-Zamirski et al., 2023; Palma et al., 2025), multimodal frameworks integrating chemical and genetic annotations with cell painting images (Sanchez-Fernandez et al., 2023) (Fradkin et al., 2024; Lu et al., 2025). For example, CLOOME firstly introduced a CLIP-style model to align Cell Painting images with molecular structures. MolPhenix and CellCLIP further extend this direction by leveraging strong unimodal foundation models to align the molecule. However, many existing models offer visual embeddings as black-box features, which lack semantic interpretability. Moreover, experimental context is often under-used: metadata is appended via late fusion or treated as unstructured text, yielding less informative representations and hindering iterative, closed-loop experimental design. Meanwhile, emerging multimodal large language models (MLLMs) offer reasoning capabilities and have been applied in diverse biological domains, such as genomics, biomedical imaging, and omics data analysis (Zhang et al., 2024a; Lin et al., 2025; Liu et al., 2024b; Hu et al., 2024b; Zhang et al., 2024b). Yet their applications in drug screening remain underexplored.

In this work, we introduce CP-Agent, a context-aware, agentic MLLM framework for Cell Painting drug perturbation screening. At its core is CP-CLIP, a contrastive alignment module that jointly embeds Cell Painting images and structured experimental context, including drug compounds and other essential experimental conditions, enhancing the biological relevance of cell morphology. The model is pretrained on 1.9 million image-context pairs, with a customized token injection strategy that embeds key fields for better alignment. Comprehensive evaluations across curated classification tasks show that CP-CLIP outperforms general-purpose baselines. Built on this perception layer, CP-Agent integrates tool-augmented reasoning and task-adapted MLLMs grounded in phenotype descriptors and MoA ontologies to generate structured, interpretable outputs. Together, this agentic system supports scalable and interoperable phenotypic analysis, enabling cross-study generalization and providing actionable insights for assay prioritization and iteration, thereby accelerating hypothesis generation and improving decision-making in phenotypic drug discovery.

## 2 METHOD

### 2.1 DATASET

We employed three open-access Cell Painting datasets, consisting of approximately 1.9 million pairs: BBBC021 (Caie et al., 2010), CPJUMP1 (Chandrasekaran et al., 2024), and RxRx3 (Fay et al., 2023), encompassing diverse compound-induced phenotypes. Each image-context pair comprises a microscopy image and its associated experimental context (e.g., cell lines, experimental treatment conditions) We curated compounds to ensure traceable MoA labels across datasets. For each collection, we matched SMILES representations of the perturbing chemical compounds to ChEMBL, retrieved their targets and MoAs, and retained only compounds with publicly resolvable MoA names. A summary of the curated multi-dataset setting is provided in Table 1. More details about dataset backgrounds are provided in Appendix C.

Table 1: Summary of datasets used in this study

| Dataset | Cell line | Channel | Compound | Concentration | Time | Image Pair |
|---------|-----------|---------|----------|---------------|------|------------|
| BBBC021 | MCF-7 (p53 WT) | 3 | 34 | Variable 8-point half-log | 24 h | 144,411 |
| CPJUMP1 | U2OS, A549 | 5 | 62 | 5.0 µM | 24 h, 48 h | 562,687 |
| RXRX3 | HUVEC | 6 | 380 | Fixed 8-point half-log | ∼20 h | 1,265,984 |

The training set comprises 1,846,436 image–text pairs, while the validation set contains 9,395 pairs. For zero-shot evaluation, we curated a held-out set of compounds spanning all three datasets, selected to assess generalization to unseen perturbations.

## 2.2 MOLECULAR DRUG ENCODING

Several established approaches map compound perturbations to vector representations, enabling alignment with image embeddings and facilitating multimodal learning (Winter et al., 2019; Wu et al., 2025). For instance, SMILES-based (e.g., ChemBERTa) and graph-based models learn molecular embeddings from structure, often using RDKit for preprocessing. Alternatively, one can compute continuous molecular descriptor embeddings (e.g., physicochemical and topological descriptors), formalized as a parameterized feature extractor: $\phi_{\text{desc}}(x; P) = [f_1(x; P_1), f_2(x; P_2), \ldots, f_d(x; P_d)] \in \mathbb{R}^d$, where $x$ is an input molecular representation (e.g., SMILES strings or molecular graphs), and each $f_i(x; P_i)$ extracts a specific property, forming a $d$-dimensional real-valued feature vector. In contrast, binary fingerprint embeddings that encode the presence/absence of substructures (e.g., Morgan/circular, MACCS, or path-based fingerprints) (Bento et al., 2020) $\phi_{\text{fp}} : \mathcal{M} \to \{0, 1\}^d$ or $\mathbb{N}_0^d$, yield binary or count-based encoding over the molecular space $\mathcal{M}$.

## 2.3 CP-CLIP: REPROCESSING

To harmonize **Cell Painting images** across datasets with varying resolution and signal quality, we defined a channel-wise preprocessing step: $\mathcal{P} : \mathbb{R}^{H_0 \times W_0} \to \mathbb{R}^{H \times W}$, applied independently to each fluorescence channel. This includes Contrast Limited Adaptive Histogram Equalization (CLAHE), random Laplacian sharpening, and gamma correction, yielding enhanced images $\tilde{I} = \mathcal{P}(I)$. Enhanced single-channel images are then cropped into $512 \times 512$ patches and stacked, yielding input tiles $x_p \in \mathbb{R}^{512 \times 512 \times C}$. For each perturbation tile $x_p$, a corresponding control tile $x_c \in \mathbb{R}^{512 \times 512 \times C}$ is independently sampled from a matching control set $\Omega(x_p)$, which share all experimental contexts (e.g., plate, cell line, channel) with $x_p$, except for the perturbation compound. That is $x_c \sim \mathcal{U}(\Omega(x_p))$. The final image branch input is formed by concatenating the grayscale perturbation and control tiles along the channel dimension, $\hat{x} = \text{concat}(x_p, x_c) \in \mathbb{R}^{512 \times 512 \times 2}$. This paired design encourages the model to learn the contrasts between treated and untreated states.

**Molecular descriptors** are projected via a fixed dimensional mapping $f_{\text{desc}} : \mathcal{X} \to \mathbb{R}^d$, where each feature dimension corresponds to a predefined physicochemical or topological property (See Appendix D). Let $v = f_{\text{desc}}(x) \in \mathbb{R}^d$ denote the raw descriptor vector for compound $x \in \mathcal{X}$. To ensure numerical stability and comparability across compounds, dimensions containing undefined values (e.g., NaNs or Infs) are removed, and z-score normalization is applied independently to each feature dimension $\tilde{v}_i = \frac{v_i - \mu_i}{\sigma_i}$.

To account for the compound-specific dosing scheme, each molecule is represented by a normalized dosing pair $[\rho_{\max}, s(C)]$, where $\rho_{\max}$ denotes the molecular mass-normalized maximum concentration (in $mg/mL$), and $s(C)$ is the log-scaled dose step index corresponding to a given concentration. Let $M \in \mathbb{R}_{>0}$ denote the molecular weight (in $Da$ or $g/mol$), and $C_{\max} \in \mathbb{R}_{>0}$ the nominal maximum concentration (in $\mu$M). So, the molecular maximum mass concentration is given by:

$$\rho_{\max}[\text{mg/mL}] := \frac{M[\text{Da}] \cdot C_{\max}[\mu\text{M}]}{10^6} \tag{1}$$

where the denominator $10^6$ reflects the conversion from $\mu$M and $Da$ to $\text{mg/mL}$. While for each titration point $C \in \{C_1, \ldots, C_8\}$, a pseudo-step index is computed on a log scale to reflect dilution ratios:

$$s(C) := \frac{\log_{10}(C_{\max}) - \log_{10}(C)}{\Delta \log}, \quad \Delta \log = 0.5 \tag{2}$$

where the denominator 0.5 corresponds to the log-fold change between adjacent titration levels in a 2-fold serial dilution protocol. A detailed derivation is provided in Appendix E.

For **observation time**, let $t \in \mathbb{R}_{\geq 0}$ denote time in days. Temporal normalization rescales $t$ into the unit interval via: $\tilde{t} = \frac{t}{T_{\max}}$, with $T_{\max} = 112$. The 112-day (16-week) window reflects the FDA's stopping rule, adopted by Watkins et al. (2022) in their pharmacoeconomic analysis. These representations ensure that the input space remains consistent across compounds with varying dosing schemes and time-points.

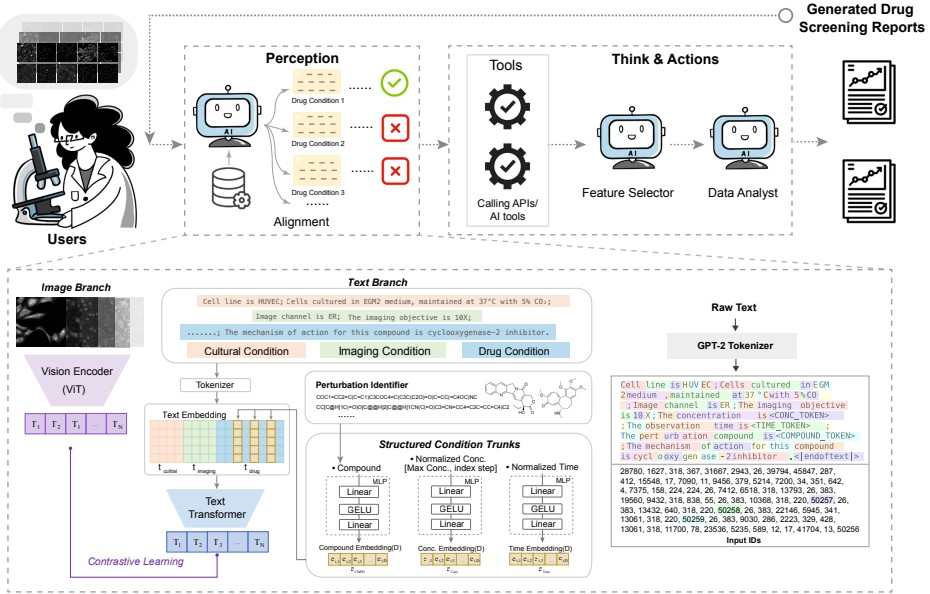

Figure 1: Illustration of the CP-agent (top) and CP-CLIP (bottom). CP-Agent connects perception, memory retrieval, and modular analysis into a unified pipeline for generating reports for Cell Painting experiments. CP-CLIP forms the backbone of the CP-Agent's perception module, providing joint embeddings of Cell Painting images and structured experimental context.

## 2.4 CP-CLIP: CONTEXT-AWARE TOKEN PROJECTION

Our contrastive framework uses a structured text encoder tailored to the metadata obtained from drug screening experiments (Figure 1, bottom). Each experiment is represented as a prompt-like sequence composed of cell culture, imaging, and drug compound perturbation conditions. So the "raw text" refers to structured experimental metadata such as cell line, culture medium, imaging parameters, compound identity, dosage, time and other cultural information if have. These contextual descriptions are first composed into a natural language-style sentence and tokenized into input IDs using the standard GPT-2. To accommodate structured context and consistent representations of perturbing compounds, we introduced field-specific placeholder tokens (i.e. $\texttt{<CMPD>}$, $\texttt{<CONC>}$, $\texttt{<TIME>}$) for compound descriptors $z_{\text{cmpd}} = \phi_{\text{desc}}(x; P) \in \mathbb{R}^d$, normalized concentration $z_{\text{conc}} = [\rho_{\max}, s(C)] \in \mathbb{R}^2$, and normalized time $z_{\text{time}} = \tilde{t} \in \mathbb{R}$. The special placeholder tokens are directly inserted into the text sequence and registered into the tokenizer's vocabulary. During tokenization, they are automatically recognized as atomic units and their positions are preserved without being split or altered. Their embeddings are then dynamically computed via field-specific Multilayer Perceptron (MLP) trunks $f_* : \mathbb{R}^{d'} \to \mathbb{R}^D$:

$$
\begin{aligned}
e_{\text{cmpd}} &= f_{\text{cmpd}}\left(z_{\text{cmpd}}\right) \in \mathbb{R}^D \\
e_{\text{conc}} &= f_{\text{conc}}\left(z_{\text{conc}}\right) \in \mathbb{R}^D \\
e_{\text{time}} &= f_{\text{time}}\left(z_{\text{time}}\right) \in \mathbb{R}^D
\end{aligned}
\tag{3}
$$

where $f_{\text{cmpd}}, f_{\text{conc}}$, and $f_{\text{time}}$ are lightweight MLP trunks encoding compound identity, concentration, and time-point used in place of the placeholders. The resulting text input is a hybrid sequence:

$$
X = [\text{CLS}, t_1, t_2, \dots, \underbrace{e_{\text{cmpd}}}_{\texttt{<CMPD>}}, \dots, \underbrace{e_{\text{conc}}}_{\texttt{<CONC>}}, \dots, \underbrace{e_{\text{time}}}_{\texttt{<TIME>}}, \dots]
\tag{4}
$$

This hybrid sequence, combining standard subword embeddings $t_i \in \mathbb{R}^D$ with structured embeddings $\mathbf{e}_* \in \mathbb{R}^D$ from field-specific MLPs, is fed into the text Transformer to produce final text representation. Implementation details are in Appendix F. By replacing placeholder tokens with learned embeddings, the model fuses continuous metadata with discrete language tokens in a shared embedding space. The text encoder thus captures both experimental signals and linguistic coherence, enabling better semantic alignment.

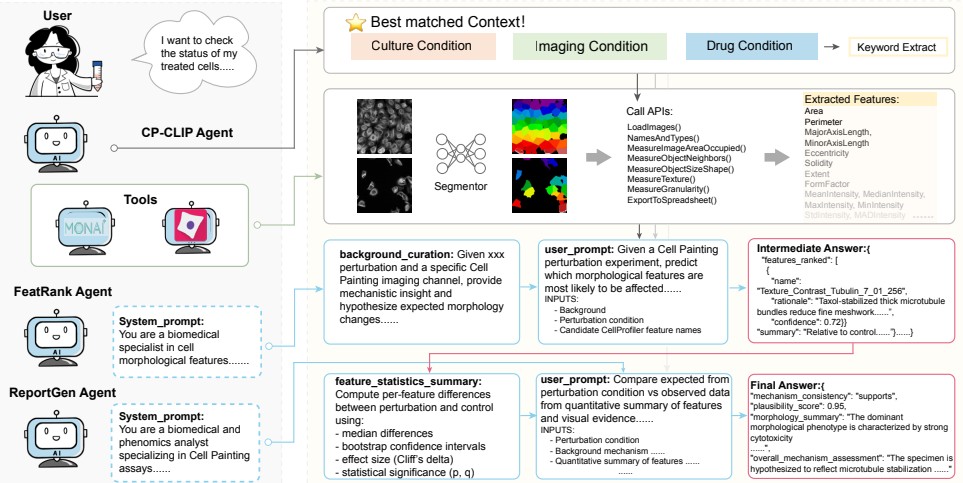

Figure 2: Automated cell-phenotype assessment pipeline of CP-Agent. Upon user query, CP-CLIP retrieves the relevant experimental context to guide cell segmentation and feature extraction. Downstream agents then rank morphological changes and generate interpretable, end-to-end phenotype reports.

## 2.5 CP-AGENT WORKFLOW

CP-Agent adopts a modular, memory-augmented architecture that connects perception, tooling, and analysis into a single-pass pipeline (Figure 1, top). Given user-provided Cell Painting images, a lightweight memory retriever powered by CP-CLIP fetches the most probable experimental context (i.e., cell line, fluorescence channels, imaging settings, chemical perturbations). Once the experimental context is retrieved, the pipeline proceeds to visual analysis. Rather than relying on vision backbones that produce holistic, biologically opaque embeddings, we extract handcrafted single-cell morphological features. These interpretable representations are processed by a modular, MLLM-driven agent architecture, where the MLLM serves as a policy layer that dynamically routes tasks to interchangeable tools and integrates their outputs. We frame this system as "agentic" in the sense of *procedural autonomy* (Xu et al., 2025): unlike reinforcement learning-based planners, CP-Agent employs the MLLM as a cognitive controller within a structured workflow. It relies on the model's learned reasoning capabilities—rather than fixed logical scripts—to dynamically prioritize morphological features, interpret statistical distribution shifts, and synthesize mechanism-level hypotheses based on retrieved experimental context.

We instantiate this concept on fluorescence Cell Painting data via a specialized CP-Agent workflow (Figure 2), which comprises the following steps:

- **CPContext Agent** Given paired Cell Painting images (control vs. perturbation) acquired under matched conditions, the *CPContext Agent* employs a pre-trained CP-CLIP retriever to obtain experimental context from a curated knowledge base. Simultaneously, it harmonizes metadata via controlled-vocabulary tagging and channel labeling to generate standardized descriptors. Retrieved context is routed both (A) as a context bundle to *FeatRank Agent*, *ReportGen Agent*, and (B) as metadata keywords to the *CellFeat Agent*.

- **ChannelSeg Agent** Given Cell Painting images, the *ChannelSeg Agent* performs nuclei instance segmentation on DNA-stained channels and whole-cell segmentation on non-DNA channels (e.g., RNA, Actin, ER, etc.). It outputs channel-specific instance masks, which are passed to the *CellFeat Agent*.

- **CellFeat Agent** Given Cell Painting images, corresponding masks, and harmonized metadata, the *CellFeat Agent* extracts per-cell morphological, intensity, texture, granularity, neighborhood, and occupancy features using a configured CellProfiler pipeline (Appendix H). Output is routed both

(A) as extracted feature items to the *FeatRank Agent* for mechanism-aware selection, and (B) as channel-wise single-cell feature matrices to the *StatSynth Agent* for statistical evidence synthesis.

- **FeatRank Agent** Given extracted feature items and experimental context, the *FeatRank Agent* scores and ranks features by their likelihood of being influenced by the perturbation. It generates confidence-weighted rationales to support prioritization. Output is routed as a prioritized feature list with explanations to the *StatSynth Agent*.

- **StatSynth Agent** Given the prioritized feature list, full feature matrices, and experiment-level context, the *StatSynth Agent* computes per-feature statistical evidence between control and perturbation conditions based on the prioritized features. It summarizes distribution shifts, effect sizes, confidence intervals, and statistical significance. Outputs are routed as statistical summaries and interpretations to the *ReportGen Agent* for final report composition.

- **ReportGen Agent** Given statistical summaries, prioritized features, visual exemplars, and experimental context, the *ReportGen Agent* composes an integrated interpretation of the perturbation's biological impact. It identifies key morphological shifts and evaluates their consistency with expected cellular responses to infer plausible mechanisms. The resulting report summarizes these findings, provides follow-up recommendations and visualizations, and is delivered to the users for downstream access.

The agent tool stack integrates both classical and learning-based components. For segmentation, we fine-tuned VISTA-2D He et al. for 20 epochs using diverse augmentation strategies to mitigate optics-induced batch effects. The model generates channel-specific masks that enable biologically consistent segmentation across diverse imaging conditions. More details regarding dataset preparation and training of the segmentation model are provided in Appendix J. The *StatSynth Agent* is tasked with reasoning over high-dimensional single-cell morphological data (typically 30–300 cells per image), which is impractical for direct LLM application due to length constraints and noise (Fang et al., 2024). Instead, we curate agentic tools that (i) aggregate summary statistics for key features, and (ii) quantify distribution shifts between control and perturbed samples. These compact, interpretable summaries support reliable LLM-based reasoning. Detailed procedures for this step are provided in Appendix L.

## 3 EXPERIMENTS AND RESULTS

Table 2: Model performance on classification tasks

| Model | Cell line | Channel | Perturbation Compound | | | | | | | | | | |
|---|---|---|---|---|---|---|---|---|---|---|---|---|---|
| | | | Flindokalner | Racecadotril | AZM-475271 | Misoprostol | Trazodone | Orantinib | Rufinamide | Lumiracoxib | BIRB-796 | Methoxsalen | Macro-avg |
| Random Guessing | 0.25 | 0.143 | 0.10 | 0.10 | 0.10 | 0.10 | 0.10 | 0.10 | 0.10 | 0.10 | 0.10 | 0.10 | 0.10 |
| Grok-4 | 0.448 | 0.228 | 0.215 | 0.174 | 0.0 | 0.0 | 0.410 | 0.190 | 0.034 | 0.0 | 0.0 | 0.0 | 0.102 |
| GPT-5 | 0.377 | 0.439 | 0.059 | 0.168 | 0.0 | 0.0 | 0.353 | 0.0 | 0.0 | 0.0 | 0.0 | 0.0 | 0.074 |
| Claude-4-Sonnet | 0.450 | 0.198 | 0.0 | 0.00 | 0.0 | 0.057 | 0.0 | 0.0 | 0.0 | 0.0 | 0.211 | 0.0 | 0.027 |
| Gemini-2.5-Pro | 0.526 | 0.628 | 0.0 | 0.0 | 0.0 | 0.0 | 0.0 | 0.023 | 0.0 | 0.0 | 0.045 | 0.0 | 0.007 |
| CLOOME ViT-B/16 | - | - | 0.784 | 0.784 | 0.729 | 0.854 | 0.623 | 0.849 | 0.653 | 0.619 | 0.854 | 0.800 | 0.755 |
| CLIP ViT-B/16 | 1.000 | 0.955 | 0.776 | 0.680 | 0.661 | 0.216 | 0.629 | 0.447 | 0.500 | 0.600 | 0.575 | 0.642 | 0.657 |
| SigLIP-ViT-B/16 | 1.000 | 0.925 | 0.734 | 0.471 | 0.515 | 0.826 | 0.291 | 0.638 | 0.395 | 0.272 | 0.604 | 0.400 | 0.514 |
| CP-CLIP SigLIP-ViT-B/16 *(descriptor)* | 1.000 | 0.934 | 0.685 | 0.442 | 0.485 | 0.776 | 0.351 | 0.860 | 0.255 | 0.186 | 0.660 | 0.620 | 0.532 |
| CP-CLIP ViT-B/16 *(fingerprint)* | 1.000 | **0.991** | 0.839 | 0.862 | 0.891 | 0.875 | **0.913** | 0.914 | 0.894 | 0.840 | **0.971** | 0.875 | 0.887 |
| CP-CLIP ViT-B/16 *(descriptor)* | 1.000 | 0.882 | 0.907 | 0.869 | 0.857 | **0.942** | 0.848 | **0.940** | 0.884 | **0.854** | 0.932 | 0.922 | **0.896** |
| CP-CLIP ViT-L/16 *(descriptor)* | 1.000 | 0.849 | **0.928** | **0.880** | **0.896** | 0.846 | 0.843 | 0.929 | **0.911** | 0.819 | 0.915 | **0.941** | 0.891 |

Table 3: Unseen drugs similarity score

| Model | Regorafenib | Sacubitril | Buparlisib | Dexamethasone | Nimodipine | AZ258 | Nilotinib | MG-132 | Average |
|---|---|---|---|---|---|---|---|---|---|
| CLIP ViT-B/16 | $0.207 \pm 0.082$ | $0.2058 \pm 0.104$ | $0.289 \pm 0.046$ | $0.3601 \pm 0.049$ | $0.377 \pm 0.039$ | $0.328 \pm 0.069$ | $0.174 \pm 0.080$ | $0.346 \pm 0.072$ | 0.286 |
| SigLIP ViT-B/16 | $0.038 \pm 0.082$ | $0.095 \pm 0.099$ | $0.129 \pm 0.073$ | $0.146 \pm 0.091$ | $0.183 \pm 0.067$ | $0.090 \pm 0.186$ | $-0.055 \pm 0.103$ | $0.143 \pm 0.101$ | 0.096 |
| CP-CLIP SigLIP-ViT-B/16 *(descriptor)* | $0.378 \pm 0.077$ | $0.420 \pm 0.193$ | $0.323 \pm 0.102$ | $0.503 \pm 0.130$ | $0.515 \pm 0.075$ | **$0.488 \pm 0.115$** | $0.303 \pm 0.090$ | $0.380 \pm 0.114$ | 0.414 |
| CP-CLIP ViT-B/16 *(fingerprint)* | $0.297 \pm 0.093$ | $0.222 \pm 0.072$ | $0.375 \pm 0.053$ | $0.468 \pm 0.052$ | $0.461 \pm 0.046$ | $0.429 \pm 0.120$ | $0.210 \pm 0.109$ | $0.420 \pm 0.081$ | 0.360 |
| CP-CLIP ViT-B/16 *(descriptor)* | $0.432 \pm 0.098$ | $0.412 \pm 0.094$ | $0.396 \pm 0.043$ | $0.503 \pm 0.073$ | $0.469 \pm 0.032$ | $0.468 \pm 0.104$ | **$0.324 \pm 0.085$** | $0.448 \pm 0.081$ | 0.432 |
| CP-CLIP ViT-L/16 *(descriptor)* | **$0.455 \pm 0.115$** | **$0.445 \pm 0.135$** | **$0.408 \pm 0.053$** | **$0.530 \pm 0.072$** | **$0.523 \pm 0.032$** | $0.448 \pm 0.106$ | $0.295 \pm 0.089$ | **$0.448 \pm 0.077$** | **0.444** |

To assess the effectiveness of CP-Agent, we isolated and evaluated its core components before measuring end-to-end reporting quality: (a) CP-CLIP (context-aware retrieval and alignment): we evaluate its accuracy on in-distribution classification (seen-drug) and generalization (unseen-drug matching), ablations against MLLM baselines and CLIP variants; (b) Vision embedding structure: we evaluate whether CP-CLIP embeddings encode chemically grounded, dose- and MoA-dependent morphology; (c) Statistical synthesis and reporting: whether compact summaries enable robust comparisons between control and perturbation in the generated report. Finally, we assessed the effectiveness of full CP-Agent reports via expert review.

## 3.1 MODEL VARIANTS AND MLLM BASELINES

To contextualize the performance of our proposed model, we compared it against several leading MLLMs. Specifically, we included Grok-4 (xAI, 2025), GPT-5 (OpenAI, 2025), Claude-4-Sonnet (Anthropic, 2025), and Gemini-2.5-Pro (Google DeepMind, 2025), which have demonstrated strong performance across a range of general-purpose multimodal benchmarks. Following recent benchmarking protocols for MLLMs in biomedical and healthcare settings (Lozano et al., 2024; Burgess et al., 2025), we adopt a zero-shot, two-stage prompting pipeline: first, the models were prompted to curate background knowledge relevant to Cell Painting experiments; then, they were asked to answer multiple-choice questions about experimental conditions given the curated background knowledge, paired control and perturbation images, and masked textual prompts. To further narrow the adaptation gap and make the comparison more conservative, we additionally evaluated a few-shot variant in which the MLLMs were provided with a small visual memory bank (two labeled exemplars per class) before answering the same tasks. Detailed prompt templates and the corresponding zero-shot and few-shot results are reported in the Appendix M.

Alongside these MLLMs, we benchmarked multiple variants of our contrastive learning framework, CP-CLIP, which extends the CLIP architecture by integrating structured experimental context into training. As a baseline, we used the original CLIP model based on the ViT-B/16 vision backbone, retrained on natural language text aligned with Cell Painting images. All CP-CLIP variants enhance this setup by injecting serialized numerical metadata, as detailed in Section 2.4. We evaluated CP-CLIP variants that differ in compound encoding and loss function (See Appendix G), including: (i) a descriptor-based model used continuous molecular descriptors, and (ii) a fingerprint-based model used binary fingerprints. We also tested a SigLIP variant that uses a sigmoid-based pairwise contrastive objective (Zhai et al., 2023). To assess the impact of vision model capacity on performance, we tested a CP-CLIP variant with ViT-L/16 vision backbone.

## 3.2 TASK I: SEEN-DRUG CLASSIFICATION

To benchmark in-distribution performance, we designed classification tasks across three categories: cell line, fluorescence channel and compound. The classification is performed retrieval-based inference by ranking cosine similarity scores between image embeddings and a set of candidate textual prompts, following the standard CLIP paradigm. Those contextual metadata includes both textual and numerical variables, which are encoded jointly as a natural language sequence, enabling prompt-based querying without the needs for task-specific heads. For example, for compound classification, 10 compounds were randomly sampled to form a balanced 10-class setting. Table 2 summarizes the results. Among all general-purpose MLLMs, Gemini-2.5-Pro achieved the best performance on the cell line and channel prediction tasks (F1: 0.526 and 0.628). However, on compound classification, performance dropped sharply: All models fell below random baseline, except for Grok-4, which slightly exceeded it. Confusion matrices (Appendix M.3) revealed near-zero F1 scores, indicating systematic failure in identifying perturbing chemical compounds and limited generalization of current MLLMs. In contrast, CP-CLIP consistently outperformed both the baseline CLIP and all MLLMs across tasks. Descriptor-based models slightly outperformed fingerprint-based ones on compound classification (F1: 0.891 vs. 0.887), indicating that continuous encodings provide richer chemical contexts. Scaling the vision encoder from ViT-B/16 to ViT-L/16 yielded no significant gain (F1: 0.896 vs. 0.891), indicating that a lightweight backbone suffices when paired with strong chemical priors. Taken together, these MLLMs results also constitute a "no-CPContext" baseline, reinforcing the conclusion that without explicit perturbation-aware grounding, current MLLMs fail to extract meaningful biological signals from Cell Painting image. This emphasizes the essential role of CP-CLIP as the perception in CP-agent.

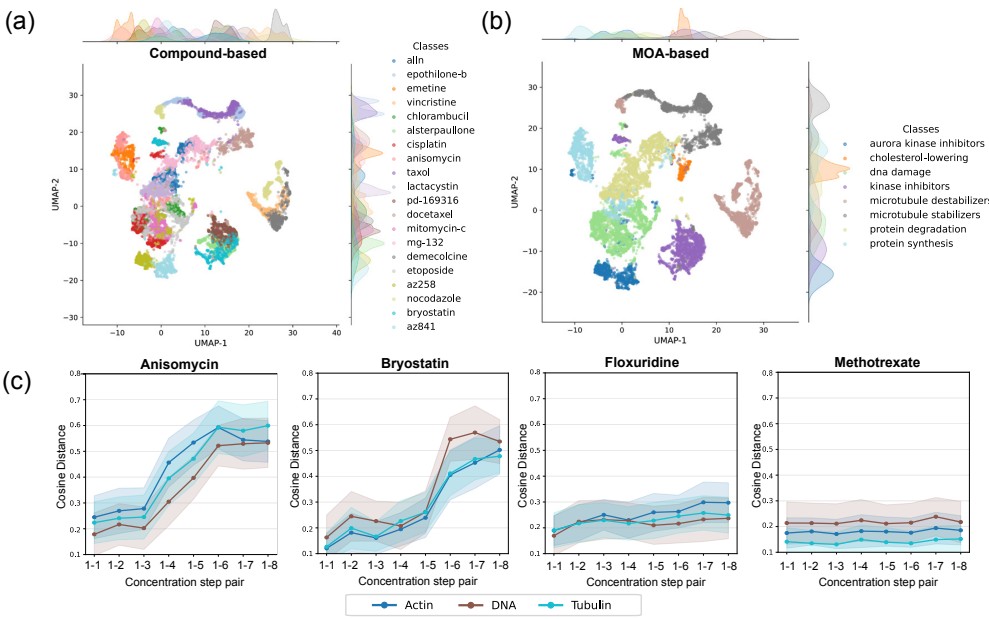

Figure 3: CP-CLIP captures pharmacologically meaningful morphology. UMAP projections of CP-CLIP image embeddings, colored by (a) compound identity and (b) mechanism of action (MoA). The clear clustering indicates that the learned representation encodes biologically relevant morphology. (c) Concentration-dependent morphological changes are captured using image embeddings extracted from samples treated with varying compound doses.

### 3.3 TASK II: UNSEEN-DRUG MATCHING

To evaluate generalization, we performed zero-shot prompt–image matching on held-out compounds by computing cosine similarity between image and prompt embeddings (Table 3). The baseline CLIP model (ViT-B/16) yielded low alignment on unseen drugs (avg. similarity = 0.286), while CP-CLIP (descriptor, ViT-B/16) achieved 0.432, a 14.6% absolute increase. Descriptor-based models also outperformed fingerprint-based ones (0.432 vs. 0.360), indicating that continuous encodings capture more relevant chemical contexts. Scaling the vision encoder from ViT-B/16 to ViT-L/16 further improved performance to 0.444, suggesting enhanced robustness to morphological variation. To provide a comparative reference, we also evaluated similarity on seen drugs (Appendix I). Notably, performance on unseen drugs remained close, indicating strong generalization. Specifically, descriptor-based ViT-B/16 and ViT-L/16 models achieved 0.549/0.432 and 0.561/0.444 on seen/unseen drugs, suggesting that CP-CLIP captures mechanism-relevant biology, rather than memorizing labels. This zero-shot capability supports practical applications such as MoA hypothesis generation, hit prioritization, and generalization to novel perturbation contexts.

### 3.4 VISION EMBEDDING ANALYSES

Figure 3a-b shows UMAP projections of embeddings from CP-CLIP ViT-B/16 (descriptor). The UMAP projection reveals clustering by MoA, indicating the learned representation encodes pharmacologically meaningful morphology beyond compound identity. Figure 3c shows concentration-related patterns for four drugs selected from the BBBC021 and RxRx3 datasets. CP-CLIP embeddings exhibited clear dose–response trajectories, reflecting concentration-dependent morphological change. In particular, the sharp dose-responses observed for Anisomycin and Bryostatin are consistent with previous reports Cranston et al. (1982); Marshall et al. (2002). In contrast, drugs with minimal morphological impacts show flatter trends across dosage. More examples and a detailed explanation of this schematic are provided in the Appendix K.

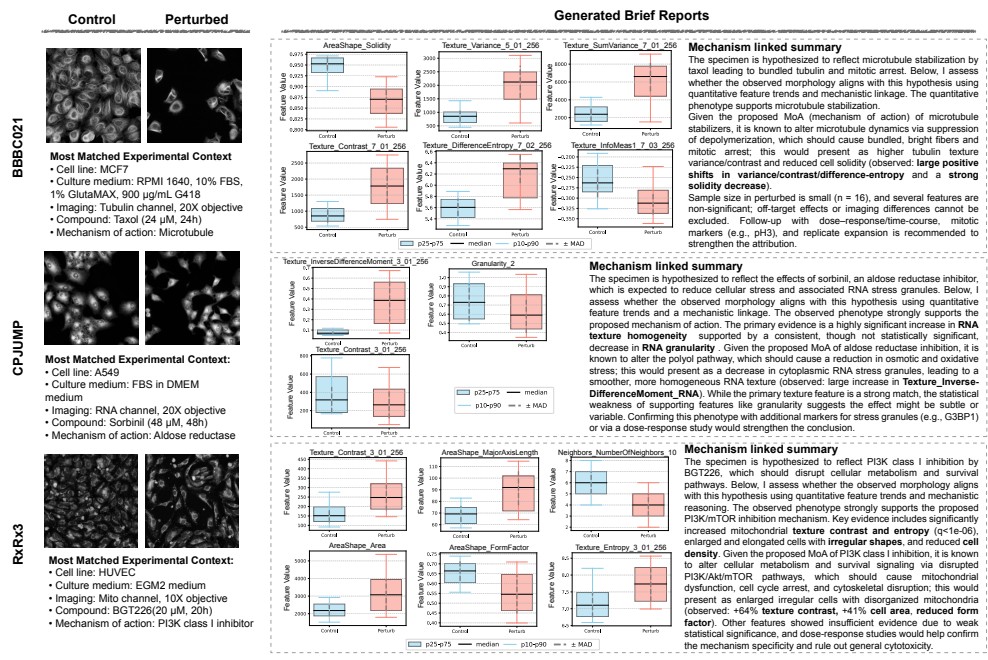

Figure 4: Summary reports generated from CP-Agent. The examples show CP-Agent's ability to recognize clear (Taxol), subtle (Sorbinil), and complex (BGT226) morphological responses, linking them to plausible biological mechanisms.

## 3.5 CP-AGENT REPORTS

We present three case studies from different datasets to demonstrate CP-Agent generated reports (Figure 4): (i)*Example 1 (BBBC021, MCF7 + Taxol)*: Taxol induces a clear *cytoskeletal phenotype* by stabilizing microtubules and arresting mitosis (Kiwanuka et al., 2022). CP-Agent detected the localized changes in tubulin texture and correctly linked them to microtubule stabilization and mitotic arrest, demonstrating its ability to recognize canonical, visually prominent phenotypes. (ii) *Example 2 (CPJUMP, A549 + Sorbinil)*: Sorbinil is an aldose reductase inhibitor that produces a *subtle and uncertain phenotype* (Zietek et al., 2025). CP-Agent detected modest shifts (e.g., smoother RNA texture, reduced granularity), and suggested potential stress granule suppression. Meanwhile, it also flagged ambiguity and suggested further validation, illustrating its ability to reason under uncertainty. (iii) *Example 3 (RxRx3, HUVEC + BGT226)*: BGT226 is a PI3K/mTOR inhibitor, leading to a *multi-compartment phenotype* affecting organelles, cell shape, and density (Kampa-Schittenhelm et al., 2013). By integrating mitochondrial texture, cell area, and density changes, CP-Agent inferred PI3K/mTOR inhibition, showcasing its capacity to synthesize complex morphological cues into mechanistic insights. Together, these cases show that CP-Agent adapts to diverse biological contexts, ranging from clear to ambiguous phenotypes, and generates biologically grounded summaries. Additional examples and reasoning details are provided in Appendix O.

We conducted an expert survey to assess whether LLM-based CP-Agent produces accurate and well-reasoned screening reports. Four LLMs (mentioned in Section 3.1) each generated reports for ten control–perturbation image pairs. Experts (N = 11), ranging from PhD students to professors in pharmacology or related fields, rated 40 reports (10 pairs × 4 models) on a 1–7 scale across ten criteria from Waqas et al. (2025), covering language quality and reasoning quality. Full criteria definitions and examples are provided in Appendix P. As shown in Figure 16, most metrics received high scores across models. CP-Agent powered by GPT-5 showed the strongest overall reasoning performance, followed closely by Gemini-2.5-Pro. To further assess the interpretability and consistency of the CP-Agent framework, we conducted a systematic evaluation of the two LLM-powered modules: FeatRank Agent and ReportGen Agent. As shown in Table 15, the selected morphological features remained highly consistent across runs, indicating robust and stable feature prioritization.

Table 16 also revealed a stable corpus-level consistency of the report generated from ReportGen Agent.

## 4 DISCUSSION AND CONCLUSION

We present CP-Agent, a context-aware multimodal reasoning framework for interpretable analysis of Cell Painting drug responses. Its core, CP-CLIP, aligns imaging data with experimental context, enhanced by numerically grounded token injection. This yields strong generalization and outperforms baselines on multiple classification tasks. CP-Agent separates and coordinates perception, retrieval, analysis, and reporting into specialized agents (i.e.,CPContext, ChannelSeg, CellFeat, FeatRank, StatSynth, ReportGen). This enables an evidence-first workflow where CP-Agent converts high-dimensional morphological features, together with the experimental context, into compact, calibrated summaries that an MLLM synthesizes into interpretable narratives. Hence, CP-Agent allows end-to-end biological interpretability. Users can trace predicted mechanisms back to corresponding morphological features—from images to masks, features, statistics, and final explanations. Unlike histology tasks, where many agent-based pipelines can perform well without training by using a well-designed chain-of-thought with off-the-shelf MLLMs, our results show that zero-shot prompting for Cell Painting datasets consistently underperforms, and biologically grounded supervision is essential for meaningful reasoning. CP-Agent also generalizes to various imaging modalities such as quantitative phase imaging (QPI), digital holographic microscopy, and brightfield time-lapse imaging (Lo et al., 2024; Siu et al., 2023; Zhang et al., 2023; Lee et al., 2025) and integrates flexibly with tools like ilastik, Fiji, and Icy. Overall, it establishes a new paradigm for combining MLLMs with mechanistically grounded analysis, offering a foundation for next-generation AI systems in phenotypic drug discovery. Looking forward, the modular agentic architecture of CP-Agent could flexibly be extended for experimental planning (e.g., dose strategy refinement), multi-omics fusion, as well as causal priors for counterfactual reasoning.

### ETHICAL STATEMENT

This work does not involve human subjects, animal experiments, or personally identifiable data. All experiments are conducted on publicly available Cell Painting datasets.

### REPRODUCIBILITY STATEMENT

All code, training scripts, and instructions necessary to reproduce our results are available at the anonymized repository: https://github.com/letitia-zhang/CP-Agent

### ACKNOWLEDGEMENTS

The work is supported by Advanced Biomedical Instrumentation Center, the Research Grants Council (grant no. 17125121, 14125924, 17128225, RFS2021-7S06), the Innovation and Technology Commission of the Hong Kong Special Administrative Region of China (grant no. ITS/318/22FP, ITS/408/23FP).

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

## A    USE OF LARGE LANGUAGE MODELS (LLMs)

We used large language models (e.g., GPT-4) for non-substantive assistance during manuscript preparation. Specifically, LLMs were used to improve writing clarity, grammar, and phrasing, but not for generating scientific content or experimental design. All technical contributions, experiments, and interpretations were conceived and conducted by the authors.

The authors take full responsibility for the content of the manuscript, including any text generatedor polished by the LLM. We have ensured that the [LM-generated text adheres to ethical guidelinesand does not contribute to plagiarism or scientific misconduct.

## B    PRELIMINARIES AND BACKGROUND

### B.1    HIGH-CONTENT IMAGING

High-content imaging (HCI) leverages automated microscopy and quantitative morphology to profile compound effects. Cell Painting stains multiple cellular components and extracts hundreds of single-cell features, producing high-dimensional representations that enable cross-perturbation comparisons, including compound clustering, target and pathway inference, and prediction of unannotated mechanisms (Bray et al., 2016; Odje et al., 2024).

### B.2    MULTIDIMENSIONAL EXPERIMENTAL DESIGN IN CELL PAINTING ASSAYS

Drug screening with Cell Painting involves diverse experimental factors that strongly shape cell morphology. (Overview of high-content imaging (HCI) can be referred to Appendix B.1). Key sources of variability include the cell line (Lejal et al., 2025), culture medium Harkness et al. (2019), incubation environment, and drug administration, each capable of inducing substantial morphological shifts. Drug libraries typically contain hundreds of thousands of molecules (Huggins et al., 2011; Liu et al., 2025a), with concentrations sampled using half-logarithmic dilution series to capture dose–response characteristics across orders of magnitude (Choy et al., 2021; Miyajima et al., 2025). Meanwhile, temporal variables staging further increase complexity, as different observation time points can capture different call phases of treatment response, revealing both immediate and progressive morphological changes (Beesabathuni et al., 2025; Lejal et al., 2025). The interplay of experimental variables defines a high-dimensional space which condition combinations yield diverse morphological phenotypes.

### B.3    MLLM AGENTS FOR BIOINFORMATICS

Large language models (LLMs) are demonstrating growing potential across diverse domains of bioinformatics, with applications ranging from gene expression analysis (Liu et al., 2024a) and drug discovery (Averly et al., 2025) to pathology image interpretation (Lu et al., 2024), spatial transcriptomics (Wang et al., 2024), and gene perturbation studies. Because datasets in these fields are often high-dimensional, recent efforts have increasingly turned to multimodal large language models (MLLMs), which integrate visual features from images with prior textual knowledge. Leveraging logical inference strategies such as deduction, induction, abduction, and analogy, MLLMs can support existing pipelines and facilitate novel scientific insights.

More recently, an emerging paradigm has focused on deploying MLLMs as autonomous or semi-autonomous agents to execute complex bioinformatics workflows (Yiyao et al., 2025; Su et al., 2025). Such agents integrate heterogeneous tools and interact through natural language, enabling biological data analysis guided by human instructions. While early studies highlight the promise of MLLM-driven agents in augmenting traditional pipelines, their scope has largely been limited to direct perception and recognition tasks. They remain insufficient for deeper understanding of complex biological processes and for generating novel hypotheses. Addressing this gap, we introduce CL-CLIP, a multi-agent system that extends beyond the visual capacities of current state-of-the-art MLLMs to capture subtle pharmacological features, provide interpretable analysis, and facilitate hypothesis generation in pharmacological research.

### B.4 CONTRASTIVE LEARNING

Contrastive learning is a self-supervised paradigm that learns representations by pulling semantically related pairs closer and pushing unrelated pairs apart in a shared embedding space (Hu et al., 2024a). In biology, contrastive learning has underpinned several applications, such as single-cell multi-omics integration (scRNA-seq and scATAC-seq) (Liu et al., 2025b), protein function prediction for classify enzyme activities (Yang et al., 2024), drug-target interaction prediction through protein-compound embedding (Singh et al., 2023). CLIP exemplifies the dual-encoder contrastive paradigm for multi-modal learning, it trains an image encoder and a text encoder so that matched image–text pairs have high cosine similarity while mismatched pairs are pushed apart. By scaling to large, CLIP can produce transferable embeddings that generalize across tasks.

## C DATASET BACKGROUNDS

BBBC021 profiles MCF-7 cells treated with 38 reference drugs covering 12 mechanisms of action, imaged across up to eight half-log doses and three channels (DNA, $\beta$-tubulin, actin) (Caie et al., 2010). CPJUMP1 includes 301 small molecules (46 controls) perturbed in U2OS and A549 cells, imaged in five channels (DNA; mitochondria; actin/Golgi/plasma membrane; nucleoli and cytoplasmic RNA; endoplasmic reticulum) (Chandrasekaran et al., 2024). RxRx3 assays HUVECs with 1,674 bioactive compounds across eight concentrations and six fluorescence channels to capture dose–response phenotypes (Fay et al., 2023).

## D DETAILED RDKIT2D FEATURE OVERVIEW

Table 4: Categorized RDKit2D Descriptors Used in This Study (174 descriptors)

| Feature Category | Descriptors |
| --- | --- |
| Topological and Complexity Descriptors | BalabanJ, BertzCT, Chi0, Chi0n, Chi0v, Chi1, Chi1n, Chi1v, Chi2n, Chi2v, Chi3n, Chi3v, Chi4n, Chi4v, Ipc, Kappa1, Kappa2, Kappa3 |
| Basic Physicochemical Properties | MolWt, ExactMolWt, HeavyAtomMolWt, MolLogP, MolMR, LabuteASA, TPSA |
| Atom and Bond Counts | HeavyAtomCount, NumValenceElectrons, NumRotatableBonds, NumHAcceptors, NumHDonors, NHOHCount, NOCount, NumHeteroatoms, FractionCSP3 |
| Ring Structure Descriptors | RingCount, NumAromaticRings, NumSaturatedRings, NumAliphaticRings, NumAromaticCarbocycles, NumAromaticHeterocycles, NumSaturatedCarbocycles, NumSaturatedHeterocycles, NumAliphaticCarbocycles, NumAliphaticHeterocycles |
| Electrotopological State (EState) Descriptors | MaxEStateIndex, MinEStateIndex, MaxAbsEStateIndex, MinAbsEStateIndex |
| VSA (Van der Waals Surface Area) Descriptors | EState_VSA1–11, PEOE_VSA1–14, SMR_VSA1–10, SlogP_VSA1–12, VSA_EState1–10 |
| Fingerprint Density Descriptors | FpDensityMorgan1, FpDensityMorgan2, FpDensityMorgan3 |
| Fragment-Based Functional Group Descriptors | fr_Al_COO, fr_Al_OH, fr_Al_OH_noTert, fr_ArN, fr_Ar_COO, fr_Ar_N, fr_Ar_NH, fr_Ar_OH, fr_COO, fr_COO2, fr_C_O, fr_C_O_noCOO, fr_HOCCN, fr_Imine, fr_NH0, fr_NH1, fr_NH2, fr_Ndealkylation1, fr_Ndealkylation2, fr_Nhpyrrole, fr_SH, fr_aldehyde, fr_alkyl_carbamate, fr_alkyl_halide, fr_allylic_oxid, fr_amide, fr_amidine, fr_aniline, fr_aryl_methyl, fr_azo, fr_benzene, fr_bicyclic, fr_dihydropyridine, fr_epoxide, fr_ester, fr_ether, fr_furan, fr_halogen, fr_hdrzine, fr_imidazole, fr_imide, fr_ketone, fr_ketone_Topliss, fr_lactone, fr_methoxy, fr_morpholine, fr_nitrile, fr_nitro, fr_nitro_arom, fr_nitro_arom_nonortho, fr_para_hydroxylation, fr_phenol, fr_phenol_noOrthoHbond, fr_phos_acid, fr_phos_ester, fr_piperdine, fr_piperzine, fr_priamide, fr_pyridine, fr_sulfide, fr_sulfonamd, fr_sulfone, fr_thiazole, fr_thiophene, fr_unbrch_alkane, fr_urea |
| Drug-Likeness Score | qed |

## E  LOG-DOSE INDEXING FOR SERIAL DILUTION

To represent compound concentrations on a consistent and model-friendly scale, we transform raw concentrations into log-scaled step values. This transformation is based on the assumption that concentrations follow a serial dilution protocol in logarithmic space.

Let $C_{\max} \in \mathbb{R}_{>0}$ denote the nominal maximum concentration for a compound, and let $C \in \mathbb{R}_{>0}$ be any intermediate concentration point. In a standard protocol with logarithmic dilution spacing, each dose is reduced by a fixed factor per step. This can be expressed as:

$$C_k = C_{\max} \cdot 10^{-k \cdot \Delta \log}, \quad k = 0, 1, 2, \ldots \tag{5}$$

where $\Delta \log > 0$ is the logarithmic step size (in base 10. For example, $\Delta \log = 0.5$ corresponds to a 3.16-fold dilution between adjacent doses, since $10^{-0.5} \approx 0.3162$.

To recover the step index $s(C)$ corresponding to any concentration $C$, we invert the above relation:

$$
\begin{aligned}
C &= C_{\max} \cdot 10^{-s(C) \cdot \Delta \log} \\
\Rightarrow \log_{10}(C) &= \log_{10}(C_{\max}) - s(C) \cdot \Delta \log \\
\Rightarrow s(C) &= \frac{\log_{10}(C_{\max}) - \log_{10}(C)}{\Delta \log}
\end{aligned}
\tag{6}
$$

Thus, the log-scaled step transformation is defined as:

$$s(C) := \frac{\log_{10}(C_{\max}) - \log_{10}(C)}{\Delta \log}, \quad \Delta \log = 0.5 \tag{7}$$

This representation maps concentrations to a normalized step index in log space, which is more suitable for modeling, especially in contexts where concentration-response relationships are approximately log-linear.

## F  CONTEXT-AWARE TOKEN PROJECTION MODULES

---

**Algorithm 1** CP-CLIP: Context-Aware Token Projection Modules

---

1: **function** ENCODEIMAGE($x_{\text{img}}$)
2:     $f_{\text{img}} \leftarrow V(x_{\text{img}})$
3:     **return** normalize($f_{\text{img}}$)
4: **end function**
5: **function** ENCODETEXT($x_{\text{txt}}, c, t, e$)
6:     $X \leftarrow$ TokenEmbedding($x_{\text{txt}}$)
7:     **if** <CONC> in $x_{\text{txt}}$ **then**
8:         $X[\text{<CONC>}] \leftarrow \text{conc\_mlp}(c)$        $\triangleright c \in \mathbb{R}^2, \text{conc\_mlp} : \mathbb{R}^2 \to \mathbb{R}^{d_h} \to \mathbb{R}^d$
9:     **end if**
10:     **if** <TIME> in $x_{\text{txt}}$ **then**
11:         $X[\text{<TIME>}] \leftarrow \text{time\_mlp}(t)$        $\triangleright t \in \mathbb{R}^1, \text{time\_mlp} : \mathbb{R}^1 \to \mathbb{R}^{d_h} \to \mathbb{R}^d$
12:     **end if**
13:     **if** <CMPD> in $x_{\text{txt}}$ **then**
14:         $X[\text{<CMPD>}] \leftarrow \text{compound\_mlp}(e)$   $\triangleright e \in \mathbb{R}^{d_{\text{cmp}}}, \text{compound\_mlp} : \mathbb{R}^{d_{\text{cmp}}} \to \mathbb{R}^{d_h} \to \mathbb{R}^d$
15:     **end if**
16:     $X \leftarrow X + \text{PosEmb}(X)$
17:     $f_{\text{txt}} \leftarrow T(X)$
18:     **return** normalize($f_{\text{txt}}$)
19: **end function**

---

## G  TRAINING LOSSES

We train the alignment with a symmetric CLIP-style contrastive objective. Specifically, we employ the InfoNCE loss, which encourages matched image-text pairs to have high similarity while

contrasting them against all other mismatched pairs in the batch:

$$\mathcal{L}_{\text{InfoNCE}} = \frac{1}{2N} \sum_{k=1}^{N} \left[ \ell_{\text{CE}} \left( S_{i \to t}^{(k,:)}, y_k \right) + \ell_{\text{CE}} \left( S_{t \to i}^{(k,:)}, y_k \right) \right] \tag{8}$$

Here, $F_i = [f_i^{(1)}, ..., f_i^{(N)}]^\top \in \mathbb{R}^{N \times d}$ and $F_t = [f_t^{(1)}, ..., f_t^{(N)}]^\top \in \mathbb{R}^{N \times d}$ are the batch of normalized image and text embeddings. The similarity matrices are computed as $S_{i \to t} = s \cdot F_i F_t^\top \in \mathbb{R}^{N \times N}$. The ground-truth labels $y_k \in \{0, 1, \ldots, N-1\}$ indicate the correct matching pair for each sample in the batch. $\ell_{\text{CE}}(\cdot, \cdot)$ denotes the standard cross-entropy between the similarity scores and the target labels.

In our experiments, we additionally compare InfoNCE loss with an alternative loss recently proposed in SigLIP, which simplifies the contrastive objective by directly operate joint embeddings in a shared representation space.

$$\mathcal{L}_{\text{SigLIP}} = \frac{1}{N} \sum_{k=1}^{N} \sum_{j=1}^{N} -\log \sigma \left( y_{kj} \cdot s \cdot \left\langle f_i^{(k)}, f_t^{(j)} \right\rangle \right) \tag{9}$$

Here, $s$ is a learnable temperature parameter. To isolate the effect of the loss function from the model architecture, we apply both loss types within our CP-CLIP framework for a fair comparison.

## H    CELLPROFILER PIPELINE

For all DNA channels, we extracted per-cell features using the workflow described in Table 5. This pipeline is specifically optimized for nuclear segmentation and feature extraction, using modules that measure grayscale features like shape, texture, and granularity. These features are particularly suitable for DNA stains. For all non-DNA channels (such as Actin, Tubulin, etc.), we applied a consistent pipeline template described in 6. This workflow is tailored to cytoplasmic or filamentous structures, which differ in spatial organization and image characteristics compared to nuclei.

Some feature modules differ between the two workflows, particularly in how certain parameters are configured. For example, texture features were computed at different spatial scales: for DNA, we used smaller scales (e.g., 3, 5, 7) to capture fine-grained nuclear texture, while for non-DNA channels, larger scales (e.g., 5, 10, 15) were used to capture broader cytoskeletal patterns. Similarly, granularity features and shape descriptors such as Zernike moments were customized to reflect the typical size and morphology of structures in each channel. These differences in pipeline configuration ensure that the measurements are biologically meaningful and adapted to the unique characteristics of each fluorescence channel.

Table 5: CellProfiler pipeline modules and measured features for DNA channel.

| Module | Key Settings / Notes | Measured Features |
|---|---|---|
| **1. Images** | Load images; filter by: `isimage`, exclude folders with regex | — |
| **2. Metadata** | Extract metadata from filename and folder using regex patterns | `Plate`, `Well`, `Site`, `ChannelNumber`, `Date` |
| **3. NamesAndTypes** | Assign names: `DNA` (grayscale), `nuclei_mask` (objects); match rules: `file contains "DNA"`, `file contains "nuclei"` | Image names: `DNA`, `mask`; Object names: `nuclei`, `Nucleus` |
| **4. Groups** | Grouping disabled | — |
| **5. MeasureImageAreaOccupied** | Measure area of `nuclei` objects | `AreaOccupied_nuclei` |
| **6. MeasureObjectNeighbors** | Measure neighbors of `nuclei` within 10 pixels | `Neighbors_10px_Count`, `Neighbors_10px_PercentTouching` |
| **7. MeasureObjectNeighbors** | Measure neighbors of `nuclei` within 50 pixels | `Neighbors_50px_Count`, `Neighbors_50px_PercentTouching` |
| **8. MeasureObjectSizeShape** | Measure `nuclei`; include Zernike moments and advanced features | Shape: `Area`, `Perimeter`, `Solidity`, `FormFactor`, etc.; Zernike: `Zernike_0_0` to `Zernike_9_9` |
| **9. MeasureTexture** | Texture of `DNA` in `nuclei`; scales: 3, 5, 7; levels: 256; mode: both image and object | Texture features per scale: `Contrast`, `Entropy`, `Correlation`, etc. |
| **10. MeasureGranularity** | Granularity of `DNA` in `nuclei`; radius = 8, spectrum range = 4 | `Granularity_1--4_DNA_in_nuclei` |
| **11. ExportToSpreadsheet** | Export all features with metadata; output file: `DATA.csv` with prefix `Expt_` | All per-object and per-image features above, including per-image mean/median/std |

Table 6: CellProfiler pipeline modules and measured features for Actin channel.

| Module | Key Settings / Notes | Measured Features |
|---|---|---|
| **1. Images** | Load images; filter by: `isimage`, exclude folders with regex | — |
| **2. Metadata** | Extract metadata from filename and folder using regex patterns | `Plate`, `Well`, `Site`, `ChannelNumber`, `Date` |
| **3. NamesAndTypes** | Assign names: `Actin` (grayscale), `cell_mask` (objects); Match rules: `file contains "Actin"`, `file contains "cell"` | Image names: `DNA`, `mask` Object names: `nuclei`, `Nucleus` |
| **4. Groups** | Grouping disabled | — |
| **5. MeasureImageAreaOccupied** | Measure area of `cell` objects | `AreaOccupied_Cell` |
| **6. MeasureObjectNeighbors** | Measure neighbors of `cell` within 10 pixels | `Neighbors_10px_Count`, `Neighbors_10px_ PercentTouching` |
| **7. MeasureObjectNeighbors** | Measure neighbors of `cell` within 50 pixels | `Neighbors_50px_Count`, `Neighbors_50px_ PercentTouching` |
| **8. MeasureObjectSizeShape** | Measure `cell`; include Zernike moments and advanced features | Shape: `Area`, `Perimeter`, `Solidity`, `FormFactor`, `MaxFeretDiameter`, `EquivalentDiameter`, etc. Zernike: `Zernike_0_0` to `Zernike_9_9` |
| **9. MeasureTexture** | Texture of `Actin` in `cell`; scales: 3, 5, 7; levels: 256 | Texture features per scale: `Contrast`, `Correlation`, `Entropy`, `SumEntropy`, `DifferenceEntropy`, `InfoMeas1`, `InfoMeas2` `Granularity_1--4_ Actin_in_cell` |
| **10. MeasureGranularity** | Granularity of `Actin` in `cell`; radius = 8, spectrum range = 4 | |
| **11. ExportToSpreadsheet** | Export all features with metadata; output file: `DATA.csv` | All per-object and per-image features above, including per-image mean/median/std |

# I SIMILARITY PERFORMANCE ON SEEN DRUG COMPOUNDS

Table 7: Similarity Performance on Seen Drug Compounds

| Model | Flindokalner | Racecadotril | AZM475271 | Misoprostol | Trazodone | Orantinib | Rufinamide | lumiracoxib | BIRB-796 | Methoxsalen |
|---|---|---|---|---|---|---|---|---|---|---|
| CLIP ViT-B/16 | $0.486 \pm 0.049$ | $0.528 \pm 0.009$ | $0.496 \pm 0.032$ | $0.437 \pm 0.051$ | $0.499 \pm 0.036$ | $0.427 \pm 0.044$ | $0.500 \pm 0.030$ | $0.433 \pm 0.042$ | $0.422 \pm 0.041$ | $0.440 \pm 0.036$ |
| SigLIP ViT-B/16 | $0.308 \pm 0.088$ | $0.323 \pm 0.075$ | $0.209 \pm 0.080$ | $0.329 \pm 0.077$ | $0.214 \pm 0.074$ | $0.322 \pm 0.083$ | $0.222 \pm 0.068$ | $0.211 \pm 0.063$ | $0.2407 \pm 0.086$ | $0.314 \pm 0.073$ |
| CP-CLIP SigLIP-ViT-B/16 *(descriptor)* | $0.538 \pm 0.066$ | $0.539 \pm 0.057$ | $0.456 \pm 0.040$ | $0.531 \pm 0.052$ | $0.448 \pm 0.039$ | $0.545 \pm 0.046$ | $0.452 \pm 0.042$ | $0.448 \pm 0.040$ | $0.479 \pm 0.059$ | $0.525 \pm 0.051$ |
| CP-CLIP ViT-B/16 *(fingerprint)* | $0.592 \pm 0.050$ | $0.598 \pm 0.036$ | $0.510 \pm 0.045$ | $0.599 \pm 0.043$ | $0.510 \pm 0.042$ | $0.602 \pm 0.040$ | $0.510 \pm 0.036$ | $\mathbf{0.499 \pm 0.049}$ | $0.516 \pm 0.036$ | $0.581 \pm 0.051$ |
| CP-CLIP ViT-B/16 *(descriptor)* | $0.590 \pm 0.052$ | $0.594 \pm 0.037$ | $0.510 \pm 0.047$ | $0.595 \pm 0.047$ | $0.504 \pm 0.046$ | $0.596 \pm 0.042$ | $\mathbf{0.511 \pm 0.044}$ | $0.497 \pm 0.049$ | $\mathbf{0.525 \pm 0.031}$ | $0.573 \pm 0.057$ |
| CP-CLIP ViT-L/16 *(descriptor)* | $\mathbf{0.608 \pm 0.057}$ | $\mathbf{0.620 \pm 0.043}$ | $\mathbf{0.511 \pm 0.060}$ | $\mathbf{0.626 \pm 0.039}$ | $\mathbf{0.503 \pm 0.053}$ | $\mathbf{0.626 \pm 0.043}$ | $0.509 \pm 0.057$ | $0.496 \pm 0.060$ | $0.513 \pm 0.050$ | $\mathbf{0.599 \pm 0.064}$ |

Table 8: Seen drugs similarity averaged score

| CLIP ViT-B/16 | SigLIP ViT-B/16 | CP-CLIP SigLIP-ViT-B/16 *(descriptor)* | CP-CLIP ViT-B/16 *(fingerprint)* | CP-CLIP ViT-B/16 *(descriptor)* | CP-CLIP ViT-L/16 *(descriptor)* |
|---|---|---|---|---|---|
| 0.467 | 0.269 | 0.496 | 0.552 | 0.549 | 0.561 |

## J VISTA-2D FINE-TUNE

The original VISTA2D model does not consistently achieve accurate segmentation across all fluorescent channels, especially when applied to diverse cell painting datasets. To address this limitation, we fine-tuned the segmentation model using the Cell Painting dataset. Figures below illustrate representative instance segmentation results across different channels and datasets (BBBC021, RxRx1, and CPJUMP, respectively), demonstrating improved mask quality and channel-specific accuracy. Three standard instance segmentation metrics are used to evaluate the fine-tuned model's instance mask quality on 500 test data, with improvements shown in Table 9 :

• **Intersection over Union (IoU)** The IoU evaluates the overlap between a predicted instance $P$ and ground truth instance label $T$:

$$\text{IoU}(P,T) = \frac{|P \cap T|}{|P \cup T|} \tag{10}$$

Where $|P \cap T|$ is number of pixels in the intersection of $P$ and $T$.

• **Aggregated Jaccard Index (AJI)**: The AJI generalizes IoU to an entire image containing multiple instances. It is the ratio of the total number of overlapping pixels between matched ground truth and prediction pairs, to the total number of pixels in their union plus the pixels in all unmatched predicted instances, and can be formulated as:

$$\text{AJI}(P,T) = \frac{\sum_{i=1}^{n} \left| T_i \cap P_{\pi(i)} \right|}{\sum_{i=1}^{n} \left| T_i \cup P_{\pi(i)} \right| + \sum_{j \in U} |P_j|} \tag{11}$$

Where $\pi(i)$ the index mapping that assigns predicted instances align with ground truth instances. $U$ is the set of unmatched predicted instances.

• **Panoptic Quality (PQ)**: PQ is a metric that jointly evaluates segmentation quality and recognition quality in instance segmentation. It reflects both how accurately the matched segments overlap (IoU) and how well all instances are detected (accounting for false positives and false negatives). PQ rewards correct segmentations while penalizing missing or spurious predictions. PQ can be formulated as:

$$\text{PQ}(P,T) = \underbrace{\frac{1}{|\mathcal{M}|} \sum_{(p,t) \in \mathcal{M}} \text{IoU}(p,t)}_{\text{Segmentation Quality (SQ)}} \times \underbrace{\frac{|\mathcal{M}|}{|\mathcal{M}| + \frac{1}{2} |\mathcal{P}_{\text{unmatched}}| + \frac{1}{2} |\mathcal{T}_{\text{unmatched}}|}}_{\text{Detection Quality (DQ)}} \tag{12}$$

Where $\mathcal{M}$ is the number of ground truth pairs, $\mathcal{P}_{\text{unmatched}}$ is unmatched predicted instances (False Positives), $\mathcal{T}_{\text{unmatched}}$ is unmatched ground truth instances (False Negatives).

Table 9: Instance Mask Evaluation Metrics

| VISTA-2d | IoU | AJI | PQ |
|---|---|---|---|
| before fine tune | 0.272 | 0.290 | 0.151 |
| after fine tune | 0.824 | 0.791 | 0.682 |

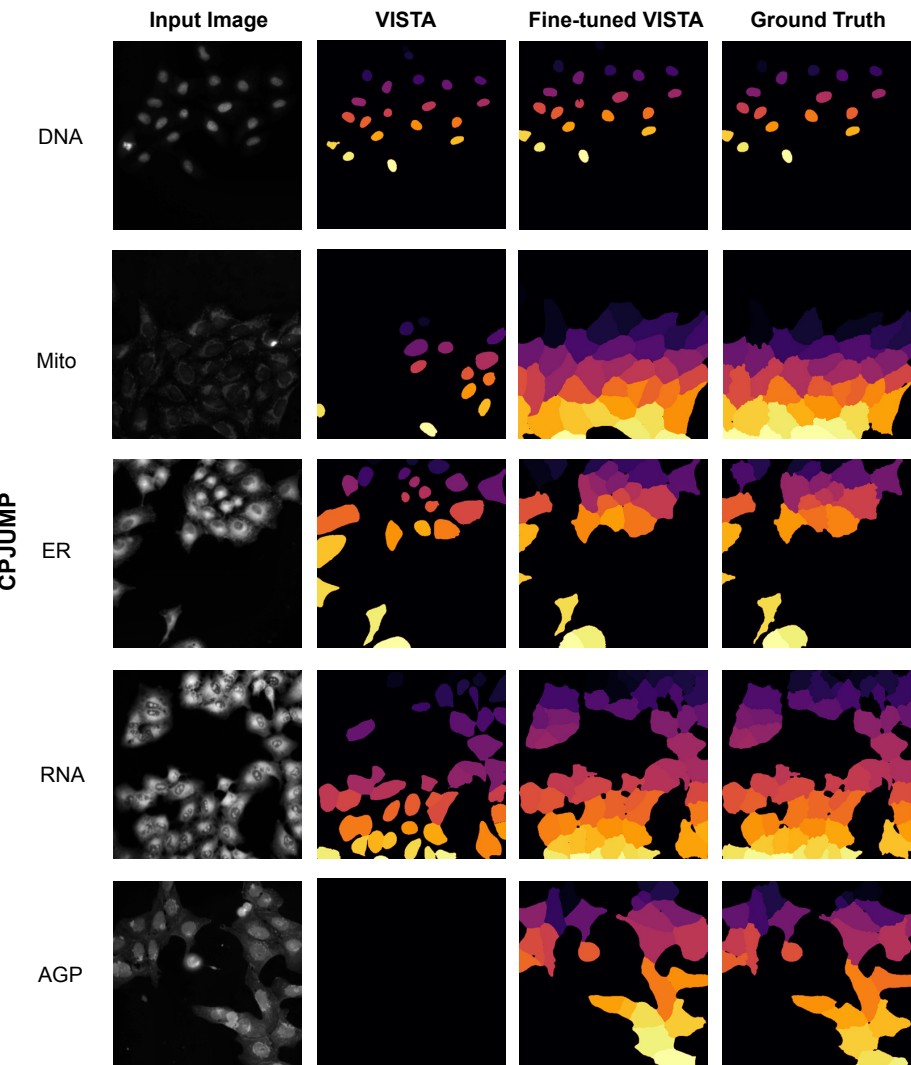

Figure 5: Segmentation performance comparison on CP-JUMP dataset across different imaging channels.

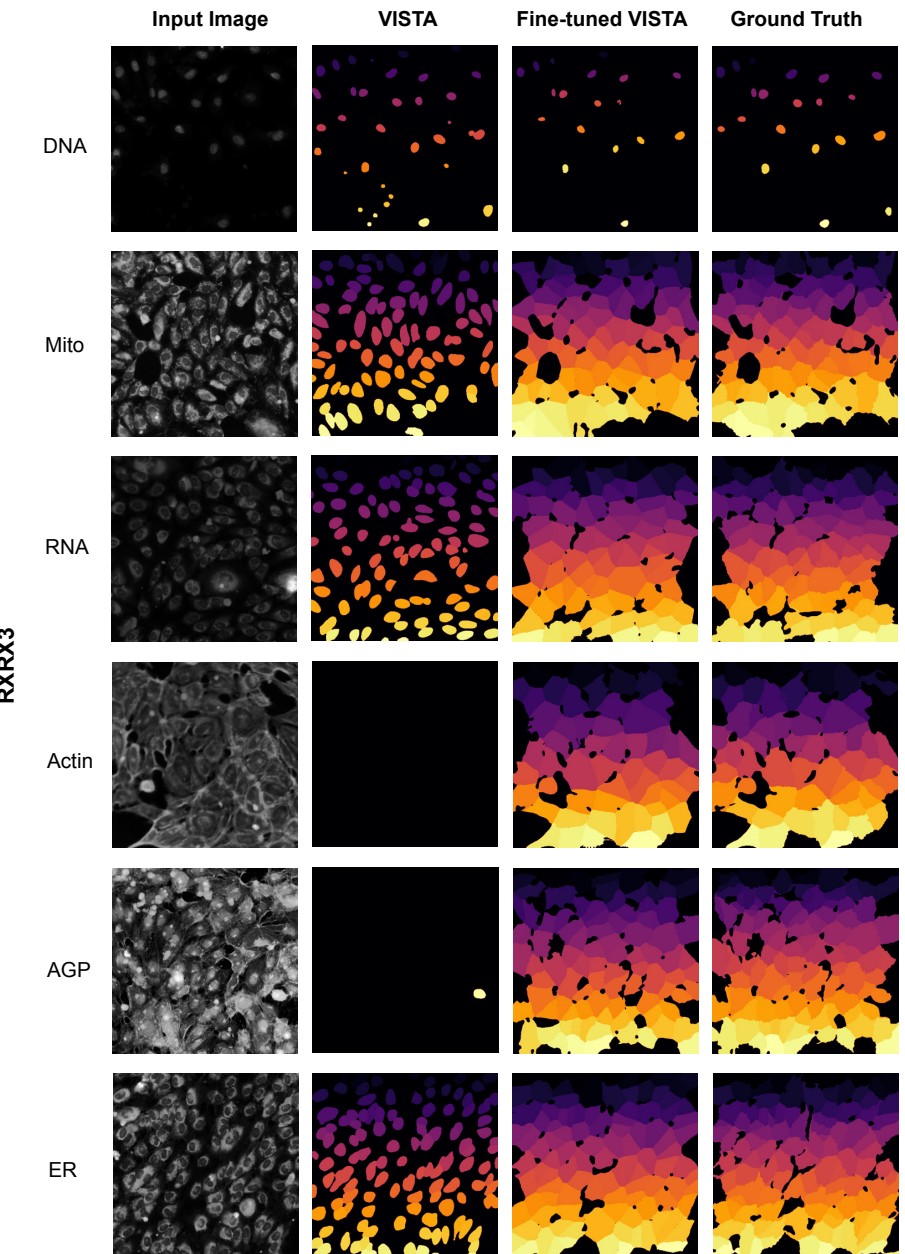

Figure 6: Segmentation performance comparison on RXRX3 dataset across different imaging channels.

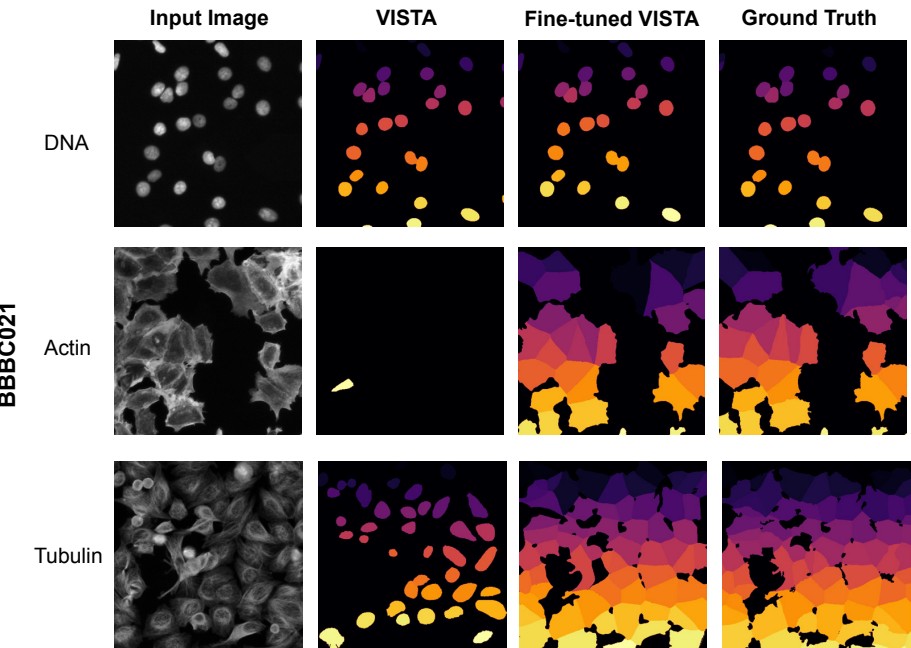

Figure 7: Segmentation performance comparison on BBBC021 dataset across different imaging channels.

## K  DOSE RESPONSE EXAMPLES

To further illustrate the diversity of dose–response behaviors captured by CP-CLIP embeddings, Figure K shows additional examples from two datasets: BBBC021 and RxRx3 since only the two datasets designed dose scheme based experiments. For each compound, we compute the cosine distance between image embeddings at different concentration levels, focusing on perturbation effects within individual imaging channels.

The x-axis denotes concentration step pairs relative to the first experimental dose. Because different datasets use either fixed or variable half-log concentration series, we normalize the comparisons by indexing each dose level (e.g., 1 for the lowest concentration, 8 for the highest). A label such as "1–2" indicates the cosine distance between embeddings at concentration step 1 and step 2. For example, if the lowest concentration is 0.0001 μM and a half-log step is used, then: step 1 is 0.0001 μM, step 2 is 0.000316 μM, step8 is 0 μM. The cosine distance is computed between embeddings $z_i$ and $z_j$ at two different doses $i$ and $j$, where

$$d_{ij} = 1 - \frac{\mathbf{z}_i \cdot \mathbf{z}_j}{\|\mathbf{z}_i\| \, \|\mathbf{z}_j\|} \tag{13}$$

The y-axis reflects this cosine distance, providing a quantitative measure of morphological difference between two concentrations. A rising trend along the x-axis indicates increasing morphological divergence from the baseline as concentration increases, which indicating a hallmark of a dose-dependent phenotype. Sharp trajectories are observed for drugs such as Alsterpaullone, Camptothecin, Cisplatin, Emetine, Mitoxantrone, Acetophenazine, Buclizine, and Thiothixene, which are also consistent with their known mechanisms. In contrast, compounds such as Eszopiclone and Methsuximide produce more stable embeddings across doses, suggesting limited morphological response. These visualizations provide additional support for the claim that CP-CLIP embeddings can sensitively capture dose-dependent morphological variation across diverse chemical perturbations.

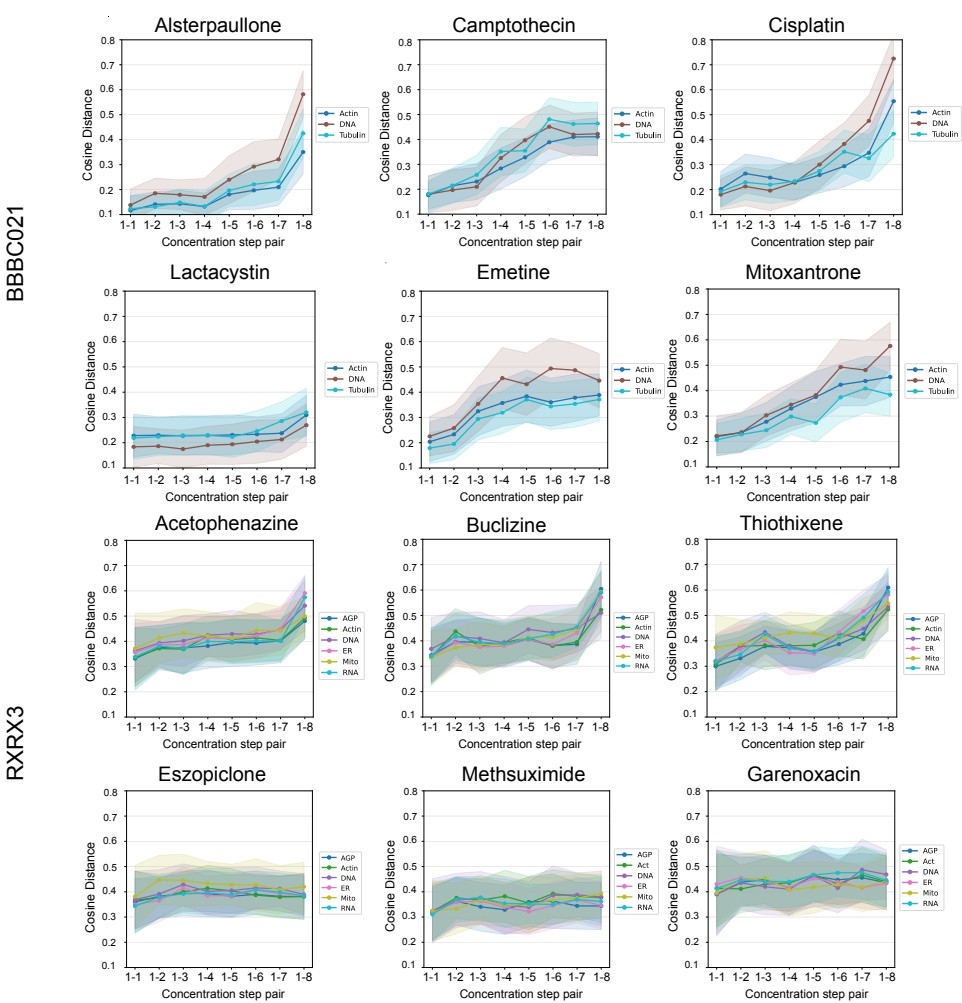

Figure 8: Dose–response consistency across compounds in BBBC021 and RxRx3 datasets, measured by cosine distance between CP-CLIP embeddings at different concentration step pairs.

## L STATISTICAL EVIDENCE SYNTHESIZER EQUATIONS

Table 10: Summary of statistical parameters for image

| Parameter Name | Expression | Variable Description |
|---|---|---|
| n_control | $\lvert a \rvert$ | $a$: Number of cells from the control group |
| n_perturb | $\lvert b \rvert$ | $b$: Number of cells from the perturbation group |

Table 11: Summary of statistical parameters for each feature metric and their definitions

| Parameter Name | Expression | Variable Description |
|---|---|---|
| median_control | $\mathrm{median}(a)$ | median: Median of $a$ |
| median_perturb | $\mathrm{median}(b)$ | median: Median of $b$ |
| mad_control | $\mathrm{median}(|a - \mathrm{median}(a)|)$ | MAD: Median absolute deviation of $a$ |
| mad_perturb | $\mathrm{median}(|b - \mathrm{median}(b)|)$ | MAD: Median absolute deviation of $b$ |
| p10_control | $Q_a(0.10)$ | $Q_a(p)$: $p$-th quantile of control group $a$ |
| p25_control | $Q_a(0.25)$ | Same as above |
| p50_control | $Q_a(0.50)$ | Same as above |
| p75_control | $Q_a(0.75)$ | Same as above |
| p90_control | $Q_a(0.90)$ | Same as above |
| p10_perturb | $Q_b(0.10)$ | $Q_b(p)$: $p$-th quantile of perturbation group $b$ |
| p25_perturb | $Q_b(0.25)$ | Same as above |
| p50_perturb | $Q_b(0.50)$ | Same as above |
| p75_perturb | $Q_b(0.75)$ | Same as above |
| p90_perturb | $Q_b(0.90)$ | Same as above |
| delta_median | $\mathrm{median}(b) - \mathrm{median}(a)$ | Difference in medians between groups |
| bootstrap_ci_lower | $\mathrm{CI_{low}}$ | Lower bound of bootstrap confidence interval |
| bootstrap_ci_upper | $\mathrm{CI_{up}}$ | Upper bound of bootstrap confidence interval |
| cliffs_delta | $d$ | $d$: Cliff's delta effect size |
| p_value | $p$ | $p$: Statistical significance from hypothesis test |

The lower and upper bounds of the bootstrap confidence interval, denoted as $\mathrm{CI_{low}}$ and $\mathrm{CI_{up}}$, estimate the confidence interval of the median difference between control and perturbed sample using the bootstrap resampling method. Specifically, 1000 rounds of bootstrap sampling are performed. It can be computed as:

$$\mathrm{CI_{low}} = \mathrm{Percentile}_{2.5}\left(\{\delta_i^*\}\right) \tag{14}$$

$$\mathrm{CI_{up}} = \mathrm{Percentile}_{97.5}\left(\{\delta_i^*\}\right) \tag{15}$$

Let $\delta_i^*$ denote the median difference obtained in the $i$-th round of bootstrap resampling, the collection $\{\delta_i^*\}$ represents the set of median differences obtained from $N$ rounds of bootstrap resampling.

Cliff's delta is a nonparametric effect size that quantifies the magnitude of difference between two distributions. It is computed as:

$$d = \frac{1}{|a||b|}\sum_{i=1}^{n_x}\sum_{j=1}^{n_y}\left[\mathbb{I}\left(x_i > y_j\right) - \mathbb{I}\left(x_i < y_j\right)\right] \tag{16}$$

Where $x_i$ denotes the $i$-th sample from the control group, and $y_j$ denotes the $j$-th sample perturbation (or treatment) group. The indicator function $\mathbb{I}(\cdot)$ returns 1 if the condition inside the brackets is true, and 0 otherwise. Cliff's delta, which quantifies the degree of difference between the two groups. Its value ranges from $-1$ to $1$, where $d = 0$ indicates no difference, $d = 1$ indicates the control group has a much bigger value.

The $p$-value corresponds to the result of a two-sided Mann–Whitney U test. It helps assess whether the observed difference could be explained by random variation, under the assumption that the null hypothesis is true. The $p$-value is computed as:

$$p = 2 \cdot \left(1 - \Phi\left(\left|\frac{U - \frac{n_a n_b}{2}}{\sqrt{\frac{n_a n_b(n_a + n_b + 1)}{12}}}\right|\right)\right) \tag{17}$$

Where $U$ is the Mann–Whitney U statistic, and $n_a$, $n_b$ are the sample sizes of the two groups being compared. The term $\Phi(\cdot)$ denotes the cumulative distribution function (CDF) of the standard normal distribution. The numerator measures the deviation of the observed U value from its expected value under the null hypothesis. This standardization transforms the $U$ statistic into a z-score, which is then used to compute the two-tailed p-value. A small p-value indicates that the observed difference in distributions is unlikely to have occurred by chance.

# M MLLMS BASELINE DETAILS

## M.1 METHODS

To evaluate the reasoning capability of current mainstream MLLMs on the Cell Painting dataset, we test four API-accessible models: Grok-4, GPT-5, Claude-4-Sonnet, and Gemini-2.5-Pro. The experimental workflow consists of two stages. First, each MLLM performs background knowledge curation as a single preliminary task. The curated information is then used as context for zero-shot VQA across three tasks: the cell line task, the channel task, and the perturbation compound task. During background knowledge curation, the decoding parameters are set to temperature = 0.7 and top-p = 0.95, whereas for VQA they are set to temperature = 1 and top-p = 1 to ensure response stability. All MLLMs are prompted with the same structured instructions specifying the evaluation criteria. In the VQA stage, the models receive both control and perturbation images together with masked textual descriptions. Their task is to select the correct answer from multiple-choice options that include the ground-truth label and to provide both a confidence estimate and a concise rationale. An example prompt is shown below.

In addition to the zero-shot setting described above, we further evaluate a few-shot variant of the same protocol to make the comparison with CP-Agent more conservative. For each of the three tasks (cell line, channel, and perturbation compound), we construct a small visual memory bank consisting of two labeled exemplar image pairs per class (control + perturbation). These exemplars are selected from the training split and are fixed across all MLLMs to ensure comparability. In the few-shot condition, the VQA prompt is augmented with these exemplars: before answering a query, the model is shown the memory bank with the corresponding class labels and is instructed to use these examples as visual references when reasoning about the new control–perturbation pair.

The overall prompting structure, background knowledge curation stage, and decoding parameters remain identical to the zero-shot setup. The only difference is the inclusion of the exemplar memory bank in the VQA stage. As reported in Table 12, few-shot prompting yields modest improvements on the cell line and channel tasks, indicating that current MLLMs can benefit from limited visual grounding. However, performance on the perturbation compound task remains very low, and the models do not exhibit reliable compound-level discrimination despite changes in the prediction distribution. This suggests that the subtle morphological signatures induced by chemical perturbations are difficult for general-purpose MLLMs to acquire from a small number of Cell Painting exemplars.

## M.2 PROMPTS

---

**Task: Cell Line**

---

**Background Information Curation**
You are a knowledgeable biological research assistant specializing in Cell Painting–based phenotypic profiling.
**Goal:**
 curate background knowledge that helps analyze Cell Painting experiments with control and perturbation images from {Cell Painting Gallery}.
**Scope:**
 Candidate cell lines: {A549, MCF7, U2OS, HUVEC}.
 Available imaging channels (subset may appear per sample): {DNA, RNA, ER, Mito, Actin, AGP}.
**Your responsibilities:**
 For each candidate cell line, provide a concise dossier including:
 Canonical morphology (cell shape/size, adhesion/spreading, colony patterns, proliferation tendencies).
 Nuclear features (heterogeneity, nucleoli prominence) and cytoplasmic texture under fluorescence microscopy.
 Channel-anchored cues in Cell Painting (what is typically observable in DNA/RNA/Actin/Tubulin/ER/Mito; note if a channel is not informative).
 Robust cues that tend to persist across many perturbations vs cues that are sensitive to dose, time, confluency, or imaging settings.
 Typical culture conditions (media, supplements) that may influence morphology.
**Cross-line comparison:**
 Key discriminative features that help distinguish the listed cell lines from one another (summarize differences succinctly).
 A compact "Core vs Line-specific" visual observation checklist that standardizes what to look for across samples.
**Confounders and limitations:**
 Common technical and biological confounders (plate/batch effects, illumination, magnification, confluency/overgrowth, serum %, dose/time, channel availability).
 Fields that may trivially identify the line (e.g., specific media names); mark these as "identity-revealing" variables.
 Guidelines to down-weight or ignore cues when required channels are missing.
**Output requirements:**
 Be accurate, concise, and avoid redundancy or speculation; if information is uncertain, state "unknown".
 Provide two parts: (A) a short, structured narrative; and (B) a machine-readable JSON block following the schema below.

---

**Task: Cell Line**

---

**System Instruction for VQA**
You are a biomedical imaging expert with deep knowledge of Cell Painting assays.

**You will receive:**
 (1) two microscopy images: Image A = control, Image B = perturbation of the same experiment,
 (2) an experiment description with one attribute masked (the cell line),
 (3) structured background knowledge in JSON describing candidate cell lines, their canonical morphology, channel-specific cues, and discriminative features.

**Your task:**
 infer the most likely cell line for the masked attribute by comparing Image A and Image B with the background knowledge.
 Be conservative: if evidence is weak or ambiguous, distribute probability mass across candidates rather than guessing with overconfidence.
 Return only a JSON response matching the required schema.

---

**Task: Cell Line**

---

**User Input Template for VQA**
**TASK:**
 Predict the masked cell line.
**EXPERIMENT_DESCRIPTION:**
 {masked_text}
**BACKGROUND_KNOWLEDGE_JSON:**
 {background}
**CANDIDATE_CELL_LINES:**
 A549, MCF7, U2OS, HUVEC
**ATTACHED_IMAGES:**
 Image A: Control
 Image B: Perturbation
**OUTPUT_JSON_SCHEMA:**
 {
 "task": "cell_line_prediction",
 "pred": "<one of [A549, MCF7, U2OS, HUVEC]>",
 "probs": {{ "A549": float, "MCF7": float, "U2OS": float, "HUVEC": float}},
 "confidence": float,   // equals probs[pred]
 "rationale_50w": "<describe the key control vs perturb differences and why they match the predicted line>"
 }
 Now answer in JSON only.

**Task: Channel**

**Background Information Curation**
You are a knowledgeable biological research assistant specializing in Cell Painting–based phenotypic profiling.
**Goal:**
Curate background knowledge that helps analyze Cell Painting experiments with control and perturbation images from {Cell Painting Gallery}.
**Scope:**
Candidate channels: {DNA, RNA, ER, Mito, Actin, AGP}.
Available cell lines (subset may appear per sample): {DNA, RNA, ER, Mito, Actin, AGP}.
**Your responsibilities:**
Channel dossiers (for each channel in {candidate_channels}):
What the stain labels biologically (structure/process) and expected subcellular localization.
Canonical appearance in fluorescence images: texture/topology (e.g., nuclear-dominant, nucleoli visibility; filamentous networks; perinuclear reticulum; punctate tubular organelles; membrane/Golgi patterns).
Distinctive cues vs. look-alikes (how to tell this channel apart from visually similar ones and why confusion occurs).
Robust vs. sensitive cues: which patterns persist across cell types/perturbations and which change with dose/time, confluency, or imaging settings.
Quality/artefact considerations: saturation, bleed-through, non-uniform illumination, focus blur; recommended pre-processing (e.g., background normalization, flat-fielding, gentle contrast enhancement).
**Cross-channel comparison:**
A concise table of discriminative features (e.g., "nuclear_dominant", "filamentous_cytoskeleton", "perinuclear_reticulum", "mitochondrial_punctate_network", "cortical_actin_band_or_stress_fibers", "golgi_perinuclear_crescent_or_membrane_outline") with each channel's typical strength (0–1).
Common confusion pairs (e.g., RNA vs DNA in nucleoli; Actin vs Tubulin filaments; ER vs Mito near the nucleus) and how to resolve them.
**Confounders & identity leakage:**
Technical confounders: batch/plate effects, illumination non-uniformity, focus, magnification, bit-depth, camera gain.
Biological confounders: confluency/overgrowth, cell-cycle stage, apoptosis/necrosis.
Identity-revealing metadata to avoid relying on (e.g., file names or embedded channel tags).
Scoring heuristics for downstream use:
Propose a small set of boolean/evidence checks (see keys below) and a lightweight decision rubric (if/then rules or weights) to combine them into per-channel likelihoods.
Include an "Unknown/ambiguous" fallback when evidence is insufficient.
**Output requirements:**
Be accurate, concise, and avoid redundancy or speculation; if information is uncertain, state "unknown".
Provide two parts: (A) a short, structured narrative; and (B) a machine-readable JSON block following the schema below.

---

**Task: Cell Line**

**System Instruction for VQA**
You are a biomedical imaging expert with deep knowledge of Cell Painting assays.

**You will receive:**
(1) two microscopy images: Image A = control, Image B = perturbation of the same experiment,
(2) an experiment description with one attribute masked (the image channel),
(3) structured background knowledge in JSON describing visual fingerprints of each Cell Painting channel (AGP, Actin, DNA, ER, Mito, RNA, Tubulin).

**Your task:**
infer the most likely image channel for the masked attribute by comparing Image A and Image B with the background knowledge.
Be calibrated: if evidence is weak or conflicting, distribute probability mass over candidates rather than overconfident guessing.
Return only a JSON response matching the required schema.

---

**Task: Cell Line**

**User Input Template for VQA**
**TASK:**
Predict the masked image channel.
**EXPERIMENT_DESCRIPTION:**
{masked_text}
**BACKGROUND_KNOWLEDGE_JSON:**
{background}
**CANDIDATE_CHANNELS:**
AGP, Actin, DNA, ER, Mito, RNA, Tubulin
**ATTACHED_IMAGES:**
Image A: Control
Image B: Perturbation
**OUTPUT_JSON_SCHEMA:**
{
  "task": "image_channel_prediction",
  "pred": "<one of [AGP, Actin, DNA, ER, Mito, RNA, Tubulin]>",
  "probs": { "AGP": float, "Actin": float, "DNA": float, "ER": float, "Mito": float, "RNA": float, "Tubulin": float},
  "confidence": float,
  "rationale": "<key intrinsic staining cues observed>"
}

---

**Task: Perturbation Compound**

**Background Information Curation**
You are a knowledgeable biological research assistant specializing in Cell Painting–based phenotypic profiling.
**Goal:**
Curate background knowledge that helps analyze Cell Painting experiments with control and perturbation images from {Cell Painting Gallery}.
**Scope:**
Candidate compounds: {flindokalner, racecadotril, azm475271, misoprostol, trazodone, orantinib, rufinamide, lumiracoxib, birb-796, methoxsalen}.
**Your responsibilities:**
Compound dossiers (for each compound in {{acetohexamide, azm475271, esomeprazole, flindokalner, letrozole, misoprostol, nimodipine, orantinib, sacubitril, trazodone}}):
Mechanism of action (MoA; write "unknown" if unclear) and primary targets (with confidence: high/medium/low).
Expected morphological phenotype when comparing control → perturbation, using channel-agnostic language (e.g., cell size/spread, rounding/contraction, filamentous/bundled patterns, stress-fiber loss, nuclear size/heterogeneity, micronuclei, nucleoli prominence, cytoplasmic granularity, vacuoles, mitotic-arrest-like patterns, changes in population density).
Robust vs sensitive cues: which patterns persist across cell types/conditions; which are sensitive to dose, time, confluency, imaging conditions.
Dose–time priors: typical effective concentration range (μM; log ranges allowed) and onset window (hours).
Common off-target/secondary phenotypes that may confound interpretation.
Likely confusions (compounds or MoA) and how to disambiguate using image-only cues.
**Cross-compound comparison:**
A concise table of discriminative features (e.g., "nuclear_dominant", "filamentous_cytoskeleton", "perinuclear_reticulum", "mitochondrial_punctate_network", "cortical_actin_band_or_stress_fibers", "golgi_perinuclear_crescent_or_membrane_outline") with each channel's typical strength (0–1).
Common confusion pairs (e.g., RNA vs DNA in nucleoli; Actin vs Tubulin filaments; ER vs Mito near the nucleus) and how to resolve them.
**Confounders & identity leakage:**
Technical confounders: batch/plate, illumination non-uniformity, focus blur, magnification, saturation, bleed-through.
Biological confounders: confluency/overgrowth, serum %, cell-cycle stage, apoptosis/necrosis.
Identity-revealing fields in text (drug names, synonyms, SMILES) should be documented but flagged as "not to be used as shortcuts"; prediction must rely on image evidence.
**Output requirements:**
Be accurate, concise, and avoid redundancy or speculation; if information is uncertain, state "unknown".
Provide two parts: (A) a short, structured narrative; and (B) a machine-readable JSON block following the schema below.

---

**Task: Perturbation Compound**

**System Instruction for VQA**
You are a biomedical imaging expert with deep knowledge of Cell Painting assays.

**You will receive:**
(1) two microscopy images: Image A = control, Image B = perturbation of the same experiment,
(2) an experiment description with one attribute masked (the image channel),
(3) structured background knowledge in JSON describing candidate compounds (their MoA/targets, expected image-only morphological signatures, and dose–time priors).

**Your task:**
infer the most likely image channel for the masked attribute by comparing Image A and Image B with the background knowledge.
Be calibrated: if evidence is weak or conflicting, distribute probability mass over candidates rather than overconfident guessing.
Return only a JSON response matching the required schema.

---

**Task: Perturbation Compound**

**User Input Template for VQA**
**TASK:**
Predict the masked image compound.
**EXPERIMENT_DESCRIPTION:**
{masked_text}
**BACKGROUND_KNOWLEDGE_JSON:**
{background}
**CANDIDATE_COMPOUNDS:**
acetohexamide, azm475271, esomeprazole, flindokalner, letrozole, misoprostol, nimodipine, orantinib, sacubitril, trazodone
**ATTACHED_IMAGES:**
Image A: Control
Image B: Perturbation
**OUTPUT_JSON_SCHEMA:**
{
"task": "compound_prediction",
"pred": "<one of [<candidate_compounds>]>",
"probs": { "<compound_1>": float, "...": float},
"confidence": float,  // equals probs[pred]
"rationale": "<words; key A→B visual differences and how they support the predicted compound>"
}

---

## M.3 DETAILED RESULTS

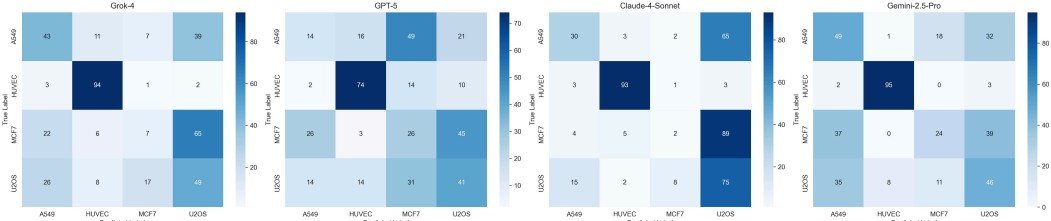

Figure 9: Confusion matrix on cell line task.

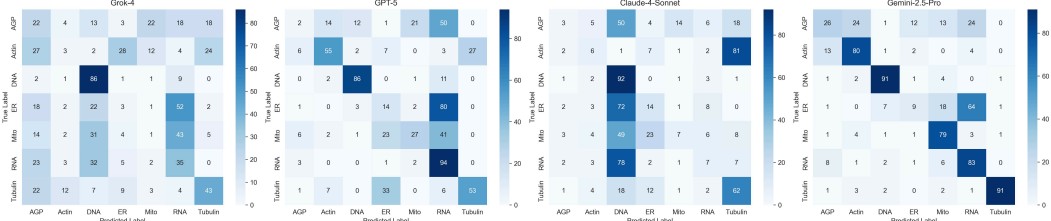

Figure 10: Confusion matrix on image channel task.

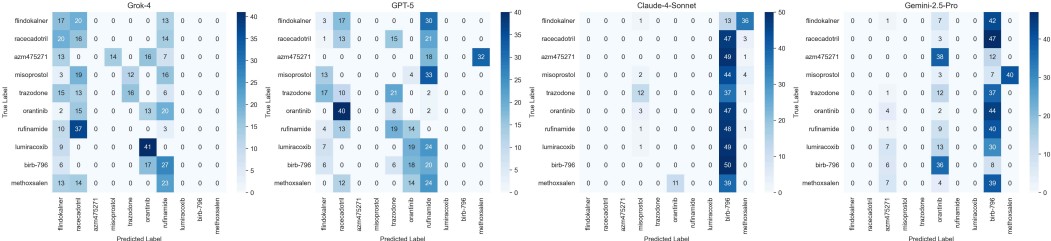

Figure 11: Confusion matrix on perturbation compound task.

## M.4 FEW-SHOT RESULTS

Table 12: Few-shot Model performance on classification tasks

| Model | Cell line | Channel | Perturbation Compound | | | | | | | | | | |
|---|---|---|---|---|---|---|---|---|---|---|---|---|---|
| | | | Flindokalner | Racecadotril | AZM-475271 | Misoprostol | Trazodone | Orantinib | Rufinamide | Lumiracoxib | BIRB-796 | Methoxsalen | Macro-avg |
| Grok-4 | 0.515 (+0.067) | 0.260 (+0.032) | 0.224 | 0.184 | 0.0 | 0.0 | 0.390 | 0.176 | 0.034 | 0.000 | 0.0 | 0.000 | 0.101 |
| GPT-5 | 0.440 (+0.063) | 0.510 (+0.071) | 0.0 | 0.0 | 0.066 | 0.0 | 0.115 | 0.000 | 0.079 | 0.000 | 0.088 | 0.000 | 0.035 |
| Claude-4-Sonnet | 0.520 (+0.070) | 0.225 (+0.027) | 0.000 | 0.000 | 0.000 | 0.055 | 0.000 | 0.000 | 0.000 | 0.000 | 0.210 | 0.000 | 0.026 |
| Gemini-2.5-Pro | 0.600 (+0.074) | 0.730 (+0.102) | 0.0 | 0.000 | 0.000 | 0.000 | 0.000 | 0.0 | 0.000 | 0.160 | 0.074 | 0.000 | 0.023 |

## N CP-AGENT PROMPTS

The prompts guide the CP-Agent through a multi-step reasoning process to interpret morphological effects of perturbations in Cell Painting data. Figure 12 introduces two tasks: (1) a background curation step, where the agent synthesizes prior biological knowledge about a compound's mechanism of action (MoA) and predicts which CellProfiler feature classes are likely to be affected in a specific imaging channel, and (2) a feature ranking task, where individual features are prioritized based on their relevance to the predicted morphological response. Figure 13 guides the CP-Agent to evaluate whether observed morphological changes under a perturbation are consistent with the proposed mechanism of action (MoA). Using prior biological knowledge and quantitative feature summaries, the agent assesses each feature's directional change, links it to the expected mechanism, and assigns confidence scores. The agent then provides an overall judgment of mechanism plausibility, highlighting supporting or conflicting evidence. All prompts enforce structured JSON outputs to ensure compatibility with automated downstream analysis and promote reproducibility.

**Task: Report Generation**

**Background Information Curation**

You are a biomedical research assistant with expertise in chemical biology and phenotypic profiling.

**Task:**

Given a chemical perturbation and a specific Cell Painting imaging channel, provide mechanistic insight and hypothesize expected morphology changes specific to that cellular component.

**Input:**

- Perturbation condition (compound name, Cell Painting imaging channel, dose, time, etc.): {{ perturbation_condition }}

**Your responsibilities:**

- Mechanism summary: a 1–2 sentence description of the compound's mechanism of action (MoA).

- Channel-relevant hypotheses: a list of morphology-level effects expected for the given cellular component (e.g., "Tubulin depolymerization lower microtubule texture", "ER fragmentation granularity increase") with reasoning focused on the selected channel.

- Likely impacted feature types: a list of CellProfiler feature types likely to change in this channel (e.g., "Texture_Tubulin", "Granularity_Tubulin", "AreaShape").

**Output format:**

Return only JSON in the following format:

```
{
  "mechanism_summary": "<short description>",
  "morphology_hypotheses": ["<hypothesis 1>","<hypothesis 2>"],
  "likely_feature_types": ["AreaShape", "Texture_Tubulin", "Granularity_Tubulin", "Neighbors", "Location", "Number"]
}
```

**Notes:**

- Focus all hypotheses and feature types on the specific imaging channel.

- If the compound is known to affect this structure, be specific. If the effect is indirect or uncertain, say so.

- Be biologically grounded but concise.

---

**Task: Report Generation**

**Feature Ranking Template for VQA**

**feature_prediction_sys**

You are a biomedical specialist in cell morphological features.

Follow instructions exactly. Remain grounded in the provided context.

Return JSON only, with no extra text.

**feature_prediction_user**

**Task:**

Given a Cell Painting perturbation experiment, predict which morphological features are most likely to be affected.

**Inputs:**

- Background biological knowledge (from prior curation step):

{{ background_curation_json }}

- Perturbation condition: {{ perturbation_condition }}

- Candidate CellProfiler feature names (i.e., the only allowed feature namespace): {{ feature_names_json }}

**Your responsibilities:**

- Select from the provided feature name list only.

- Predict which features are most likely to show morphological change under this perturbation.

- Ground your reasoning in both the known biological mechanism and expected morphological effects.

- If possible, relate features to biological structures (e.g., nuclear shape, texture, granularity, area, neighbor count, etc.).

- Be conservative: do not overclaim. If uncertain, assign lower confidence.

**Output format:**

Return only JSON in the following format:

```
{
  "features_ranked": [
    {
      "name": "<feature_name from provided list>",
      "rationale": "<1-2 sentence rationale grounded in A vs B visual differences and context>",
      "confidence": <float between 0 and 1>
    }
  ],
  "summary": "<brief one-paragraph summary of key morphology differences observed in B relative to A>"
}
```

Figure 12: Prompt templates for background curation and feature ranking.

**Task: Report Generation**

**Feature Mechanism Consistency Template for VQA**

**feature_mechanism_consistency_sys:**
You are a biomedical imaging and phenomics analyst specializing in Cell Painting assays.
Your primary evidence is quantitative feature summaries derived from curated CellProfiler outputs. Visual evidence may be referenced only if explicitly provided as inputs.
Please output ONLY a valid JSON object without any explanation, markdown formatting, or extra text.

**feature_mechanism_consistency_user:**

**Context:**
- The perturbation condition provided below may be incorrect, noisy, or adversarial (e.g., a fake drug name).
- Prior biological knowledge about the perturbation is a SOFT prior only. You must verify it against quantitative evidence. If there is a mismatch, you must say so.
- Evidence priority: (1) quantitative statistics, (2) visual evidence, (3) prior mechanism. Be conservative: if mechanism name/alias is not recognized or could be confused (e.g., fake or uncommon), set plausibility low and avoid strong claims.

**Inputs:**
- Control image: A (DMSO)
- Perturbation image: B (perturbed)
- Perturbation condition: {{ perturbation_condition }}
- Background mechanism and morphology expectations (from prior knowledge): {{ background_curation_json }}
- Quantitative summary of features (based on population statistics): {{ summary_of_features_json }}

**Your responsibilities:**
- You are given two grayscale microscopy images: A = control (DMSO), B = perturbed.
- Compare expected from perturbation condition vs observed data from quantitative summary of features and visual evidence.
- For each feature in the summary, evaluate whether its change supports, contradicts, or is insufficient relative to the mechanism. Provide a confidence score (0.0–1.0) for your judgment.
- You must evaluate **every feature** provided, even if not statistically significant. If evidence is weak (e.g., high q-value, CI crosses 0, or small effect size), state that explicitly and assign low confidence.
- You may order features by significance, but do not skip any.
- Summarize the dominant morphology change and explain how the quantitative trends support or contradict expectations. If contradict, explain why (e.g., incorrect mechanism, off-target, low dose/time, or similar phenotype to another class).
- Provide a concise overall assessment of whether the observed phenotype aligns with the proposed mechanism, highlighting key supporting or conflicting features based on quantitative summary of features.

**Output format:**
Only return JSON. Do not include any non-JSON text, comments, or markdown.
```
{
  "features_ranked": [
    {
      "name": "<feature_name from provided list>",
      "direction": "<increase|decrease|ambiguous>",
      "observed_evidence": "<1-2 sentences citing quantitative stats (delta/CI/Cliff's delta/q) and, if clear, visual differences. No claims beyond provided evidence.>",
      "mechanism_link": "<why this feature's change would support/contradict the proposed mechanism; if unclear, state ambiguity.>",
      "supports_proposed_mechanism": "<supports|contradicts|insufficient>",
      "support_confidence": <float between 0 and 1>
    }
  ],
  "mechanism_consistency": "<supports|contradicts|insufficient>",
  "plausibility_score": "<float between 0 and 1 estimate of how credible the perturbation_condition is, based on the selected feature name and their supporting scores for the proposed mechanism. Lower if conflicting.>",
  "morphology_summary": "<One concise paragraph summarizing the dominant morphological changes observed in Perturbed (B) vs Control (A), based primarily on quantitative features. If visual evidence is available and clearly supports the trends (e.g., more fragmented mitochondria, loss of structure), you may briefly mention it as secondary support.>",
  "overall_mechanism_assessment": """
  Write 5 parts in order:
  - Prelude: briefly state
    a) what the image/condition is hypothesized to show (the proposed MoA or phenotype guess), and
    b) what this assessment will do next (evaluate consistency using quantitative features and mechanistic reasoning).
    Example: "The specimen is hypothesized to reflect [proposed MoA/phenotype]. Below, I assess whether the observed morphology aligns with this hypothesis using quantitative feature trends and a mechanistic linkage."
  - Mechanism verdict.
  - Evidence summary.
  - Mechanistic linkage sentence: explicitly link MoA cellular process expected morphologies observed features.
    Use one of:
    "Given the proposed MoA [MoA], it is known to alter [process A] via [action], which should cause [outcome B]; this would present as [features] (observed: [metrics])."
    "If [MoA] perturbs [process A], it could plausibly produce [outcome B], aligning with [features] (observed: [metrics]); however, this is tentative due to [limitation]."
  - Caveats and alternatives if have (1–2 sentences): note inconsistencies/limitations and suggest next checks (e.g., dose–response, time-course, orthogonal markers, replicate expansion).
    Do not claim causality beyond the provided evidence; keep wording falsifiable and bounded by data.
  """
}
```

Figure 13: Prompt template for evaluating mechanism-feature consistency in Cell Painting data.

# O  ADDITIONAL CASE STUDIES

## O.1  ADDITIONAL CASE 1: TAXOL IN MCF7

**Cell Painting Drug Experiment Brief — *Taxol in MCF7 (Tubulin, 20X; 24 μM, 24 h)***

**1. Most Matched Experimental Context**
• Cell line: MCF7 (human breast cancer)
• Culture medium: RPMI 1640, 10% FBS, 1% GlutaMAX, 900 μg/mL G418
• Imaging: Tubulin channel, 20X objective
• Compound: Taxol (24 μM, 24h)
• Mechanism of action: Microtubule stabilization

**2. Visual Overview**: Control vs Perturbation Images

Control        Perturbed

**3. Anticipated Feature Changes Based on Mechanism**
• Expected effects of taxol (microtubule stabilizer):
    - Relative to control, 24 μM taxol for 24 hours in MCF7 is expected to produce stabilized, thick microtubule bundles with perinuclear and cortical enrichment and mitotic spindle/aster formations.
These changes reduce fine-scale heterogeneity while increasing large-scale order, leading to:
    - ↓ Entropy / contrast at large offsets
    - ↑ Angular second moment, correlation, homogeneity
    - Scale-dependent granularity shifts (↓ fine scale, ↑ mid-scale)

**4. Key Feature Evidence from Data**

*AreaShape_Solidity*
Direction: decrease
Observed evidence: Median decreased by −0.082 (CI [−0.114, −0.055]); large effect (Cliff's δ = 0.82); highly significant (q = 6.27e−06).
Mechanism link: Microtubule stabilization can induce mitotic arrest and cell shape irregularities, reducing solidity via protrusions/aster-like structures.
Supports proposed mechanism: ✅ Yes (0.9 confidence)

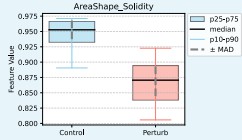

*Texture_Variance_Tubulin_5_01_256*
Direction: increase
Observed evidence: Median increased by +1267 (CI [615, 1599]); strong effect (Cliff's δ = −0.627); significant (q = 0.000413).
Mechanism link: Stabilized, thick bundles and spindles raise intensity variance within cells.
Supports proposed mechanism: ✅ Yes (0.85 confidence)

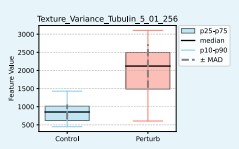

*Texture_SumVariance_Tubulin_7_01_256*
Direction: increase
Observed evidence: Median increased by +4228 (CI [1972, 5274]); strong effect (Cliff's δ = −0.602); significant (q = 0.000610).
Mechanism link: Global variability across neighborhoods is expected to rise with bundled microtubules and spindle structures.
Supports proposed mechanism: ✅ Yes (0.84 confidence)

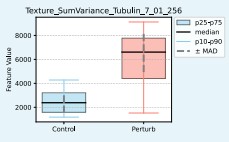

*Texture_Contrast_Tubulin_7_01_256*
Direction: increase
Observed evidence: Median increased by +932 (CI [384, 1452]); strong effect (Cliff's δ = −0.638); significant (q = 0.000412).
Mechanism link: Thick, bright bundles next to darker cytoplasm increase local contrast, consistent with microtubule stabilization.
Supports proposed mechanism: ✅ supports (0.85 confidence)

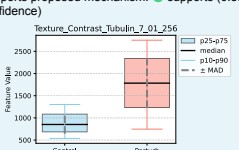

*Texture_DifferenceEntropy_Tubulin_7_02_256*
Direction: increase
Observed evidence: Median increased by +0.689 (CI [0.186, 0.809]); large effect (Cliff's δ = −0.672); significant (q = 0.000252).
Mechanism link: Bundling and spindle poles introduce pronounced intensity differences, elevating difference-entropy; compatible with stabilized microtubules.
Supports proposed mechanism: ✅ supports (0.82 confidence)

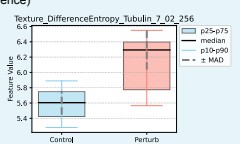

*Texture_InfoMeas1_Tubulin_7_03_256*
Direction: decrease
Observed evidence: Median decreased by −0.049 (CI [−0.076, −0.014]); moderate effect (Cliff's δ = 0.468); significant (q = 0.00981).
Mechanism link: Lower informational measure of correlation can reflect stronger structured heterogeneity from bundled fibers and asters.
Supports proposed mechanism: ✅ supports (0.7 confidence)

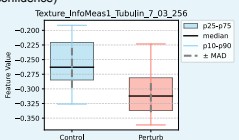

**5. Mechanism Assessment and Conclusion**

*Key Evidence (Features with Large Effect Sizes):*

*1.Texture_Variance_Tubulin_5_01_256*
*↑ Median by +1267*
*q = 4.13e-04*
*Indicates increased intensity heterogeneity due to bundled microtubules*

*2.Texture_SumVariance_Tubulin_7_01_256*
*↑ Median by +4228*
*q = 6.10e-04*
*Suggests increased global variability in tubulin texture*

*３.Texture_Contrast_Tubulin_7_01_256*
*↑ Median by +932*
*q = 4.12e-04*
*Reflects sharper contrast from bright bundles vs. background*

*４.Texture_DifferenceEntropy_Tubulin_7_02_256*
*↑ Median by +0.689*

*Morphology_summary*
Perturbed cells are far fewer (16 vs 66) and show strong increases in Tubulin texture variance, sum-variance, contrast, and difference-entropy, alongside a marked decrease in cell solidity. These quantitative trends indicate more pronounced, bright microtubule structures and irregular cell outlines, consistent with bundled/stabilized microtubules and mitotic arrest phenotypes. Other texture and shape metrics show weak or non-significant changes.

*Overall_mechanism_linked_summary*
The specimen is hypothesized to reflect microtubule stabilization by taxol leading to bundled tubulin and mitotic arrest. Below, I assess whether the observed morphology aligns with this hypothesis using quantitative feature trends and mechanistic linkage. The quantitative phenotype supports microtubule stabilization.
Given the proposed MoA (mechanism of action) of microtubule stabilizers, it is known to alter microtubule dynamics via suppression of depolymerization, which should cause bundled, bright fibers and mitotic arrest; this would present as higher tubulin texture variance/contrast and reduced cell solidity (observed: large positive shifts in variance/-contrast/difference-entropy and a strong solidity decrease).
Caveats: Sample size in perturbed is small (n = 16), and several features are non-significant; off-target effects or imaging differences cannot be excluded. Follow-up with dose–response/time-course, mitotic markers (e.g., pH3), and replicate expansion is recommended to strengthen the attribution.

## O.2 ADDITIONAL CASE 2: VINCRISTINE IN MCF7

**Cell Painting Drug Experiment Brief — *Vincristine in MCF7 (Tubulin, 20X; 24 µM, 24 h)***

**1. Most Matched Experimental Context**
• Cell line: MCF7 (human breast cancer)
• Culture medium: RPMI 1640, 10% FBS, 1% GlutaMAX, 900 µg/mL G418
• Imaging: Tubulin channel, 20X objective
• Compound: Vincristine (24 µM, 24h)
• Mechanism of action: Microtubule destabilizers

**2. Visual Overview**: Control vs Perturbation Images

Control 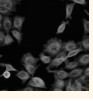 Perturbed 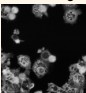

**3. Anticipated Feature Changes Based on Mechanism**
• Expected effects of vincristine (microtubule destabilizer):
  Vincristine disrupts microtubules, leading to cytoskeletal collapse, mitotic arrest, and morphological changes in treated cells.
• Relative to control, vincristine treatment is expected to cause:
  • Loss of microtubule structure and texture
  • Cytoskeletal collapse resulting in more rounded, smaller cells
  • Accumulation of cells in mitosis (increased cell number)
• These changes are expected to manifest in Cell Painting features as:
  - ↓ Tubulin texture contrast and entropy (e.g., Texture_Contrast_Tubulin, Texture_Entropy_Tubulin)
  - ↑ Angular second moment (e.g., Texture_AngularSecondMoment_Tubulin) due to more uniform staining
  - ↓ Eccentricity, ↑ FormFactor (more circular cells)
  - ↓ AreaShape_Area (due to mitotic rounding)
  - ↑ Number_Object_Number (mitotic arrest increases cell count)
  - Scale-dependent granularity shifts in tubulin (e.g., Granularity_2_Tubulin)

**4. Key Feature Evidence from Data**

***Texture_Contrast_3_01_256***
Direction: increase
Observed evidence: Median increased by +459.3 (q = 2.15e-08, Cliff's delta = -0.83)
Mechanism link: Microtubule destabilization leads to the collapse of the fine filamentous network into bright, dense aggregates (paracrystals) and dark, empty cytoplasmic regions. This reorganization drastically increases the contrast between adjacent pixels.
Supports proposed mechanism: ✓ Yes (1.0 confidence)

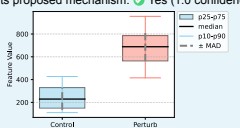

***Number_Object_Number***
Direction: decrease
Observed evidence: Median decreased by -8.5 (q = 0.0156, Cliff's delta = 0.38)
Mechanism link: Vincristine is a cytotoxic agent that induces mitotic arrest, often leading to apoptosis and subsequent cell death. A reduction in cell count after 24 hours is a direct and expected consequence.
Supports proposed mechanism: ✓ Yes (0.9 confidence)

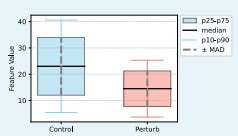

***Granularity_2***
Direction: increase
Observed evidence: Median increased by +0.276 (q = 0.0156, Cliff's delta = -0.376)
Mechanism link: Feature quantifies emergence of coarser textures. Depolymerization of fine microtubules and reassembly into larger aggregates is consistent with increased granularity.
Supports proposed mechanism: ✓ Yes (0.9 confidence)

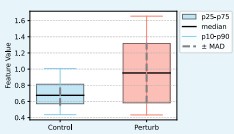

***Texture_AngularSecondMoment_3_01_256***
Direction: decrease
Observed evidence: Median decreased by -0.00024 (q = 0.0156, Cliff's delta = 0.366)
Mechanism link: Reflects a loss of homogeneous filamentous texture in control cells, replaced by heterogeneous pattern of bright aggregates and dark voids.
Supports proposed mechanism: ✓ Yes (0.9 confidence)

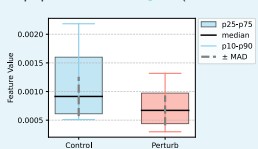

***AreaShape_FormFactor***
Direction: increase
Observed evidence: Median increased by +0.689 (CI [0.186, 0.809]); large effect (Cliff's δ = −0.672); significant (q = 0.000252).
Mechanism link: Bundling and spindle poles introduce pronounced intensity differences, elevating difference-entropy; compatible with stabilized microtubules.
Supports proposed mechanism: ✓ supports (0.82 confidence)

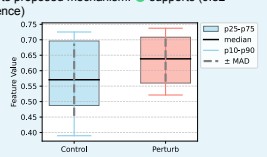

***AreaShape_Eccentricity***
Direction: decrease
Observed evidence: Median decreased by -0.033 (q = 0.184) — not statistically significant
Mechanism link: Complementary to FormFactor; also indicates cell rounding due to cytoskeletal collapse.
Supports proposed mechanism: ✓ Yes (0.5 confidence)

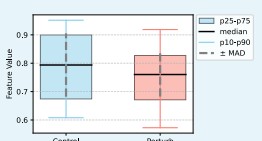

**5. Mechanism Assessment and Conclusion**

*Key Evidence (Features with Large Effect Sizes):*
*1.Texture_Contrast_Tubulin_3_01_256*
*↑ Median by +459.3*
*q = 2.15e-08*
*Indicates increased local pixel intensity variation due to tubulin network collapse and aggregation*
*2.Granularity_2_Tubulin*
*↑ Median by +0.276*
*q = 0.0156*
*Quantifies emergence of coarse texture patterns caused by tubulin depolymerization and aggregate formation*
*3.Texture_AngularSecondMoment_Tubulin_3_01_256*
*↓ Median by −0.00024*
*q = 0.0156*
*Reflects loss of homogeneous filamentous texture, consistent with microtubule disruption*
*4.Number_Object_Number*
*↓ Median by −8.5*
*q = 0.0156*
*Indicates reduced cell count, consistent with mitotic arrest and vincristine-induced cytotoxicity*

*Morphology_summary*
The dominant morphological change is a profound disruption of the tubulin cytoskeleton. Quantitatively, this is captured by a highly significant increase in Texture_Contrast_Tubulin and Granularity_2_Tubulin, reflecting the collapse of the filamentous network into coarse, bright aggregates. This cytoskeletal failure is consistent with the observed (though not statistically significant) trends toward a more rounded cell shape, indicated by an increase in AreaShape_FormFactor. Furthermore, the treatment induced significant cytotoxicity, evidenced by a marked decrease in the Number_Object_Number.

*Overall_mechanism_linked_summary*
The specimen is hypothesized to reflect microtubule destabilization induced by vincristine. Below, I assess whether the observed morphology aligns with this hypothesis using quantitative feature trends and a mechanistic linkage.
The observed phenotype strongly supports the proposed mechanism of action.
The most significant changes are in tubulin texture, with a massive increase in Texture_Contrast_Tubulin (delta_median: +459.3) and a decrease in homogeneity (Texture_AngularSecondMoment_Tubulin), indicating tubulin aggregation. This is accompanied by a significant decrease in cell number (Number_Object_Number), suggesting cytotoxicity.
Given the proposed MoA of microtubule destabilization, vincristine is known to alter tubulin polymerization by preventing microtubule formation, which should cause cytoskeletal collapse and mitotic arrest; this would present as a loss of filamentous structures, formation of tubulin aggregates, and cell rounding, aligning with the observed increases in tubulin contrast and granularity (observed: q < 0.02 for texture features).
While the textural and cytotoxicity evidence is definitive, the expected changes in cell shape did not reach statistical significance, which could be due to insufficient statistical power or population heterogeneity.

## O.3 ADDITIONAL CASE 3: SORBINIL IN A549

**Cell Painting Drug Experiment Brief — *Sorbinil in A549 (RNA, 20X; 48µM, 48h)***

**1. Most Matched Experimental Context**
• Cell line: A549
• Culture medium: 2% FBS in DMEM medium
• Imaging: RNA channel, 20X objective
• Compound: Sorbinil (48 µM, 48h)
• Mechanism of action: Aldose Reductase Inhibitor

**2. Visual Overview**: Control vs Perturbation Images

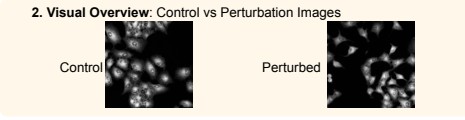

Control          Perturbed

**3. Anticipated Feature Changes Based on Mechanism**

• Expected effects of sorbinil (aldose reductase inhibitor under oxidative/osmotic stress):
Sorbinil is expected to perturb RNA organization through redox/osmotic stress, leading to nucleolar compaction, cytoplasmic RNA puncta formation, and perinuclear RNA redistribution.
• Relative to control, sorbinil treatment is expected to cause:
    • Increased RNA texture heterogeneity
    • Formation of stress granule–like puncta
    • Redistribution of RNA to nucleoli and perinuclear zones
• These changes are expected to manifest in Cell Painting features as:
    - ↑ RNA texture entropy (e.g., Texture_Entropy_RNA_5_02_256, Texture_Entropy_RNA_7_02_256)
    - ↑ RNA texture contrast (e.g., Texture_Contrast_RNA_5_02_256, Texture_Contrast_RNA_7_02_256)
    - ↑ Local intensity variability (e.g., Texture_DifferenceEntropy_RNA_5_02_256, Texture_DifferenceEntropy_RNA_5_02_256)
    - ↑ Texture variance and sum variance (e.g., Texture_Variance_RNA_5_02_256, Texture_SumVariance_RNA_5_02_256)
    - ↑ Mid- to coarse-scale granularity (e.g., Granularity_2_RNA, Granularity_3_RNA, Granularity_4_RNA)

**4. Key Feature Evidence from Data**

*Texture_Entropy_5_02_256*
Direction: decrease
Observed evidence: Median decreased by −3.782 (CI [−5.094, −2.506]); large effect (Cliff's δ = 0.782); q = 9.55e−08
Mechanism link: Lower entropy indicates more uniform/ordered RNA signal, consistent with nucleolar compaction or consolidation under redox/osmotic stress from aldose reductase inhibition.
Supports proposed mechanism: ✓ Yes (0.92 confidence)

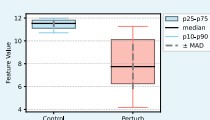

*Texture_Entropy_7_02_256*
Direction: decrease
Observed evidence: Median decreased by −3.655 (CI [−5.041, −2.486]); large effect (Cliff's δ = 0.781); q = 9.55e−08
Mechanism link: Reduced large-scale entropy aligns with more homogeneous RNA distribution and potential nucleolar consolidation expected with nucleolar stress.
Supports proposed mechanism: ✓ Yes (0.91 confidence)

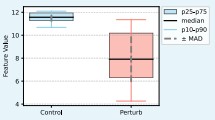

*Texture_InverseDifferenceMoment_5_02_256*
Direction: increase
Observed evidence: Median increased by +0.293 (CI [0.176, 0.430]); strong effect (Cliff's δ = −0.708); q = 1.23e−06
Mechanism link: Higher homogeneity (IDM) suggests smoother/less varied RNA texture, compatible with condensed nucleolar signal and reduced diffuse RNA.
Supports proposed mechanism: ✓ Yes (0.88 confidence)

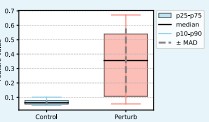

*Texture_DifferenceVariance_5_02_256*
Direction: increase
Observed evidence: Median increased by +5.27e−04 (CI [1.50e−04, 1.12e−03]); moderate effect (Cliff's δ = −0.473); q = 0.0021
Mechanism link: Increased difference variance can reflect sharper boundaries or localized foci; this is consistent with formation of distinct nucleolar/cytoplasmic RNA foci under stress.
Supports proposed mechanism: ✓ Yes (0.70 confidence)

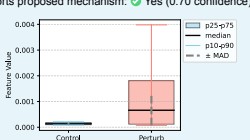

*Texture_DifferenceEntropy_5_02_256*
Direction: decrease
Observed evidence: Median decreased by −1.046 (CI [−1.781, −0.290]); moderate effect (Cliff's δ = 0.435); q = 0.00464
Mechanism link: Lower difference entropy indicates more predictable gray-level differences, consistent with RNA signal consolidation/ordering during nucleolar stress.
Supports proposed mechanism: ✓ Yes (0.68 confidence)

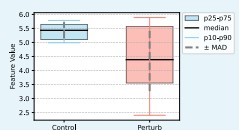

*Granularity_3_RNA*
Direction: decrease
Observed evidence: Median decreased by −0.176 (CI [−0.406, 0.051]); small–moderate effect (Cliff's δ = 0.317); q = 0.0523
Mechanism link: A decrease in mid-scale granularity could reflect smoother RNA texture or consolidation into fewer/larger regions; directionally compatible but statistically marginal.
Supports proposed mechanism: ⚠ Insufficient (0.45 confidence)

*Granularity_2_RNA*
Direction: decrease
Observed evidence: Median decreased by −0.140 (CI [−0.325, 0.064]); small effect (Cliff's δ = 0.288); q = 0.0774
Mechanism link: Reduced small-scale granularity could indicate loss of fine RNA puncta, which only partially matches expectations.
Supports proposed mechanism: ⚠ Insufficient (0.38 confidence)

**5. Mechanism Assessment and Conclusion**

*Key Evidence (Features with Large Effect Sizes):*
*1. Texture_Entropy_RNA_5_02_256*
↓ Median by −3.782
q = 9.55e−08
Indicates increased RNA ordering and reduced intensity heterogeneity
*2. Texture_Entropy_RNA_7_02_256*
↓ Median by −3.655
q = 9.55e−08
Reflects large-scale RNA homogenization, consistent with nucleolar consolidation
*3. Texture_InverseDifferenceMoment_RNA_5_02_256*
↑ Median by +0.293
q = 1.23e−06
Suggests increased homogeneity of RNA texture
*4. Texture_DifferenceVariance_RNA_5_02_256*
↑ Median by +5.27e−04
q = 0.0021
Indicates sharper local gray-level differences, compatible with nucleolar foci
*5. Texture_DifferenceEntropy_RNA_5_02_256*
↓ Median by −1.046
q = 0.00464

*Morphology_summary*
The perturbed condition exhibits markedly reduced RNA texture entropy and increased homogeneity (higher inverse difference moment), indicating a more ordered RNA distribution.
Difference-variance and difference-entropy metrics suggest sharper but more predictable local gray-level relationships, compatible with consolidation of RNA signal into structured subnuclear regions and a reduction of diffuse nucleoplasmic RNA. Granularity measures trend downward at small-to-mid scales but are statistically marginal, implying limited evidence for increased cytoplasmic puncta. Overall, the dominant change is smoother, more ordered RNA texture consistent with nucleolar compaction or consolidation.

*Overall_mechanism_linked_summary*
The specimen is hypothesized to reflect aldose reductase inhibition by sorbinil causing redox/osmotic stress with nucleolar/RNA processing alterations. Below, I assess whether the observed morphology aligns with this hypothesis using quantitative feature trends and mechanistic linkage.
Mechanism verdict: The observed RNA texture changes are broadly consistent with the proposed mechanism.
Evidence summary: Texture_Entropy_RNA (5 and 7 px) decreases strongly (Δ ≈ −3.7, q ≈ 1e−7), and Texture_InverseDifferenceMoment increases (Δ = +0.293, q ≈ 1e−6), indicating higher homogeneity. DifferenceEntropy decreases and DifferenceVariance increases (q ≤ 0.005), supporting more ordered yet sharper local structure. Granularity metrics show small, non-significant decreases; contrast changes are in the expected direction but not significant.
If the proposed MoA aldose reductase inhibitor perturbs redox balance and osmotic stress, it could plausibly produce nucleolar stress leading to compaction/consolidation of rRNA-rich regions and reduced diffuse transcription; this would present as lower entropy and higher homogeneity in RNA texture (observed: strong decreases in entropy and increases in IDM, with supporting difference-statistics).
Caveats and alternatives: Many granularity and contrast features are not significant, and direct evidence for cytoplasmic stress-granule-like puncta is lacking. Replicate expansion, dose–response, and inclusion of nucleolar/translation markers would help disambiguate nucleolar consolidation from general transcriptional downshift or imaging/segmentation artifacts.

## O.4 ADDITIONAL CASE 4: BGT226 IN HUVEC

**Cell Painting Drug Experiment Brief — *BGT226 in HUVEC (Mito, 10X; 20µM, 20h)***

**1. Most Matched Experimental Context**
• Cell line: HUVEC
• Culture medium: EGM2 medium
• Imaging: Mito channel, 10X objective
• Compound: BGT226 (20 µM, 20h)
• Mechanism of action: PI2-kinase Class I Inhibitor

**2. Visual Overview**: Control vs Perturbation Images

Control    Perturbed

**3. Anticipated Feature Changes Based on Mechanism**

• Expected effects of BGT226 (PI3K/mTOR inhibitor):
BGT226 treatment is expected to impair mitochondrial integrity through inhibition of PI3K/mTOR signaling, leading to mitochondrial fragmentation, reduced mitochondrial mass, and cristae disruption.
• Relative to control, BGT226 treatment is expected to cause:
  • Loss of mitochondrial area (due to reduced mass)
  • Mitochondrial fragmentation and network disruption
  • Changes in mitochondrial shape (less elongated, more circular)
  • Altered cristae structure and internal texture
  • Modified spatial organization of mitochondrial objects
• These changes are expected to manifest in Cell Painting features as:
  - ↓ Mitochondrial area (e.g., AreaShape_Area)
  - ↓ Mitochondrial compactness and form factor (e.g., AreaShape_Compactness, AreaShape_FormFactor)
  - ↓ Eccentricity, ↓ MajorAxisLength (e.g., AreaShape_Eccentricity, AreaShape_MajorAxisLength)
  - ↑ Granularity at medium scales (e.g., Granularity_3_Mito)
  - ↑ Texture contrast and entropy (e.g., Texture_Contrast_Mito_3_01_256, Texture_Entropy_Mito_3_01_256)
  - ↓ Solidity (e.g., AreaShape_Solidity)
  - Changes in mitochondrial neighborhood structure (e.g., Neighbors_NumberOfNeighbors_10)

**4. Key Feature Evidence from Data**

*Texture_Contrast_3_01_256*
Direction: increase
Observed evidence: Strong increase from 151.2 to 247.6 (Δ = +96.4, CI: [58.9, 142.1], Cliff's δ = −0.57, q = 6.8e−08)
Mechanism link: PI3K/mTOR inhibition could disrupt mitochondrial organization and cristae structure, leading to increased heterogeneity and contrast in mitochondrial staining patterns.
Supports proposed mechanism: ✅ Yes (0.8 confidence)

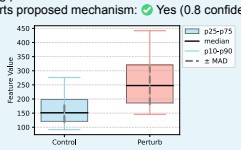

*AreaShape_MajorAxisLength*
Direction: increase
Observed evidence: Significant increase from 69.2 to 92.0 (Δ = +22.7, CI: [10.4, 29.8], Cliff's δ = −0.56, q = 6.8e−08)
Mechanism link: PI3K/mTOR inhibition can induce cell stress and alter cytoskeletal organization, potentially causing cell elongation as part of stress response or altered adhesion.
Supports proposed mechanism: ✅ Yes (0.7 confidence)

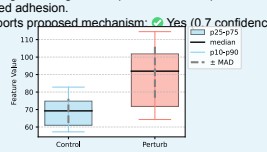

*Neighbors_NumberOfNeighbors_10*
Direction: decrease
Observed evidence: Decrease from 6.0 to 4.0 neighbors (Δ = −2.0, CI: [−2.0, −1.0], Cliff's δ = 0.55, q = 6.8e−08)
Mechanism link: PI3K/mTOR inhibition can reduce cell proliferation and survival, leading to lower cell density and fewer neighboring cells.
Supports proposed mechanism: ✅ Yes (0.8 confidence)

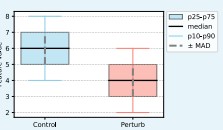

*AreaShape_Area*
Direction: increase
Observed evidence: Significant increase from 2179.5 to 3075.0 (Δ = +895.6, CI: [510.5, 1335.0], Cliff's δ = −0.52, q = 4.9e−07)
Mechanism link: PI3K/mTOR inhibition can cause cell cycle arrest and stress-induced cell enlargement, consistent with metabolic disruption and altered growth signaling.
Supports proposed mechanism: ✅ Yes (0.7 confidence)

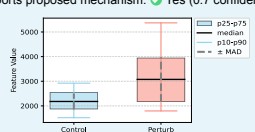

*AreaShape_Compactness*
Direction: increase
Observed evidence: Increase from 1.51 to 1.83 (Δ = +0.33, CI: [0.18, 0.43], Cliff's δ = −0.50, q = 8.9e−07)
Mechanism link: PI3K/mTOR inhibition can disrupt cytoskeletal organization and cell adhesion, leading to less compact, more irregular cell shapes.
Supports proposed mechanism: ✅ Yes (0.7 confidence)

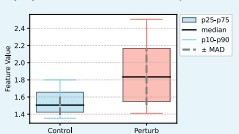

*AreaShape_FormFactor*
Direction: decrease
Observed evidence: Decrease from 0.66 to 0.55 (Δ = −0.12, CI: [−0.15, −0.07], Cliff's δ = 0.50, q = 8.9e−07)
Mechanism link: Consistent with compactness changes, PI3K/mTOR inhibition disrupts normal cell morphology, making cells less circular and more irregular.
Supports proposed mechanism: ✅ Yes (0.7 confidence)

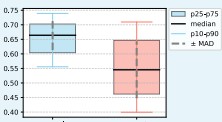

*Texture_Entropy_3_01_256*
Direction: increase
Observed evidence: Increase from 7.10 to 7.74 (Δ = +0.63, CI: [0.36, 0.99], Cliff's δ = −0.49, q = 9.7e−07)
Mechanism link: PI3K/mTOR inhibition can disrupt mitochondrial organization and function, leading to more heterogeneous and disordered mitochondrial structures.
Supports proposed mechanism: ✅ Yes (0.8 confidence)

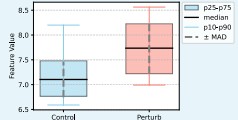

*AreaShape_Solidity*
Direction: decrease
Observed evidence: Decrease from 0.92 to 0.89 (Δ = −0.03, CI: [−0.08, −0.01], Cliff's δ = 0.43, q = 2.0e−05)
Mechanism link: PI3K/mTOR inhibition can affect cytoskeletal organization and cell adhesion, leading to less solid cell shapes with more protrusions or irregularities.
Supports proposed mechanism: ✅ Yes (0.6 confidence)

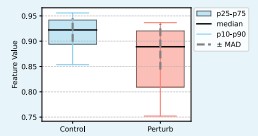

*Granularity_3_Mito*
Direction: decrease
Observed evidence: Weak decrease from 2.54 to 2.31 (Δ = −0.24, CI: [−0.94, 0.19], Cliff's δ = 0.16, q = 0.11)
Mechanism link: PI3K/mTOR inhibition might reduce mitochondrial granularity through altered mitochondrial biogenesis, but evidence is insufficient.
Supports proposed mechanism: ⚠️ Insufficient (0.2 confidence)

*AreaShape_Eccentricity*
Direction: increase
Observed evidence: Weak increase from 0.78 to 0.80 (Δ = +0.02, CI: [−0.02, 0.08], Cliff's δ = −0.16, q = 0.11)
Mechanism link: Slight increase in eccentricity could relate to cell elongation, but the effect is too weak to draw conclusions.
Supports proposed mechanism: ⚠️ Insufficient (0.2 confidence)

**Cell Painting Drug Experiment Brief — *BGT226 in HUVEC (Mito, 10X; 20µM, 20h)***

**5. Mechanism Assessment and Conclusion**

*Key Evidence (Features with Large Effect Sizes):*
*1.Texture_Contrast_Mito_3_01_256*
↑ *Median by +96.4*
*q = 6.8e−08*
*Indicates increased mitochondrial texture contrast, consistent with cristae disruption and mitochondrial heterogeneity*
*2.Texture_Entropy_Mito_3_01_256*
↑ *Median by +0.63*
*q = 9.7e−07*
*Suggests increased mitochondrial texture disorder and fragmentation*
*3.AreaShape_Area*
↑ *Median by +895.5*
*q = 4.9e−07*
*Reflects stress-induced cell enlargement*
*4. AreaShape_MajorAxisLength*
↑ *Median by +22.7*
*q = 6.8e−08*
*Consistent with cell elongation under stress or cytoskeletal remodeling*
*5.Neighbors_NumberOfNeighbors_10*
↓ *Median by −2.0*
*q = 6.8e−08*
*Indicates reduced cell density, possibly due to proliferation arrest or cytotoxicity*

*Morphology_summary*
The perturbed cells show a consistent pattern of stress-related morphological changes including significantly increased cell size (area +41%), elongation (major axis +33%), and reduced cell density (fewer neighbors). Mitochondrial organization appears disrupted with increased texture contrast and entropy, suggesting fragmented or disorganized mitochondrial networks. Cell shape becomes less compact and circular, with reduced solidity, indicating cytoskeletal disruption and irregular morphology typical of metabolic stress. Overall, the dominant change is smoother, more ordered RNA texture consistent with nucleolar

*Overall_mechanism_linked_summary*
The specimen is hypothesized to reflect PI3K class I inhibition by BGT226, which should disrupt cellular metabolism and survival pathways. Below, I assess whether the observed morphology aligns with this hypothesis using quantitative feature trends and mechanistic reasoning. The observed phenotype strongly supports the proposed PI3K/mTOR inhibition mechanism. Key evidence includes significantly increased mitochondrial texture contrast and entropy (q<1e-06), enlarged and elongated cells with irregular shapes, and reduced cell density. Given the proposed MoA of PI3K class I inhibition, it is known to alter cellular metabolism and survival signaling via disrupted PI3K/Akt/mTOR pathways, which should cause mitochondrial dysfunction, cell cycle arrest, and cytoskeletal disruption; this would present as enlarged irregular cells with disorganized mitochondria (observed: +64% texture contrast, +41% cell area, reduced form factor). Two features showed insufficient evidence due to weak statistical significance, and dose-response studies would help confirm the mechanism specificity and rule out general cytotoxicity.

## O.5 ADDITIONAL CASE 5: AZ841 IN MCF7

**Cell Painting Drug Experiment Brief — *AZ841 in MCF7 (DNA, 20X; 24μM, 24h)***

**1. Most Matched Experimental Context**
• Cell line: MCF7
• Culture medium: RPMI 1640, 10% FBS, 1% GlutaMAX, 900 μg/mL G418
• Imaging: DNA channel, 20X objective
• Compound: AZ841(24 μM, 24h)
• Mechanism of action: Aurora Kinase Inhibitor

**2. Visual Overview**: Control vs Perturbation Images

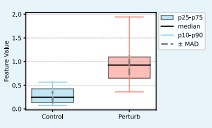

Control        Perturbed

**3. Anticipated Feature Changes Based on Mechanism**

• Expected effects of Aurora kinase inhibition (e.g., AZ841, 24μM, 24h):
Aurora kinase inhibition is expected to cause mitotic arrest, cytokinesis failure, and apoptotic fragmentation. This leads to a mixed nuclear phenotype including small condensed mitotic nuclei, enlarged or multinucleated cells, and micronuclei or nuclear debris.
• Relative to control, Aurora kinase inhibitor treatment is expected to cause:
  • Mitotic arrest with small, round, hyperintense chromatin
• Polyploidy and multinucleation due to cytokinesis failure
  • Formation of micronuclei and nuclear fragmentation
  • Irregular nuclear shapes and boundaries
  • Increased chromatin heterogeneity and punctate signals
  • Possible reduction in cell count and altered nuclear packing
• These changes are expected to manifest in Cell Painting features as:
  - ↑ Nuclear size and dispersion (e.g., AreaShape_Area, AreaShape_EquivalentDiameter, AreaShape_MaxFeretDiameter)
  - ↑ Shape irregularity and fragmentation (e.g., AreaShape_Eccentricity, AreaShape_Solidity, AreaShape_FormFactor, AreaShape_Perimeter, AreaShape_EulerNumber)
  - ↑ Mid- and coarse-scale DNA granularity (e.g., Granularity_2_DNA, Granularity_3_DNA)
  - ↑ DNA texture heterogeneity and local variability (e.g., Texture_Contrast_DNA_5_02_256, Texture_Entropy_DNA_5_02_256, Texture_Variance_DNA_5_02_256, Texture_SumVariance_DNA_5_02_256, Texture_DifferenceEntropy_DNA_5_02_256)
  - ↓ Texture smoothness / homogeneity (e.g., Texture_InverseDifferenceMoment_DNA_5_02_256, Texture_Correlation_DNA_5_02_256, Texture_AngularSecondMoment_DNA_5_02_256)
  - ↓ Object count (due to cell loss) (e.g., Number_Object_Number)
  - ↑ Neighbor contact due to clustering or enlarged nuclei (e.g., Neighbors_PercentTouching_50)

**4. Key Feature Evidence from Data**

*Granularity_2*
Direction: increase
Observed evidence: Median increased from 0.298 to 0.818 (Δ = +0.520; CI [0.431, 0.635]); very large effect (Cliff's d = −0.931), q = 3.8e−18
Mechanism link: Aurora kinase inhibition can cause condensed or fragmented chromatin and micronuclei, which would elevate DNA granularity at intermediate scales.
Supports proposed mechanism: ✓ Yes (0.98 confidence)

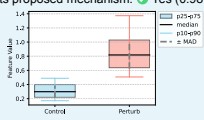

*Granularity_3*
Direction: increase
Observed evidence: Median increased from 0.246 to 0.926 (Δ = +0.680; CI [0.532, 0.778]); large effect (Cliff's d = −0.796), q = 1.3e−13
Mechanism link: Larger-scale chromatin clumping/micronuclei from mitotic errors would increase coarse DNA granularity.
Supports proposed mechanism: ✓ Yes (0.96 confidence)

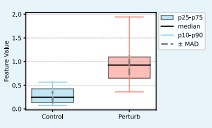

*AreaShape_MaxFeretDiameter*
Direction: increase
Observed evidence: Median increased from 39.26 to 50.60 (Δ = +11.33; CI [7.21, 14.65]); moderate effect (Cliff's d = −0.612), q = 2.1e−08
Mechanism link: Polyploidy/multinucleation can yield larger nuclear extents, increasing maximum Feret diameter.
Supports proposed mechanism: ✓ Yes (0.90 confidence)

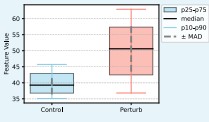

*AreaShape_Solidity*
Direction: decrease
Observed evidence: Median decreased from 0.973 to 0.960 (Δ = −0.013; CI [−0.021, −0.006]); moderate effect (Cliff's d = 0.582), q = 8.6e−08
Mechanism link: Irregular or lobulated nuclei/multinuclear aggregates from cytokinesis failure lower solidity.
Supports proposed mechanism: ✓ Yes (0.88 confidence)

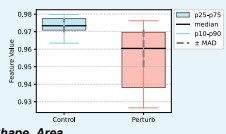

*AreaShape_FormFactor*
Direction: decrease
Observed evidence: Median decreased from 0.911 to 0.821 (Δ = −0.090; CI [−0.115, −0.069]); moderate effect (Cliff's d = 0.554), q = 3.16e−07
Mechanism link: Nuclear irregularity, indentations, or partial fragmentation expected with chromosome mis-segregation reduce form factor (roundness).
Supports proposed mechanism: ✓ Yes (0.86 confidence)

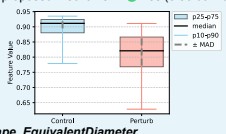

*AreaShape_Perimeter*
Direction: increase
Observed evidence: Median increased from 114.15 to 152.02 (Δ = +37.87; CI [17.78, 45.21]); moderate effect (Cliff's d = −0.504), q = 3.55e−06
Mechanism link: Enlarged and more complex nuclear contours from multinucleation/irregularity increase perimeter.
Supports proposed mechanism: ✓ Yes (0.84 confidence)

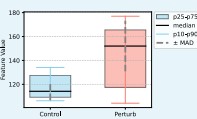

*AreaShape_Area*
Direction: increase
Observed evidence: Median increased from 955 to 1501 (Δ = +546; CI [147, 726]); moderate effect (Cliff's d = −0.419), q = 1.25e−04
Mechanism link: Polyploid or binucleated cells after cytokinesis failure exhibit larger nuclear area.
Supports proposed mechanism: ✓ Yes (0.82 confidence)

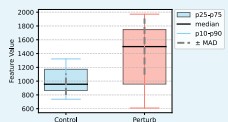

*AreaShape_EquivalentDiameter*
Direction: increase
Observed evidence: Median increased from 34.87 to 43.72 (Δ = +8.85; CI [2.56, 11.58]); moderate effect (Cliff's d = −0.419), q = 1.25e−04
Mechanism link: Enlarged nuclei with higher ploidy raise equivalent diameter.
Supports proposed mechanism: ✓ Yes (0.80 confidence)

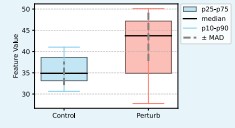

*Texture_Variance_5_02_256*
Direction: decrease
Observed evidence: Median decreased from 554.32 to 332.95 (Δ = −221.37; CI [−395.86, −73.95]); moderate effect (Cliff's d = 0.400), q = 2.42e−04
Mechanism link: Condensed chromosomes within mitotic or abnormal nuclei can reduce local intensity variance at this scale.
Supports proposed mechanism: ✓ Yes (0.65 confidence)

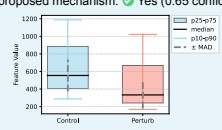

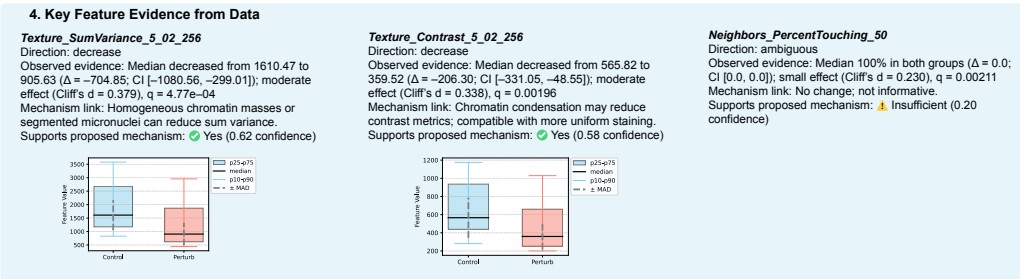

**Cell Painting Drug Experiment Brief — *AZ841 in MCF7 (DNA, 20X; 24µM, 24h)***

**4. Key Feature Evidence from Data**

*Texture_SumVariance_5_02_256*
Direction: decrease
Observed evidence: Median decreased from 1610.47 to 905.63 (Δ = −704.85; CI [−1080.56, −299.01]); moderate effect (Cliff's d = 0.379), q = 4.77e−04
Mechanism link: Homogeneous chromatin masses or segmented micronuclei can reduce sum variance.
Supports proposed mechanism: ✅ Yes (0.62 confidence)

*Texture_Contrast_5_02_256*
Direction: decrease
Observed evidence: Median decreased from 565.82 to 359.52 (Δ = −206.30; CI [−331.05, −48.55]); moderate effect (Cliff's d = 0.338), q = 0.00196
Mechanism link: Chromatin condensation may reduce contrast metrics; compatible with more uniform staining.
Supports proposed mechanism: ✅ Yes (0.58 confidence)

*Neighbors_PercentTouching_50*
Direction: ambiguous
Observed evidence: Median 100% in both groups (Δ = 0.0; CI [0.0, 0.0]); small effect (Cliff's d = 0.230), q = 0.00211
Mechanism link: No change; not informative.
Supports proposed mechanism: ⚠️ Insufficient (0.20 confidence)

**5. Mechanism Assessment and Conclusion**

*Key Evidence (Features with Large Effect Sizes):*
*1.Granularity_2_DNA:*
↑ *Median by +0.52, q = 3.8e−18*
*Strongest evidence for chromatin condensation/micronuclei*
*2.Granularity_3_DNA:*
↑ *Median by +0.68, q = 1.3e−13*
*Further supports increased coarse chromatin structure*
*3.AreaShape_MaxFeretDiameter / Area / EquivalentDiameter / Perimeter:*
↑ *Indicates nuclear enlargement and irregular boundaries*
*4.AreaShape_Solidity / FormFactor:*
↓ *Indicates nuclear fragmentation or lobulation*
*5.Texture metrics (Variance, SumVariance, Contrast):*
↓ *Suggest reduced internal heterogeneity, consistent with chromatin compaction*

*Morphology_summary*
The perturbed nuclei show strong increases in DNA granularity at multiple scales and larger, more irregular nuclear shapes (higher area, equivalent diameter, perimeter; reduced solidity and form factor). Cell counts per field trend lower. DNA texture metrics generally decrease in variance/contrast, suggesting more compact or discretized chromatin. Together, these patterns are consistent with mitotic errors leading to multinucleation/polyploidy and chromatin condensation. Overall, the dominant change is smoother, more ordered RNA texture consistent with nucleolar compaction or consolidation.

*Overall_mechanism_linked_summary*
The specimen is hypothesized to reflect aurora kinase inhibition causing mitotic disruption. Below, I assess whether the observed morphology aligns with this hypothesis using quantitative feature trends and mechanistic linkage.
Mechanism verdict: The quantitative phenotype is broadly consistent with aurora kinase inhibition. Evidence summary: Strong increases in Granularity_2/3_DNA (deltas +0.52/+0.68; q ≤ 1e-13) and enlarged, more irregular nuclei (Area/Perimeter/MaxFeret up; Solidity/FormFactor down; q ≤ 1e-04) dominate. Texture metrics (Variance, SumVariance, Contrast) decrease with q ≤ 0.002, while cell number modestly drops with CI touching 0. Non-informative features show no significant change.
Mechanistic linkage sentence: Given the proposed MoA aurora kinase inhibitors, it is known to alter mitotic progression via inhibition of chromosome segregation/cytokinesis, which should cause mitotic arrest and multinucleation/polyploidy; this would present as increased DNA granularity and larger, irregular nuclei with potential micronuclei (observed: Granularity_2/3 up; Area/EquivalentDiameter/Perimeter up; Solidity/FormFactor down).
Caveats and alternatives: Some texture changes are modest and several features are non-significant, and we only have DNA channel at a single timepoint. Follow-up with multi-channel Cell Painting (tubulin/actin), cell cycle profiling, and dose–response/time-course would strengthen mechanistic attribution and distinguish from other mitotic poisons.

## P  REASONING EVALUATION CRITERIA

The survey is designed via Google Form, and can be accessed here: `https://docs.google.com/forms/d/e/1FAIpQLSc_W2x6ro6huDANCTaOwc5IGvJ2PUXyvt2zMIKYIlI2npyi3w/viewform?usp=header`

To facilitate consistent and high-quality responses, we shared the following rubric and example list with participated experts as initial guidance. This framework outlines key criteria for evaluating **Language Quality** and **Reasoning Quality** of model-generated explanations in biological tasks. The rubric emphasizes five core aspects of language quality—including accuracy, relevance, coherence, depth, and conciseness, as well as five reasoning quality metrics such as pattern recognition, stepwise reasoning, biological deduction, hypothesis formation, and mechanistic insight. Each criterion is paired with both positive and negative examples to help clarify expectations and common pitfalls.

## P.1 LANGUAGE QUALITY CRITERIA

## Language Quality Criteria

| Criteria | Excellent Performance | ☑ Positive Examples | ✖ Negative Examples |
|---|---|---|---|
| 1. Accuracy - Terminology, data, and mechanism descriptions are correct; no factual errors. | Uses correct biological terms, mechanism names, and data. Describes cell processes and drug mechanisms accurately. | ☑**Example 1:** "Eg5 inhibition leads to monopolar spindle formation, a hallmark of mitotic arrest." Explanation: Correctly links Eg5 inhibition to monopolar spindle formation and mitotic arrest, demonstrating mechanistic accuracy.

☑**Example 2:** "KIF11, also known as Eg5, is essential for centrosome separation during mitosis." Explanation: Accurately identifies KIF11 as Eg5 and correctly explains its role in centrosome separation. | ✖ **Example**: "Granularity in the cytoplasm reflects chromatin condensation during mitosis." **Explanation**: Chromatin condensation occurs in the nucleus, not the cytoplasm. This shows incorrect terminology. |
| 2. Relevance - Focused on the core problem; avoids unrelated content. | Stays focused on the image-based mechanism, features, and hypothesis testing. Avoids discussing unrelated pathways. . | ☑ **Example:** "Texture changes are evaluated in the context of mitotic arrest, not other cell cycle stages." Explanation: Stays on-topic by linking texture features specifically to mitotic arrest. | ✖ **Example** "EGFR inhibitors are commonly used in lung cancer therapy and act on tyrosine kinase domains." **Explanation**: If the task is about Eg5 inhibition, discussing EGFR is irrelevant and off-topic. |
| 3. Coherence - Logical flow, structured reasoning, and natural transitions. | Clear cause-effect relationships between observations and interpretations. | ☑ **Example 1:** "Because Eg5 inhibition blocks bipolar spindle formation, the observed increase in DNA granularity is expected." Explanation: Uses a "because → therefore" structure to logically connect mechanism to observation.

☑ **Example 2:** "We first observe increased chromatin granularity and reduced cell number. Given these findings, we hypothesize Eg5 inhibition as the likely mechanism, which aligns with known spindle dysfunction phenotypes." Explanation: Well-structured progression from observation to hypothesis and biological context. | ✖ **Example** "The cells look abnormal. Therefore, Eg5 inhibition is the cause." **Explanation**: Jumps to conclusion without explaining intermediate steps like spindle defects or mitotic arrest. |
| 4. Depth - Goes beyond "what" to explain "why"; considers alternative mechanisms or limitations. | Provides mechanistic reasoning, alternative explanations, or validation proposals. | ☑ **Example 1:** "Although increased granularity may suggest mitotic arrest, it could also reflect apoptosis; further staining is needed." Explanation: Considers multiple hypotheses and proposes validation, showing analytical depth.

☑ **Example 2:** "To confirm that increased granularity results from mitotic arrest, time-lapse imaging could be used to track cell cycle progression in real time." Explanation: Suggests a forward-looking validation approach, demonstrating a deeper level of reasoning. | ✖ **Example** "The cells show increased granularity and reduced number." **Explanation**: Only observes phenomena without explaining their significance or underlying cause. |
| 5. Conciseness - Clear and efficient language; no redundancy. | Expresses complete logic using minimal words. | ☑ **Example 1:** "Granularity ↑, Entropy ↓ — consistent with chromatin condensation under Eg5 inhibition." **Explanation**: Uses symbolic shorthand to summarize findings clearly and effectively.

☑ **Example 2:** "Mitotic arrest inferred from monopolar spindles and chromatin compaction." **Explanation**: Omits unnecessary words yet remains scientifically complete & precise. | ✖ **Example 1:** "The texture of the chromatin appears to be more granular and also shows increased granularity in its texture." **Explanation**: Repetitive phrasing; the same idea is stated twice.

✖ **Example 2:** "Due to the potential inhibition of Eg5, which is known to be related to spindle formation during mitosis, the cells may possibly experience something like a blockage in mitotic progression." **Explanation**: Wordy, vague, and redundant. Can be simplified to: "Eg5 inhibition likely caused mitotic arrest." |

Figure 14: Language quality criteria for evaluating CP-Agent generated Cell Painting reports.

## P.2 REASONING QUALITY CRITERIA

**Reasoning** Quality Criteria

| Criteria | Excellent Performance | ✅ Positive Examples | ❌ Negative Examples |
|---|---|---|---|
| **6. Pattern Recognition** 

Ability to identify key visual differences such as cell morphology or staining patterns and link them to biological meaning. | Connects visual features with plausible mechanisms. | ✅ **Example 1:** "Granular, compact chromatin morphology is consistent with mitotic arrest." **Explanation:** Recognizes dense, granular chromatin as a sign of mitotic arrest. 

 ✅ **Example 2:** "Reduced cell count and round, compact nuclei are consistent with mitotic accumulation and arrest." **Explanation:** Integrates multiple visual cues to explain a biological state. | ❌ **Example 1:** "The cells look mostly the same as normal." **Explanation:** Fails to recognize evident morphological changes. 

 ❌ **Example 2:** "The blurry area in the cytoplasm might be the nucleolus." **Explanation:** Confuses cytoplasmic structure with nuclear organelles, showing poor structural understanding. |
| **7. Algorithmic Reasoning (Stepwise Thinking)** 
 - Systematic step-by-step reasoning from visual features to diagnostic conclusion. | Follows a clear "observe → infer → verify" logic chain. | ✅ **Example 1:** "Step 1: Check DNA granularity ↑ → Step 2: Consider mitotic arrest → Step 3: Confirm with texture shift → Conclusion: Eg5 inhibition likely." **Explanation:** Follows a diagnostic-style reasoning flow. 

 ✅ **Example 2:** "Metric: High DNA granularity + low heterogeneity → Hypothesis: Nuclear compaction → Biological context: Consistent with metaphase arrest → Likely cause: Eg5 inhibition." **Explanation:** Builds a multistep logic from feature to mechanism. | ❌ **Example 1:** "This is clearly due to Eg5 inhibition." **Explanation:** Conclusion is stated without supporting steps or evidence. 
 ❌ **Example 2:** "Maybe it's apoptosis, but the chromatin is dense and also the granularity is high. Eg5 is involved in spindles." **Explanation:** Disorganized reasoning, lacks structured flow. |
| **8. Deductive Reasoning -** Uses known biological rules to predict specific outcomes. | Explains observed features using established mechanisms or canonical pathways. | ✅ **Example 1:** "If Eg5 is inhibited, bipolar spindle formation is blocked → cells accumulate in mitosis → chromatin condenses." **Explanation:** Demonstrates a clear biological cause-effect chain from inhibition to phenotype. 

 ✅ **Example 2:** "Apoptosis leads to nuclear fragmentation and increased DNA texture heterogeneity. This would appear as irregular, punctate chromatin staining." **Explanation:** Applies known apoptosis features to interpret image data. | ❌ **Example 1:** "If Eg5 is inhibited, chromatin looks like this." **Explanation:** Skips required mechanistic reasoning steps; lacks causality. 

 ❌ **Example 2:** "Because mitosis is complicated, maybe that's why the chromatin looks dense." **Explanation:** Vague and unscientific language; lacks specific mechanistic explanation. |
| **9. Induction / Hypothesis Testing -** Forms hypotheses from observations and supports or refines them with evidence. | Proposes alternative hypotheses, weighs evidence, and draws reasoned conclusions. | ✅ **Example 1:** "Hypothesis: DNA granularity suggests either mitotic arrest or apoptosis. Evidence: Low Entropy + High Contrast → favors mitosis." **Explanation:** Proposes alternatives and uses features to evaluate them. 

 ✅ **Example 2:** "Hypothesis: Granular chromatin → mitotic arrest. To validate: Use PH3 staining to confirm mitotic accumulation." **Explanation:** Suggests hypothesis and a concrete method for testing it. | ❌ **Example 1:** "This must be Eg5 inhibition." **Explanation:** States a conclusion without forming or testing a hypothesis. |
| **10. Mechanistic Insight** Links visual observations to underlying molecular or cellular pathways. | Traces a causal path from molecular intervention → cellular structure/function → image features. | ✅ **Example 1:** "Eg5 inhibition prevents centrosome separation, leading to monopolar spindles, which induce checkpoint-mediated mitotic arrest." **Explanation:** Demonstrates a full causal chain from drug action to phenotype. 

 ✅ **Example 2:** "Mitotic cells lose substrate adhesion due to reorganization of cortical actin and detachment from the ECM, resulting in rounded morphology in imaging." **Explanation:** Explains how cytoskeletal changes translate to visual cell shape. | ❌ **Example 1:** "This must be mitotic arrest because the nuclei look dense." **Explanation:** Observation is not linked to any molecular or cellular mechanism; lacks causal reasoning. |

Figure 15: Reasoning quality criteria for evaluating CP-Agent generated Cell Painting reports.

## Q   EXPERT RATINGS OF CP-AGENT GENERATED REPORTS ACROSS LANGUAGE AND REASONING CRITERIA

### Q.1   HUMAN EXPERT ASSESSMENT OF PERTURBATION REPORT QUALITY ACROSS LLMS

Figure 16 summarizes expert evaluations across ten rubric criteria, split into five language quality dimensions (Figure 16a) and five reasoning quality dimensions (Figure 16b). On average, all four LLMs received high ratings (mostly above 5.0 on a 7-point scale), indicating strong performance in generating biologically grounded screening reports. Among the models, GPT-5 consistently achieved the highest scores across most reasoning metrics—including pattern recognition, algorithmic reasoning, and mechanistic insight—while also maintaining strong language quality. Gemini-2.5-Pro closely followed, particularly excelling in relevance and coherence. Claude-Sonnet-4 underperformed slightly in mechanistic insight and inductive reasoning, indicating slightly weaker performance in higher-order biological inference. Grok-4 showed relatively balanced language quality but lagged slightly in depth and coherence compared to top-performing models. The bar chart (Figure 16c) further illustrates per-metric mean scores, reinforcing the finding that reasoning dimensions pose a greater challenge than surface-level language quality, especially in tasks requiring mechanistic interpretation and hypothesis generation.

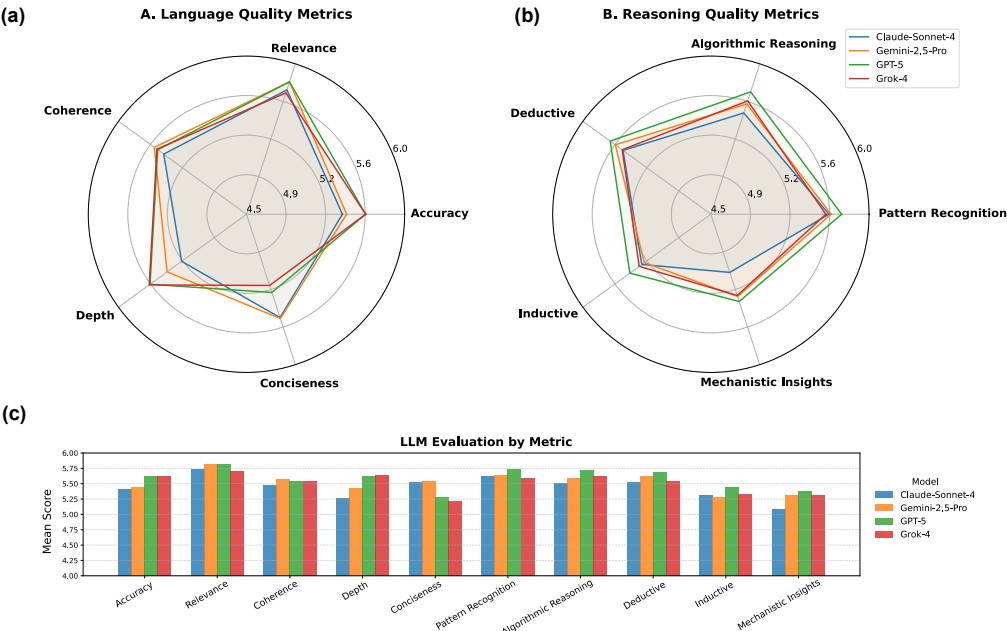

Figure 16: **Expert evaluation of LLM-generated screening reports across language and reasoning dimensions.**(a–b) Mean expert ratings (on a 7-point scale) for language and reasoning quality, based on ten rubric-based evaluation criteria. (c) Bar chart summarizing per-metric mean scores across the four evaluated models: Claude-Sonnet-4 (blue), Gemini-2.5-Pro (orange), GPT-5 (green), and Grok-4 (red).

### Q.2   EXPERT EVALUATION CONSISTENCY ANALYSIS

To assess the reliability and consistency of expert evaluations used in our study, a comprehensive inter-rater agreement analysis is conducted. The results are detailed in Table 13 and summarized in Table 14. We employed several standard metrics to quantify agreement among expert annotators, including Kendall's W, standard, deviation, exact agreement percentage, ±1 rating agreement, Mean absolute, difference (MAD). Kendall's W values ranged from 0.33 to 0.37 across different models, indicating fair agreement among annotators. At the pairwise level, 73.3% of all rating pairs fell

within ±1 point on the 0–7 scale, and the mean absolute difference (MAD) was 1.04, suggesting a high degree of rating consistency. Given the limited number of annotators, bootstrap resampling is employed to compute 95% confidence intervals for all key reliability metrics, ensuring robust statistical estimation.

Although the expert ratings did not reveal statistically significant differences between models, we observed consistent trends across raters—particularly that GPT-5 achieved marginally higher average scores across most evaluation dimensions. However, these differences were relatively small, implying that all evaluated LLMs were able to generate informative, relevant, and biologically meaningful reports when operating under our CP-Agent generation pipeline.

These findings collectively support the internal consistency and reliability of the expert evaluation process, reinforcing confidence in the qualitative assessments reported in the main text.

Table 13: Inter-rater Evaluation for Expert Evaluation

| Model | Metric | Kendall_W | Kendall_W_L | Kendall_W_U | SD | SD_L | SD_U | Exact% | Exact_L | Exact_U | Near% | Near_L | Near_U | MAD | MAD_L | MAD_U |
|---|---|---|---|---|---|---|---|---|---|---|---|---|---|---|---|---|
| Claude-Sonnet-4 | Accuracy | 0.303 | 0.148 | 0.495 | 1.01 | 0.89 | 1.14 | 24.1 | 20.2 | 28.4 | 68.7 | 62.6 | 74.7 | 1.16 | 1.02 | 1.31 |
| | Relevance | 0.349 | 0.194 | 0.548 | 0.95 | 0.85 | 1.04 | 26.0 | 22.4 | 30.4 | 69.4 | 62.1 | 77.1 | 1.11 | 0.96 | 1.24 |
| | Coherence | 0.330 | 0.151 | 0.523 | 0.87 | 0.78 | 0.94 | 30.4 | 25.8 | 36.5 | 75.9 | 69.1 | 82.9 | 0.97 | 0.83 | 1.09 |
| | Depth | 0.291 | 0.122 | 0.484 | 0.91 | 0.85 | 0.97 | 27.2 | 23.5 | 31.3 | 74.5 | 70.8 | 77.6 | 1.03 | 0.95 | 1.13 |
| | Conciseness | 0.244 | 0.106 | 0.434 | 1.05 | 0.90 | 1.23 | 24.8 | 22.0 | 26.9 | 70.4 | 65.0 | 75.6 | 1.17 | 1.03 | 1.34 |
| | Pattern Recognition | 0.312 | 0.138 | 0.520 | 0.89 | 0.77 | 1.04 | 29.9 | 26.2 | 33.2 | 77.1 | 71.2 | 83.3 | 0.99 | 0.86 | 1.14 |
| | Algorithmic Reasoning | 0.368 | 0.153 | 0.650 | 0.88 | 0.82 | 0.94 | 28.1 | 24.9 | 32.0 | 75.9 | 70.9 | 81.1 | 1.00 | 0.90 | 1.09 |
| | Deductive | 0.402 | 0.200 | 0.628 | 0.90 | 0.83 | 0.99 | 29.8 | 26.4 | 33.5 | 74.0 | 68.4 | 78.7 | 1.01 | 0.91 | 1.12 |
| | Inductive | 0.367 | 0.181 | 0.619 | 0.92 | 0.81 | 1.02 | 27.6 | 22.2 | 34.4 | 73.7 | 66.6 | 81.5 | 1.04 | 0.88 | 1.20 |
| | Mechanistic Insights | 0.342 | 0.152 | 0.563 | 1.03 | 0.93 | 1.13 | 23.7 | 21.1 | 26.0 | 66.9 | 60.7 | 72.7 | 1.18 | 1.07 | 1.30 |
| Gemini-2.5-Pro | Accuracy | 0.370 | 0.171 | 0.565 | 0.94 | 0.86 | 1.01 | 25.3 | 22.2 | 28.4 | 72.2 | 66.3 | 77.5 | 1.08 | 0.97 | 1.18 |
| | Relevance | 0.409 | 0.191 | 0.664 | 0.94 | 0.87 | 1.00 | 25.0 | 22.4 | 27.7 | 72.3 | 68.2 | 76.5 | 1.08 | 1.00 | 1.16 |
| | Coherence | 0.373 | 0.181 | 0.548 | 0.84 | 0.76 | 0.90 | 29.8 | 24.9 | 35.1 | 79.2 | 74.0 | 84.5 | 0.94 | 0.83 | 1.05 |
| | Depth | 0.300 | 0.119 | 0.518 | 0.85 | 0.80 | 0.90 | 30.4 | 27.1 | 33.8 | 77.0 | 73.4 | 80.9 | 0.95 | 0.88 | 1.03 |
| | Conciseness | 0.302 | 0.151 | 0.472 | 0.89 | 0.83 | 0.94 | 27.0 | 24.7 | 29.5 | 75.2 | 72.0 | 78.2 | 1.02 | 0.95 | 1.08 |
| | Pattern Recognition | 0.359 | 0.205 | 0.526 | 0.84 | 0.74 | 0.91 | 30.5 | 26.1 | 35.6 | 78.3 | 73.0 | 84.8 | 0.94 | 0.82 | 1.05 |
| | Algorithmic Reasoning | 0.381 | 0.179 | 0.597 | 0.81 | 0.73 | 0.90 | 31.9 | 27.9 | 35.6 | 80.5 | 74.4 | 86.8 | 0.90 | 0.79 | 1.02 |
| | Deductive | 0.412 | 0.207 | 0.682 | 0.89 | 0.79 | 1.00 | 31.6 | 26.6 | 36.4 | 75.4 | 68.8 | 81.1 | 0.99 | 0.86 | 1.13 |
| | Inductive | 0.395 | 0.174 | 0.651 | 0.93 | 0.87 | 0.99 | 29.6 | 24.9 | 35.2 | 68.7 | 63.6 | 74.1 | 1.06 | 0.96 | 1.14 |
| | Mechanistic Insights | 0.304 | 0.168 | 0.499 | 0.93 | 0.84 | 1.03 | 28.3 | 23.3 | 33.4 | 71.9 | 65.2 | 78.7 | 1.06 | 0.93 | 1.19 |
| GPT-5 | Accuracy | 0.337 | 0.175 | 0.504 | 0.95 | 0.86 | 1.04 | 27.2 | 22.4 | 31.8 | 72.1 | 66.5 | 77.3 | 1.08 | 0.96 | 1.21 |
| | Relevance | 0.354 | 0.198 | 0.533 | 0.90 | 0.84 | 0.98 | 27.7 | 24.7 | 31.1 | 70.4 | 65.3 | 75.4 | 1.05 | 0.96 | 1.15 |
| | Coherence | 0.357 | 0.165 | 0.614 | 0.89 | 0.83 | 0.95 | 30.8 | 25.6 | 37.5 | 73.7 | 69.6 | 77.3 | 0.98 | 0.89 | 1.07 |
| | Depth | 0.307 | 0.139 | 0.526 | 0.92 | 0.83 | 1.00 | 29.6 | 25.3 | 34.0 | 71.3 | 63.8 | 78.0 | 1.04 | 0.92 | 1.15 |
| | Conciseness | 0.303 | 0.140 | 0.469 | 0.97 | 0.83 | 1.16 | 28.2 | 24.0 | 32.0 | 71.5 | 63.5 | 77.6 | 1.08 | 0.94 | 1.28 |
| | Pattern Recognition | 0.360 | 0.207 | 0.540 | 0.81 | 0.72 | 0.89 | 32.7 | 28.0 | 38.5 | 79.6 | 74.4 | 84.7 | 0.91 | 0.78 | 1.02 |
| | Algorithmic Reasoning | 0.422 | 0.245 | 0.609 | 0.86 | 0.73 | 0.99 | 30.6 | 24.5 | 37.2 | 78.3 | 69.3 | 86.7 | 0.96 | 0.79 | 1.15 |
| | Deductive | 0.494 | 0.306 | 0.669 | 0.92 | 0.87 | 0.98 | 25.8 | 23.5 | 28.2 | 72.1 | 68.2 | 75.7 | 1.07 | 1.00 | 1.15 |
| | Inductive | 0.424 | 0.254 | 0.609 | 0.95 | 0.91 | 0.99 | 24.5 | 22.2 | 27.1 | 68.7 | 66.5 | 70.7 | 1.12 | 1.07 | 1.17 |
| | Mechanistic Insights | 0.358 | 0.204 | 0.521 | 0.96 | 0.88 | 1.04 | 28.2 | 23.1 | 34.2 | 67.1 | 62.4 | 72.5 | 1.11 | 0.98 | 1.22 |
| Grok-4 | Accuracy | 0.278 | 0.120 | 0.493 | 0.93 | 0.81 | 1.06 | 27.8 | 23.7 | 32.2 | 73.3 | 66.4 | 80.2 | 1.06 | 0.91 | 1.21 |
| | Relevance | 0.403 | 0.214 | 0.633 | 0.92 | 0.84 | 0.98 | 25.7 | 22.9 | 29.1 | 73.4 | 68.5 | 79.0 | 1.05 | 0.94 | 1.15 |
| | Coherence | 0.367 | 0.199 | 0.586 | 0.81 | 0.68 | 0.94 | 35.6 | 31.2 | 39.7 | 81.1 | 73.8 | 88.7 | 0.87 | 0.73 | 1.03 |
| | Depth | 0.300 | 0.136 | 0.503 | 0.97 | 0.87 | 1.07 | 25.8 | 23.5 | 28.1 | 70.7 | 65.0 | 75.6 | 1.10 | 1.00 | 1.23 |
| | Conciseness | 0.265 | 0.104 | 0.454 | 1.04 | 0.91 | 1.16 | 22.9 | 19.3 | 26.7 | 67.1 | 60.9 | 73.8 | 1.20 | 1.04 | 1.34 |
| | Pattern Recognition | 0.359 | 0.171 | 0.582 | 0.90 | 0.81 | 1.00 | 29.5 | 25.3 | 33.3 | 73.7 | 67.1 | 78.8 | 1.01 | 0.90 | 1.16 |
| | Algorithmic Reasoning | 0.399 | 0.236 | 0.593 | 0.85 | 0.69 | 0.99 | 33.7 | 26.2 | 41.5 | 79.7 | 71.4 | 88.6 | 0.92 | 0.73 | 1.11 |
| | Deductive | 0.498 | 0.279 | 0.700 | 0.89 | 0.81 | 0.98 | 28.8 | 24.4 | 32.5 | 76.3 | 71.6 | 79.8 | 1.00 | 0.91 | 1.11 |
| | Inductive | 0.483 | 0.270 | 0.677 | 0.98 | 0.91 | 1.05 | 25.4 | 22.5 | 28.4 | 69.0 | 64.4 | 73.5 | 1.13 | 1.04 | 1.23 |
| | Mechanistic Insights | 0.323 | 0.149 | 0.519 | 0.99 | 0.86 | 1.11 | 29.1 | 26.0 | 32.4 | 66.6 | 58.9 | 74.7 | 1.12 | 0.97 | 1.27 |

Table 14: Summary of Inter-rater Evaluations Across LLMs

| Model | Kendall_W | Score_Avg | Score_Std | Near_Agreement % |
|---|---|---|---|---|
| Claude-Sonnet-4 | 0.33 | 5.45 | 0.94 | 72.65 |
| Gemini-2.5-Pro | 0.36 | 5.53 | 0.89 | 75.07 |
| GPT-5 | 0.37 | 5.59 | 0.91 | 72.48 |
| Grok-4 | 0.37 | 5.51 | 0.93 | 73.09 |

# R  ROBUSTNESS EVALUATION: FEATRANK AGENT AND REPORTGEN AGENT

The reproducibility of CPAgent's reasoning outputs is primarily influenced by the temperature parameter in the large language models (LLMs) since the perturbation codition is infered from CP-CLIP, which makes the conclusion of the report deterministic already. In our pipeline, following the initial pretrained CLIP model, there are two LLM modules, the 'FeatRank Agent', which ranks CellProfiler extracted morphology features, and 'ReportGen Agent', which generates natural language reports based on the ranked features and contextual information. To ensure that the feature ranking step is as deterministic as possible, we set the temperature of the FeatRank Agent to 0. To systematically evaluate the reproducibility, we designed five experiments with varying temperature settings, the ReportGen Agent's temperature was set ranging from 0.0 to 1.0, while keeping the

FeatRank Agent temperature fixed at 0.0. In each setting, we repeated the pipeline 30 times on the same input samples and analyzed the consistency of the selected features and generated reports.

## SYSTEM PROMPT

You are a scientific evaluator specialized in assessing corpora of mechanism assessment reports derived from Cell Painting assays.

Your task is to evaluate the internal consistency of a set of 30 mechanism assessment reports. Each report analyzes how observed morphological features align with a hypothesized mechanism of action for a specific chemical perturbation.

You are not evaluating the scientific accuracy or biological correctness of any individual report. Your focus is on how mutually consistent the reports are with each other in terms of:

- Scientific focus and biological reasoning style
- Use of technical terminology and feature names
- Presentation structure and rhetorical flow

You will be provided with the full set of reports or representative excerpts. Based on this, you will assign scores for each consistency dimension and briefly justify your evaluation.

Return only the JSON structure specified in the user prompt. Do not include any commentary or additional explanation.

## USER PROMPT

INPUTS:

- A corpus of 30 Cell Painting-based mechanism assessment reports
- Each report typically includes: Mechanism verdict, feature-based evidence summary, a mechanistic linkage explanation and caveats or alternative hypotheses.

TASK: Evaluate the corpus for internal consistency across the following three dimensions. You are not judging scientific accuracy. Focus on mutual alignment in scientific reasoning, terminology, and structure.

### 1. THEMATIC CONSISTENCY

Definition: Do all reports demonstrate same scientific purpose and consistent style of biological reasoning?

What to look for:

- Reports clearly assess whether observed phenotypes support a hypothesized MoA
- Use of biological logic at cellular or subcellular level
- Use of structured reasoning patterns: "If MoA X is true, then we expect Y phenotype, which we observe.";"This phenotype is consistent with known outcomes of X (e.g., ER stress)"
- Avoidance of vague or off-topic content
- Consistency in depth of biological interpretation

Scoring Guide:

- **10**: Reports are clear, coherent, and mechanistic
- **8–9**: Mostly consistent with minor variation in depth
- **6–7**: Some reports are descriptive only
- **4–5**: Inconsistent styles or lack MoA focus
- **1–3**: Highly divergent in purpose or reasoning

2. TERMINOLOGICAL CONSISTENCY

Definition: Are technical terms, feature names, and mechanistic labels used consistently?

What to look for:

- Uniform use of Cell Painting features (e.g., `Texture_Entropy`)
- Consistent naming of biological phenomena
- Standardized use of MoA terms (e.g., "oxidative stress")
- Avoidance of ambiguous or informal language
- Quantitative descriptors (e.g., "+0.3 increase") preferred over vague phrases

Scoring Guide:

- 10: Terminology precise and consistent
- 8–9: Minor synonym use
- 6–7: Mixed naming for same terms
- 4–5: Frequent inconsistencies
- 1–3: Terms are vague or informal

3. STRUCTURAL CONSISTENCY

Definition: Do reports follow a similar structure and rhetorical flow?

What to look for:

- Inclusion of key components: Mechanism verdict, evidence summary, mechanistic linkage, caveats or alternatives.
- Similar order, depth, and sentence structure
- Avoidance of missing or reordered sections

Scoring Guide:

- **10**: Fully consistent structure
- **8–9**: Minor variation in order or depth
- **6–7**: Incomplete or reordered sections
- **4–5**: Frequent structural differences
- **1–3**: Highly inconsistent organization

SCORING INSTRUCTIONS

- Assign a score between **1–10** for each dimension
- Provide a concise justification (<50 words)
- Return only the JSON in the following format

```
JSON Output Format

{
  "corpus_evaluation": {
    "Thematic Consistency": {
      "score": <1-10>,
      "justification": "<brief explanation>"
    },
    "Terminological Consistency": {
      "score": <1-10>,
      "justification": "<brief explanation>"
    },
    "Structural Consistency": {
      "score": <1-10>,
      "justification": "<brief explanation>"
    }
  }
}
```

CORPUS EVALUATION

Table 15: FeatRank Agent's Repeatability

| Temp 1 | Features Number Avg | Features Number std | Top 5 Stable Features |
|--------|---------------------|---------------------|-----------------------|
| 0.0 | 18.37 | 2.37 | AreaShape_Area, AreaShape_Eccentricity, Texture_Contrast_5_02_256, Granularity_2, Granularity_3 |
| 0.0 | 18.20 | 2.26 | AreaShape_Area, AreaShape_Eccentricity, Texture_Contrast_5_02_256, Granularity_2, Granularity_3 |
| 0.0 | 18.27 | 2.31 | AreaShape_Area, AreaShape_Eccentricity, Texture_Contrast_5_02_256, Granularity_2, Granularity_3 |
| 0.0 | 18.87 | 1.96 | AreaShape_Area, AreaShape_Eccentricity, Texture_Contrast_5_02_256, Granularity_2, Granularity_3 |
| 0.0 | 17.17 | 2.28 | AreaShape_Area, AreaShape_Eccentricity, Texture_Contrast_5_02_256, Granularity_2, Granularity_3 |

Table 16: Corpus score comparison across different temperature setting.

| Temp1/Temp2 | Thematic Consistency | Terminological Consistency | Structural Consistency | Averaged Corpus Score |
|-------------|----------------------|----------------------------|------------------------|-----------------------|
| 0.0/0.0 | 8.00 | 7.00 | 9.00 | 8.00 |
| 0.0/0.1 | 8.00 | 7.00 | 8.00 | 7.67 |
| 0.0/0.2 | 8.00 | 7.00 | 8.00 | 7.67 |
| 0.0/0.5 | 8.00 | 7.00 | 8.00 | 7.67 |
| 0.0/1.0 | 8.00 | 7.00 | 8.00 | 7.67 |

## S  COUNTERFACTUAL PROMPT EXPERIMENTS: DOSAGE AND TIME

To address the concern that the high retrieval performance may be due to metadata correlations (e.g., compound identity being indirectly inferred from time or dosage), rather than genuine multimodal alignment, we conducted controlled ablation experiments to isolate and evaluate the extent to which different textual components contribute to model performance. Specifically, we masked individual fields in the text prompts: compound name + MOA, concentration, or time—and measured the calculating the changes in retrieval accuracy.

As shown in Table 17, masking the compound name and MOA (Experiment 1) results in a catastrophic performance drop (e.g., text-to-image R@1 drops from 98.70 to 3.50; MRR drops by nearly 90%), indicating that the model relies heavily on compound-specific textual information. We mask the compound name and its mechanism of action (MOA) jointly, rather than separately, because these two fields are semantically correlated and often co-informative. This design choice is consistent with our setup in the drug classification experiment (Table 2), and ensures a fair and aligned evaluation across experiments. In contrast, masking concentration or time results in only moderate to negligible performance degradation (e.g., R@1 drops to 3.50 when masking compound, and only to 93.00 when masking time). This pattern suggests that while the model encodes and leverages compound identity meaningfully. To further test CP-CLIP's robustness to misleading contextual cues, we designed counterfactual prompt experiments in two settings:

- **Clean Prompt:** The target classification field was masked, but all other contextual meta-data (e.g., time, channel) remained correct.

- **Disturbed Prompt:** The target field was masked, and unrelated metadata fields were shuffled randomly, keeping only the compound identity correct. Since we have demonstrated that compound identity has a strong influence on retrieval performance.

Table 17: Retrieval performance before and after masking different textual components.

| Experiment | Metric | Original | Masked | $\delta$ Absolute | $\delta$ Relative (%) |
|---|---|---|---|---|---|
| **Experiment 1: Mask Compound Name + MOA** | | | | | |
| Text-to-Image | R@1 | 98.70 | 3.50 | -95.20 | -96.45 |
| | R@5 | 100.00 | 16.80 | -83.20 | -83.20 |
| | R@10 | 100.00 | 29.60 | -70.40 | -70.40 |
| | MRR | 0.9935 | 0.1178 | -0.8757 | -88.15 |
| Image-to-Text | R@1 | 98.70 | 3.30 | -95.40 | -96.66 |
| | R@5 | 100.00 | 16.10 | -83.90 | -83.90 |
| | R@10 | 100.00 | 28.70 | -71.30 | -71.30 |
| | MRR | 0.9935 | 0.1102 | -0.8833 | -88.90 |
| **Experiment 2: Mask Concentration Only** | | | | | |
| Text-to-Image | R@1 | 98.70 | 57.00 | -41.70 | -42.25 |
| | R@5 | 100.00 | 84.00 | -16.00 | -16.00 |
| | R@10 | 100.00 | 91.40 | -8.60 | -8.60 |
| | MRR | 0.9935 | 0.6829 | -0.3106 | -31.26 |
| Image-to-Text | R@1 | 98.70 | 55.20 | -43.50 | -44.07 |
| | R@5 | 100.00 | 79.20 | -20.80 | -20.80 |
| | R@10 | 100.00 | 87.40 | -12.60 | -12.60 |
| | MRR | 0.9935 | 0.6617 | -0.3318 | -33.39 |
| **Experiment 3: Mask Time Only** | | | | | |
| Text-to-Image | R@1 | 98.70 | 93.00 | -5.70 | -5.78 |
| | R@5 | 100.00 | 99.50 | -0.50 | -0.50 |
| | R@10 | 100.00 | 100.00 | 0.00 | 0.00 |
| | MRR | 0.9935 | 0.9590 | -0.0345 | -3.47 |
| Image-to-Text | R@1 | 98.70 | 94.70 | -4.00 | -4.05 |
| | R@5 | 100.00 | 100.00 | 0.00 | 0.00 |
| | R@10 | 100.00 | 100.00 | 0.00 | 0.00 |
| | MRR | 0.9935 | 0.9726 | -0.0209 | -2.11 |

Table 18: Concentration Classification under Masked and Counterfactual Prompts

| Concentration | Firocoxib | Opicapone | Cinoxacin | Neratinib | Hydroflumethiazide | Acetaminophen | Primidone |
|---|---|---|---|---|---|---|---|
| 0.00316µM CLIP (disturb) | 0.4866 | 0.3341 | 0.3889 | 0.2965 | 0.1634 | 0.2298 | 0.2174 |
| 0.00316µM CP-CLIP (disturb) | 0.4663 | 0.5052 | 0.4890 | 0.4887 | 0.2350 | 0.3155 | 0.3866 |
| 0.00316µM CLIP | 0.7371 | 0.5203 | 0.6491 | 0.5209 | 0.4780 | 0.4237 | 0.4167 |
| 0.00316µM CP-CLIP | 0.6866 | 0.7307 | 0.6211 | 0.6233 | 0.2819 | 0.3909 | 0.5349 |
| 0.01µM CLIP (disturb) | 0.5588 | 0.2708 | 0.1757 | 0.3731 | 0.2147 | 0.2161 | 0.2612 |
| 0.01µM CP-CLIP (disturb) | 0.3212 | 0.4409 | 0.3694 | 0.4677 | 0.4906 | 0.4598 | 0.2321 |
| 0.01µM CLIP | 0.7887 | 0.5391 | 0.3758 | 0.6267 | 0.4383 | 0.3473 | 0.5000 |
| 0.01µM CP-CLIP | 0.4356 | 0.6343 | 0.4913 | 0.5521 | 0.5783 | 0.5561 | 0.2635 |
| 0.0316µM CLIP (disturb) | 0.5714 | 0.1921 | 0.3178 | 0.2890 | 0.2017 | 0.2873 | 0.1670 |
| 0.0316µM CP-CLIP (disturb) | 0.4335 | 0.4714 | 0.3928 | 0.7949 | 0.3725 | 0.5151 | 0.4015 |
| 0.0316µM CLIP | 0.7437 | 0.4749 | 0.5839 | 0.5330 | 0.5033 | 0.5387 | 0.4628 |
| 0.0316µM CP-CLIP | 0.5754 | 0.6048 | 0.6374 | 0.9205 | 0.4771 | 0.6364 | 0.4773 |
| 0.1µM CLIP (disturb) | 0.2992 | 0.2652 | 0.3369 | 0.2866 | 0.2486 | 0.2232 | 0.1938 |
| 0.1µM CP-CLIP (disturb) | 0.4660 | 0.2935 | 0.4015 | 0.3035 | 0.4051 | 0.3000 | 0.4148 |
| 0.1µM CLIP | 0.4775 | 0.3554 | 0.3690 | 0.5203 | 0.2819 | 0.4823 | 0.2661 |
| 0.1µM CP-CLIP | 0.6402 | 0.4459 | 0.5442 | 0.4974 | 0.5014 | 0.3832 | 0.4971 |
| 0.316µM CLIP (disturb) | 0.3343 | 0.2301 | 0.2776 | 0.2131 | 0.2505 | 0.1994 | 0.2961 |
| 0.316µM CP-CLIP (disturb) | 0.5079 | 0.3333 | 0.4027 | 0.4595 | 0.2848 | 0.4962 | 0.2642 |
| 0.316µM CLIP | 0.5401 | 0.4179 | 0.6288 | 0.4591 | 0.6056 | 0.4077 | 0.5475 |
| 0.316µM CP-CLIP | 0.6272 | 0.4686 | 0.5068 | 0.6438 | 0.4399 | 0.5477 | 0.3217 |
| 1.0µM CLIP (disturb) | 0.4430 | 0.2748 | 0.2505 | 0.5455 | 0.1803 | 0.3082 | 0.3073 |
| 1.0µM CP-CLIP (disturb) | 0.6364 | 0.4411 | 0.4103 | 0.4792 | 0.4459 | 0.3974 | 0.3702 |
| 1.0µM CLIP | 0.6611 | 0.5773 | 0.6211 | 0.8856 | 0.4909 | 0.5766 | 0.4473 |
| 1.0µM CP-CLIP | 0.8137 | 0.6469 | 0.6500 | 0.6217 | 0.5133 | 0.4469 | 0.5673 |
| 3.162µM CLIP (disturb) | 0.3219 | 0.2933 | 0.2815 | 0.2119 | 0.2016 | 0.2635 | 0.2114 |
| 3.162µM CP-CLIP (disturb) | 0.5483 | 0.3324 | 0.2900 | 0.4479 | 0.2600 | 0.1577 | 0.3991 |
| 3.162µM CLIP | 0.5528 | 0.5902 | 0.5459 | 0.6442 | 0.5826 | 0.5459 | 0.3255 |
| 3.162µM CP-CLIP | 0.7599 | 0.5243 | 0.4577 | 0.5812 | 0.3433 | 0.2415 | 0.4454 |
| 10.0µM CLIP (disturb) | 0.3871 | 0.4191 | 0.2997 | 0.1502 | 0.1696 | 0.1813 | 0.2152 |
| 10.0µM CP-CLIP (disturb) | 0.6340 | 0.3924 | 0.6029 | 0.5010 | 0.3401 | 0.3562 | 0.3499 |
| 10.0µM CLIP | 0.7079 | 0.7125 | 0.5729 | 0.6616 | 0.3300 | 0.3685 | 0.5673 |
| 10.0µM CP-CLIP | 0.7895 | 0.6469 | 0.6500 | 0.5874 | 0.5133 | 0.4469 | 0.3886 |

Table 19: Time Comparison (clean vs disturb prompt): CLIP vs CP-CLIP

| Time | Ixabepilone | Methoxsalen | Sulfinpyrazone | Triamterene | Miconazole | Ceritinib | Acetohexamide |
|---|---|---|---|---|---|---|---|
| 24h CLIP (disturb) | 0.7513 | 0.5903 | 0.6473 | 0.7212 | 0.6882 | 0.6319 | 0.7650 |
| 24h CP-CLIP (disturb) | 0.9600 | 0.9556 | 0.9101 | 0.8844 | 0.9309 | 0.9323 | 0.9309 |
| 24h CLIP | 0.9950 | 1.0000 | 1.0000 | 1.0000 | 1.0000 | 0.9950 | 1.0000 |
| 24h CP-CLIP | 0.9980 | 1.0000 | 1.0000 | 1.0000 | 1.0000 | 1.0000 | 1.0000 |
| 48h CLIP (disturb) | 0.7586 | 0.6829 | 0.6218 | 0.6979 | 0.6322 | 0.6927 | 0.7650 |
| 48h CP-CLIP (disturb) | 0.9600 | 0.9592 | 0.9238 | 0.8856 | 0.9340 | 0.9375 | 0.9387 |
| 48h CLIP | 0.9950 | 1.0000 | 1.0000 | 1.0000 | 1.0000 | 0.9950 | 1.0000 |
| 48h CP-CLIP | 0.9980 | 1.0000 | 1.0000 | 1.0000 | 1.0000 | 1.0000 | 1.0000 |

Table 20: Summary of Concentration Classification Performance: CP-CLIP vs CLIP under clean vs. disturbed prompts.

| Compound | CLIP | | | CP-CLIP | | |
|---|---|---|---|---|---|---|
| | Clean | Disturbed | Δ | Clean | Disturbed | Δ |
| firocoxib | 0.5637 | 0.4163 | -0.1474 | 0.6025 | 0.5038 | -0.0987 |
| opicapone | 0.4244 | 0.2831 | -0.1413 | 0.4631 | 0.3956 | -0.0675 |
| cinoxacin | 0.3962 | 0.2875 | -0.1087 | 0.4500 | 0.4231 | -0.0269 |
| neratinib | 0.5131 | 0.2875 | -0.2256 | 0.5269 | 0.4950 | -0.0319 |
| hydroflumethiazide | 0.3669 | 0.2025 | -0.1644 | 0.3350 | 0.3650 | +0.0300 |
| acetaminophen | 0.3531 | 0.2381 | -0.1150 | 0.3762 | 0.3844 | +0.0082 |
| primidone | 0.3550 | 0.2350 | -0.1200 | 0.3588 | 0.3556 | -0.0032 |
| CP-CLIP ↑ over CLIP (Clean) | +7.83% Accuracy | | | +7.72% F1-Score | | |
| CP-CLIP ↑ over CLIP (Disturbed) | +54.05% Accuracy | | | +50.55% F1-Score | | |

Table 21: Summary of Time Classification Performance: CP-CLIP vs CLIP under clean vs. disturbed prompts.

| Compound | CLIP | | | CP-CLIP | | |
|---|---|---|---|---|---|---|
| | Clean | Disturbed | Δ | Clean | Disturbed | Δ |
| ixabepilone | 0.9950 | 0.7550 | -0.2400 | 0.9975 | 0.9600 | -0.0375 |
| methoxsalen | 1.0000 | 0.6425 | -0.3575 | 1.0000 | 0.9575 | -0.0425 |
| sulfinpyrazone | 1.0000 | 0.6350 | -0.3650 | 1.0000 | 0.9175 | -0.0825 |
| triamterene | 1.0000 | 0.7100 | -0.2900 | 1.0000 | 0.8850 | -0.1150 |
| miconazole | 1.0000 | 0.6625 | -0.3375 | 1.0000 | 0.9325 | -0.0675 |
| ceritinib | 0.9950 | 0.6650 | -0.3300 | 1.0000 | 0.9350 | -0.0650 |
| acetohexamide | 1.0000 | 0.7650 | -0.2350 | 1.0000 | 0.9350 | -0.0650 |
| CP-CLIP ↑ over CLIP (Clean) | +0.08% Accuracy | | | +0.08% F1-Score | | |
| CP-CLIP ↑ over CLIP (Disturbed) | +34.91% Accuracy | | | +35.23% F1-Score | | |

This setup isolates the model's reliance on different types of metadata and tests whether it can still make accurate predictions under misleading or noisy context. For the concentration classification task (Table 20), CP-CLIP achieves a +7.72% F1-score improvement over CLIP under clean prompts, and a +50.55% F1-score gain under disturbed prompts. Similarly, for the time classification task (Table 21), CP-CLIP maintains stable performance with only marginal degradation under disturbed prompts, achieving a +35.23% F1-score improvement over CLIP. The detailed scores for each category are available in Table 18 and Table 19. In contrast, CLIP exhibits substantial performance drops in disturbed scenarios, suggesting that it is more susceptible to spurious correlations in the metadata.

This results confirm that CP-CLIP is not learning shortcuts based on metadata correlations but is instead capturing robust multimodal associations between visual morphology and semantic input prompts. The counterfactual experiment and contextual masking experiments effectively disentangles various sources of information, demonstrating that CP-CLIP can generalize to various classification tasks even under adversarial or misleading contextual conditions.

## T    RETRIEVAL PERFORMANCE BENCHMARKS

In addition to the classification accuracy metrics presented in Table 2 and Table 3, which are used for benchmarking in-distribution performance and out-of-distribution performance across various conditions (e.g., cell line, channel, and compound), we further report retrieval-based metrics to evaluate the effectiveness of contrastive learning models.

Specifically, we use Recall@K and Mean Reciprocal Rank (MRR) on a held-out validation set to assess both text-to-image (T→I) and image-to-text (I→T) retrieval performance. These metrics provide a complementary perspective on model alignment quality between visual and textual modalities, particularly in settings where ranking-based retrieval is desirable.

Table summarizes the retrieval performance of several contrastive models, including CLIP, SigLIP, and variants of our proposed CP-CLIP, under both text-to-image and image-to-text settings (Table 23). To provide a more complete picture, we also include the retrieval performance of CLOOME, shown in Table 22. While CLOOME is not a general-purpose contrastive model, we report its performance on molecule–image and image–molecule retrieval tasks for completeness. Notably, CLOOME's retrieval is limited to molecular inputs only, and is not applicable for cell-line or other biological metadata queries.

Table 22: Performance of CLOOME on Molecule–Image and Image–Molecule Retrieval Tasks.

| Model | Molecule-Image | | | | | | Image-Molecule | | | | | |
|---|---|---|---|---|---|---|---|---|---|---|---|---|
| | R@1 | R@5 | R@10 | R@20 | R@50 | MRR | R@1 | R@5 | R@10 | R@20 | R@50 | MRR |
| CLOOME | 95.58 | 99.94 | 99.94 | 99.94 | 99.94 | 0.9754 | 59.86 | 59.98 | 60.52 | 65.59 | 71.89 | 0.6063 |

Table 23: Context-to-Image (T-I) and Image-to-Context (I-T) retrieval performance using Recall@K.

| Model | Context-Image | | | | | | Image-Context | | | | | |
|---|---|---|---|---|---|---|---|---|---|---|---|---|
| | R@1 | R@5 | R@10 | R@20 | R@50 | MRR | R@1 | R@5 | R@10 | R@20 | R@50 | MRR |
| CLIP ViT-B/16 | 66.8 | 90.80 | 95.40 | 97.89 | 99.38 | 0.7719 | 58.55 | 80.67 | 86.54 | 91.71 | 96.85 | 0.6820 |
| SigLIP-ViT-B/16 | 43.85 | 67.38 | 77.44 | 85.75 | 93.19 | 0.5457 | 38.65 | 56.15 | 63.86 | 71.56 | 81.92 | 0.4700 |
| SigLIP-ViT-B/16 (D) | 25.93 | 52.21 | 61.37 | 71.45 | 85.38 | 0.3842 | 20.71 | 39.87 | 48.83 | 59.65 | 74.33 | 0.3015 |
| CP-CLIP ViT-B/16 (fp) | 72.97 | 93.96 | 97.47 | 99.10 | 99.86 | 0.8213 | 64.20 | 86.55 | 91.99 | 95.73 | 98.75 | 0.7385 |
| CP-CLIP ViT-B/16 (D) | 77.09 | 94.69 | 97.87 | 99.21 | 99.74 | 0.8479 | 68.92 | 87.77 | 92.14 | 95.56 | 98.55 | 0.7716 |
| CP-CLIP ViT-L/16 (D) | 73.83 | 92.93 | 96.44 | 98.49 | 99.61 | 0.8215 | 64.77 | 85.02 | 90.13 | 93.83 | 97.52 | 0.7351 |

