# OpenReview forum: "CP-Agent: Context‑Aware Multimodal Reasoning for Cellular Morphological Profiling under Chemical Perturbations"
_ICLR.cc/2026/Conference — ICLR 2026 Poster_

### Official Review · Reviewer_3gDr · 2025-10-27

**Soundness:** 3
**Presentation:** 3
**Contribution:** 3
**Rating:** 6
**Confidence:** 3

**Summary:**

This paper introduces CP-Agent, a multimodal system for analyzing Cell Painting drug perturbation experiments. The core technical idea is inserting learned embeddings for compound, dose, and time as placeholder tokens into the text encoder to align images with experimental context. The system achieves F1=0.896 on compound classification using 1.9M training pairs across three datasets which were normalized and fused in a scalable way. CP-Clip is turned into CP-Agent via subtool definitions, scaffolding and integration with GPT-5 to generate reports.

**Strengths:**

* Clear reframing that experimental context (compound, concentration, time) is signal and should be fused into the text branch via learned token projections.
* Principled multi-dataset curation with MoA harmonization. This is scalable and onboarding additional datasets should be cheap
* The dataset integration and cross-normalization seems sensible and principled and onboarding new Cell Painting datasets should be cheap.
* Clarity: I found the paper reasonably easy to read.

**Weaknesses:**

* “Agentic” framing: The system is a single-pass orchestrated pipeline with predefined tool calls, there is little evidence of learned action selection or closed-loop planning beyond routing, so the agentic claim feels a little overstated.
* I found the MLLM comparisons somewhat unfair and uninformative. CP-CLIP receives >1M domain-specific training pairs while GPT-5/Gemini/Claude receive (presumably) zero Cell Painting training and only minimal 2-stage prompting. MLLMs with extensive prompt engineering or few-shot learning would be a better comparison
* Weak expert evaluation. No inter-rater reliability metrics or comparison to baseline methods (human-written reports, template-based summaries) are provided. This makes the high scores uninterpretable without reference points.

**Questions:**

* For the expert evaluation, what is inter-rater reliability? Are differences between models statistically significant? How do the reports compare to human-written reports or simpler template-based baselines? Without these comparisons, the scores lack context.
* Can you clarify what makes this system "agentic" vs being a fixed pipeline?
* Can you provide image-free baselines using only molecular fingerprints and metadata (no microscopy images) for the classification tasks? This would quantify how much the images actually contribute to performance.

---

> ### Author Response · Authors · 2025-11-21
> **Response to Weakness 1 & Question 2**
>
> We appreciate the reviewer’s careful distinction between an “agentic” system in the stronger, RL-style sense and a single-pass orchestrated pipeline. Our CP-Agent does not implement such RL-style planning, and we do not claim this level of autonomy. In this work, we use the term *“agentic”* in the broader sense that has become common in recent LLM-agent work, where *“agentic AI”* denotes tool-using, memory-augmented LLM workflows. For example, a recent survey on Agentic AI in Healthcare [1] explicitly distinguishes different levels of autonomy (e.g., *Procedural Autonomy*, *Human-in-the-Loop*, *Highly Autonomous*). Within that framing, CP-Agent is best viewed as a **procedurally autonomous agent**:
>
> 1. It uses a general-purpose MLLM as a policy/controller over observations derived from images, retrieved experimental context, and tool outputs to prioritize features, select appropriate statistical summaries, and form mechanism-level hypotheses and follow-up suggestions.
>
> 2. It leverages CP-CLIP plus a curated knowledge base to load experimental *“memory”*, so that new drugs or cell lines can be analyzed without hand-written rules.
>
> 3. It relies on learned behavior in the pretrained LLM rather than fixed if–else scripts to decide, for example, which morphological features to focus on and how to interpret observed distribution shifts.
>
> We appreciate the opportunity to clarify this and will make our intended meaning explicit in the revision.
>
>
> #### Reference
>
> [1] Xu G, Li X, Chen Y, et al. *A Comprehensive Survey of Agentic AI in Healthcare*, 2025.

---

> ### Author Response · Authors · 2025-11-21
> **Response to Weakness 2**
>
> Thank you very much for raising this important point. We agree that the comparison between a domain-specialized model and general-purpose MLLMs needs to be framed carefully.
>
> Our goal with the current experiments was to answer a specific question: how far can off-the-shelf MLLMs go under a lightweight, reproducible adaptation strategy, rather than to claim that MLLMs are fundamentally inferior if heavily engineered. Because there is, to our knowledge, no established benchmark protocol for Cell Painting on MLLMs, we followed recent biological LLM works [2][3] that use a two-stage strategy (background-knowledge curation + task-specific instruction prompts) as a *“reasonable, standardized”* baseline rather than an aggressively tuned upper bound.
>
> That said, we agree with the reviewer that stronger prompting strategies (e.g., few-shot exemplars) are valuable and would make the comparison more informative. In the revised manuscript, we clarify this motivation and protocol explicitly in **Section 3.1**. In addition, to make the comparison more conservative, we now also report few-shot results where each MLLM is given a small visual memory bank (two labeled exemplars per class) before answering the same tasks. The detailed setup and results are provided in **Appendix M**, and the key quantitative outcomes are summarized in the table below. Across both zero-shot and few-shot settings, **CP-Agent** continues to outperform general-purpose MLLMs on Cell Painting tasks, supporting our main conclusions.
>
> ---
>
> #### Table: Few-shot model performance on classification tasks
>
> | Model          | Cell line       | Channel         | Flindokalner | Racecadotril | AZM-475271 | Misoprostol | Trazodone | Orantinib | Rufinamide | Lumiracoxib | BIRB-796 | Methoxsalen | Macro-avg |
> |--------------------|---------------------|----------------------|------------------|------------------|----------------|------------------|---------------|----------------|------------------|------------------|---------------|------------------|---------------|
> | Grok-4          | 0.515 (+0.067)      | 0.260 (+0.032)       | 0.224            | 0.184            | 0.000          | 0.000            | 0.390         | 0.176          | 0.034            | 0.000            | 0.000         | 0.000            | 0.101         |
> | GPT-5           | 0.440 (+0.063)      | 0.510 (+0.071)       | 0.000            | 0.000            | 0.066          | 0.000            | 0.115         | 0.000          | 0.079            | 0.000            | 0.088         | 0.000            | 0.035         |
> | Claude-4-Sonnet | 0.520 (+0.070)      | 0.225 (+0.027)       | 0.000            | 0.000            | 0.000          | 0.055            | 0.000         | 0.000          | 0.000            | 0.000            | 0.210         | 0.000            | 0.026         |
> | Gemini-2.5-Pro  | 0.600 (+0.074)      | 0.730 (+0.102)       | 0.000            | 0.000            | 0.000          | 0.000            | 0.000         | 0.000          | 0.000            | 0.160            | 0.074         | 0.000            | 0.023         |
>
> ---
>
> Note: Values in parentheses indicate the performance improvement compared to the zero-shot setting.
>
> ---
>
>
> #### References
>
> [2] Wang H, He Y, Coelho P. P., et al. *SpatialAgent: An autonomous AI agent for spatial biology*. bioRxiv, 2025: 2025.04.03.646459.
>
> [3] Yiyao L., Vakharia N., Liang W., et al. *OmicsNavigator: an LLM-driven multi-agent system for autonomous zero-shot biological analysis in spatial omics*. bioRxiv, 2025: 2025.07.21.665821.

---

> ### Author Response · Authors · 2025-11-21
> **Response to Weakness 3 & Question 1**
>
> We sincerely thank the reviewer for raising this important concern regarding the expert evaluation. We fully agree that comparisons to human-written narratives and to simple templated summaries would be valuable. However, constructing such baselines in a fair and reproducible manner would require collecting and releasing a new curated dataset of expert-written reports for the same experiments, which is beyond the scope of the current work.
>
> As part of our ongoing effort, we will make the **CP-Agent pipeline fully accessible to enable the community** to: 1. Reproduce and extend our expert evaluation, and 2. Construct and compare against additional baselines, including human-written and templated methods. We view this as a first step toward a standardized benchmark for Cell Painting report generation, and we hope that our released tools will catalyze the creation of shared, curated datasets of expert-written reports on the same experiments.
>
> To assess the internal consistency and reliability of the current expert evaluation, we conducted a comprehensive inter-rater agreement analysis, reported in **Appendix Q.2, Table 12** and summarized in **Appendix Q.2, Table 13** (below), using metrics like Kendall’s W, standard deviation, exact agreement percentage, ±1 rating agreement, and mean absolute difference (MAD). The Kendall’s W ranged from 0.33 to 0.37 across models, indicating fair agreement. At the pairwise level, 73.3% of rating pairs were within ±1 point on the 0–7 scale. The mean absolute difference was 1.04, reflecting strong agreement.  These findings support the internal consistency and reliability of the expert evaluations.
>
> While this evaluation (**Appendix Q.1**) didn't show statistically significant differences between LLMs, we observed consistent trends across raters: GPT-5 achieved slightly higher average scores across most dimensions, but the differences were relatively small. This suggests that all evaluated LLMs were capable of producing informative, relevant, and biologically meaningful reports under our CP-Agent pipeline.
>
>
> #### Table: Summary of Inter-rater Evaluations Across LLMs
>
> | Model         | Kendall_W | Score_Avg | Score_Std | Near_Agreement % |
> |--------------------|---------------|---------------|---------------|-----------------------|
> | Claude-Sonnet-4    | 0.33          | 5.45          | 0.94          | 72.65                |
> | Gemini-2.5-Pro     | 0.36          | 5.53          | 0.89          | 75.07                |
> | GPT-5              | 0.37          | 5.59          | 0.91          | 72.48                |
> | Grok-4             | 0.37          | 5.51          | 0.93          | 73.09                |
>
>
> To facilitate future work, we will release all pipeline codes, LLM prompts (used for CP-Agent), and model checkpoints (e.g., CP-CLIP and the fine-tuned channel-wise VISTA2d). We hope these open resources will empower more experts to easily try and extend our evaluation, and contribute to the development of standardized benchmarking in this area.

---

> ### Author Response · Authors · 2025-11-21
> **Response to Question 3 (1/2)**
>
> We thank the reviewer for this insightful suggestion. We would like to clarify that our classification tasks are not performed via a supervised classifier. For example, for the cell line classification task, we formulate prompts in the form “The cell line is { } ..., ...” and replace the placeholder with candidate cell lines such as *MCF7*, *A549*, and others from the database. The model then selects the most likely label based on similarity scores. The same process applies for other classification tasks. The predictions are made via **similarity-based retrieval** between microscopy images and textual metadata descriptions in a shared embedding space, without using an explicit classification head.
>
> In our work, the term **"metadata"** refers to a line of natural language paragraph about all experimental conditions, which include: molecular fingerprints or descriptors, encoded dosage and treatment duration, other contextual factors such as cell line, fluorescence channels, cell culture conditions, and imaging settings.
>
> Unlike general-purpose VLMs that process richly descriptive text, multiple compounds may share similar prompt fields. This redundancy is exemplified in **Table 1**, which shows that under the same cell line, channel, time, and other contextual conditions, there can still be dozens to hundreds of distinct compounds. If the reviewer is concerned that the model might rely on textual co-occurrence or memorization to retrieve the correct label during classification. As a partial step in this direction, we conducted experiments in which we selectively masked key textual fields to observe their impact on retrieval performance (**Appendix S, Tables 17–21**).
>
> #### Table: Retrieval performance before and after masking specific fields
>
>
> | Experiment | Modality        | Recall@1 Δ Absolute |
> |------------|------------------|-------------|
> | Mask Compound + MOA | T-I | -95.20      |
> |            | I-T    | -95.40      |
> | Mask Concentration  | T-I   | -41.70      |
> |            | I-T   | -43.50      |
> | Mask Time            | T-I | -5.70       |
> |            | I-T     | -4.05          |
>
> ---
>
> The results show that masking the compound information results in a catastrophic performance drop:Text-to-image R@1 drops from 98.70 to 3.50 and MRR drops by nearly 90%. In contrast, masking concentration or time results in moderate performance degradation (e.g., R@1 drops to 3.50 when masking compound, and only to 93.00 when masking time). This pattern suggests that the model encodes and leverages different fields of the metadata (e.g., compound identity, concentration) independently and meaningfully. Also demonstrates that the model does not merely memorize surface patterns in metadata text, but instead learns meaningful semantic correspondences between microscopy images and textual descriptions.
>
> We hope the provided experiments can address the reviewer’s concern. We remain open to further clarification and are willing to conduct additional experiments.

---

> > ### Author Response · Authors · 2025-11-23
> > **Response to Question 3 (2/2)**
> >
> > To further assess the contribution of microscopy images while avoiding spurious correlations between textual fields, we conducted an image-free experiment using only textual metadata and compound identity. Specifically, we constructed a text-only input that describes the experimental setup (e.g., cell line, medium, imaging conditions), while deliberately masking all compound-related information (compound identity and corresponding MOA). An example input is:
> >
> > > `"text": "Cell line is HUVEC; Cells cultured in EGM2 medium, maintained at 37°C with 5% CO₂; Image channel is Actin; The imaging objective is 10X; The concentration is <CONC_TOKEN>; The observation time is <TIME_TOKEN>; The perturbation compound is <mask>; The mechanism of action for this compound is <mask>."`
> >
> > To ensure a fair evaluation, the text dataset was deduplicated and then split into training and validation sets (10:1 ratio), such that no instance in the validation set overlaps with those in the training set. We trained a simple classifier (LayerNorm + MLP) on top of the text encoder to predict compound labels (out of 450 unique compounds), using a pre-defined index mapping.
> >
> > The results of this experiment are shown in the table below. Across both XGBoost and MLP classifier settings, and under both pure text (original CLIP's text encoder) and hybrid token injection modes (CP-CLIP's text encoder), the model performance remained near-random. For reference, random guessing would yield an expected accuracy of approximately **0.22%** (1/450). All observed accuracies fall at or below this level, indicating that masked textual metadata alone lacks sufficient information to predict compound identity.
> >
> > #### Table: Metadata–Compound Classification Accuracy
> >
> > | Model (Input Mode)          | Accuracy | Random Guess |
> > |---------------------------------|--------------|------------------|
> > | XGBoost (Pure Text)             | 0.050%       | 0.222%           |
> > | XGBoost (Hybrid Context)           | 0.110%       | 0.222%           |
> > | MLP Classifier (Pure Text)      | 0.368%       | 0.222%           |
> > | MLP Classifier (Hybrid Context)    | 0.343%       | 0.222%           |
> >
> > These findings indicate that the model also relies on meaningful visual signals in the microscopy images related to compound identity, rather than relying on metadata or experimental context leakage.

---

### Official Review · Reviewer_uvt6 · 2025-10-27

**Soundness:** 2
**Presentation:** 2
**Contribution:** 2
**Rating:** 2
**Confidence:** 5

**Summary:**

This paper proposes an agent-based framework comprising two main components: CP-CLIP, a contrastive learning model for chemical perturbation–image alignment, and CP-Agent, a agentic system built atop CP-CLIP.

**Strengths:**

The main novelty of this work lies in the introduction of an agentic framework specifically designed for Cell Painting images. While the underlying components (e.g., CLIP-based contrastive learning and vision encoders) are not new, applying an agent-based paradigm to organize multimodal reasoning, analysis, and interpretation within this biological context is conceptually original. This represents an interesting step toward structured and interpretable automation in cellular image understanding and drug discovery workflows.

**Weaknesses:**

While incorporating agentic system for Cell Painting is interesting, there are several weakness of the proposed work,


**Insufficient survey of related work in cross-modal contrastive learning for Cell Painting**
* The related work section lacks sufficient discussion of recent multimodal and contrastive learning methods for Cell Painting. While the paper introduces CP-CLIP as a novel contribution, several recent works within this context are not adequately discussed [1, 2, 3].
* In addition, the manuscript lacks quantitative comparisons with prior frameworks. Although CP-CLIP is presented as a key innovation, the evaluation primarily contrasts against generic CLIP variants and omits stronger baselines from recent multimodal or contrastive approaches. This makes it difficult to disentangle whether the reported improvements stem from the proposed context-aware token injection mechanism or from other architectural or training differences.

**Method**
* While interpretability is highlighted as a major advantage of CP-Agent, the paper lacks quantitative evaluation of this aspect (e.g., faithfulness, consistency, or attribution accuracy).
* The authors pretrained CP-CLIP on roughly 500 distinct compounds, which is small relative to the nearly two million image–context pairs. Since standard CLIP loss assumes each image–text pair is unique, simply aligning the same perturbation would potentially lead to overfitting or biased representations.

**Limited evaluation and non-standard evaluation metrics for cross-modal contrastive learning**
* The number of chemical perturbations explored in this work is relatively limited. Prior studies in cross-modal contrastive learning for Cell Painting [1,2,3] have leveraged datasets with over 10k compounds, whereas this study appears to utilize a smaller subset (approximately 500 compounds across 1.9M image–context pairs). This limitation constrains the generalizability of CP-CLIP and raises concerns regarding its robustness across broader chemical spaces.
* The evaluation of CP-CLIP relies primarily on cosine similarity scores--the optimization objective of CLIP loss itself--rather than standard retrieval metrics such as Recall@K. This metric choice makes the reported improvements less convincing, as higher cosine similarity naturally aligns with the training objective.
* The paper benchmarks CP-Agent primarily against general-purpose MLLMs (e.g., GPT-5, Gemini-2.5-Pro, Claude-4-Sonnet), which are not pretrained on Cell Painting data. A more appropriate baseline would involve fine-tuning a VLM on the same Cell Painting datasets for a fairer comparison.
* How classification is done for standard CLIP, e.g.,whether a separate classifier is trained or retrieval is done by ranking cosine similarity, is not clearly explained.

**Missing Ablation Studies and Component Analysis**
 * The agentic system (CP-Agent) includes several modules (e.g., CPContext, FeatRank, StatSynth, ReportGen), yet the manuscript provides no ablation to assess their individual contributions. Without such analysis, it remains unclear which components are necessary or most influential for final performance.

**Unclear dataset and evaluation setup**
* The description of training, validation, and test splits is ambiguous. It is unclear which datasets are used for each stage and how compounds are selected for evaluation. Given the paper’s focus on chemical perturbations, omitting comprehensive datasets such as [4] limits the scope.

**Presentation and clarity issues**
* Several figures (e.g., Figures 1–4) lack sufficient captions and details. For example, the use of a GPT-2 tokenizer in Figure 1 is not clear, what constitutes “raw text” and how it is tokenized remain unclear. Similarly, later figures provide only high-level summaries without specifying details.

[1] CLOOME: contrastive learning unlocks bioimaging databases for queries with chemical structures Sanchez-Fernandez et al Nature com 2023

[2] How Molecules Impact Cells: Unlocking Contrastive PhenoMolecular Retrieval Fradkin et al, NeurIPS 2024

[3] CellCLIP – Learning Perturbation Effects in Cell Painting via Text-Guided Contrastive Learning Lu et al NeurIPS 2025

[4] Cell Painting, a high-content image-based assay for morphological profiling using multiplexed fluorescent dyes, Bray et al Nature protocal 2016

**Questions:**

See above

---

> ### Author Response · Authors · 2025-11-21
> **Response to Insufficient survey of related work, Point 1**
>
> Thank you for highlighting these relevant works. Regarding [1] (Sanchez-Fernandez et al., 2023, CLOOME), we mentioned this work in the Introduction (third paragraph; cited as **Ref. 41**). We also apologize for the omission of the other two recently published works. We have now updated the **Section 1** (third paragraph in the revised version) and **Reference section** to include both CellCLIP and MolPhenix (cited as **Ref. 14**, **Ref. 33**), and clarified how our approach relates to them. Our CP-Agent system extends beyond CLOOME, CellCLIP and MolPhenix in several important ways:
>
> **1. Structured Contextual Conditioning:**
> While the previous works such as CLOOME primarily focus on mapping SMILES strings or learned molecular embeddings to Cell Painting profiles to support structure-aware retrieval and MOA inference. CP-CLIP employs a hybrid token-injection strategy to embed structured experimental context beyond compound identity, incorporating cell lines, log-indexed dosage, time points, imaging channels, and other metadata necessary for experiment replication. This design enables CP-CLIP to align Cell Painting images with a much richer biological context. In contrast to CellCLIP, which adds text-guided contrastive learning using perturbation descriptions as language inputs, CP-CLIP encodes both numerical and textual metadata directly into the shared embedding space. This unified representation is able to capture dose–response trajectories, temporal progression, and cell-line specificity, thereby supporting more precise phenotype-to-perturbation alignment, and also enhanced biological relevance.
>
> **2. Agentic Reasoning Capabilities:**
> CP-Agent integrates tool-augmented reasoning and task-adapted multimodal LLMs, enabling structured, interpretable outputs grounded in phenotypic descriptors and MOA mechanisms, which helps drug scientists rapidly analyze cellular states and iteratively design experiments based on informed hypotheses from the generated reports.
>
> **3. Flexible Adaptation to Diverse Experimental Conditions:**
> CP-Agent supports scalable learning across diverse assay conditions. The CP-CLIP architecture allows seamless adaptation to new cell lines, imaging setups, and staining protocols (e.g. different channels due to differences in microscopy configuration or dye) through continued fine-tuning, enabling broad generalization across studies and platforms in the future.

---

> ### Author Response · Authors · 2025-11-21
> **Response to Insufficient survey of related work, Point 2**
>
> Thank you for this important comment. We agree that a direct quantitative comparison with recent contrastive frameworks would be ideal in principle. We note that the three frameworks are optimized for different modeling objectives, making direct benchmarking less straightforward and potentially misleading. Specifically, they aim to learn representations from the molecular structure alone, without the other key experimental conditions that are also essential for modeling compound perturbations in real-world assay settings.
>
> CP-CLIP is explicitly designed for downstream phenotypic reasoning by encoding a hybrid set of contextual variables, including both numerical and text-based metadata (e.g., dose, time, cell line). This allows it to distinguish between perturbations with the same compound but different experimental settings, and to support prompt-based reasoning over these variables.
>
> These differences are summarized in the table below, which highlights the scope of experimental metadata encoded across each method:
>
> #### Table: Comparison of encoded experimental information across recent Cell Painting contrastive learning frameworks
>
> | Method     | Compound  | Concentration    | Cell Line | Time Point | Channel | Magnification | Culture Condition | Tokenized Context | Prompt-based Extraction |
> |----------------|----------------------------|-----------------------|---------------|----------------|-------------|--------------------|------------------------|------------------------|------------------------------|
> | CLOOME     | ✓ structure    | ✘  | ✘   | ✘  | ✘ | ✘   | ✘  | ✘  | ✘  |
> | MolPhenix  | ✓structure (GNN)  |✓(implicit)         | ✘ | ✘  | ✘  | ✘  | ✘ | ✘  | ✘  |
> | CellCLIP   | ✓ LLM-encoded text         | ✘  | ✘  | ✘ | ✓     | ✘  | ✘   | ✘                     | ✘                          |
> | CP-CLIP    | ✓ structure (descriptors)  | ✓                   | ✓            | ✓            | ✓        | ✓                 | ✓                   | ✓                     | ✓                           |
>
> Because CP-CLIP explicitly encodes all key contextual variables as structured text tokens during training, it inherently supports text-prompted querying and extraction of these variables without requiring additional supervision or task-specific adaptation. This stands in contrast to prior contrastive learning frameworks, where such information is only implicitly captured within the image embedding. As a result, recovering specific experimental attributes in those models typically requires training dedicated downstream probes (e.g., linear classifiers). In contrast, CP-CLIP aligns tokenized metadata with visual features in a shared embedding space, enabling zero-shot reasoning and retrieval of experimental context via natural language prompts. We note that benchmarking this capability would be infeasible and unfair for prior methods, as they were not designed to support explicit metadata conditioning or natural language-based querying.
>
> Despite these limitations, we conducted a quantitative benchmark of CLOOME on the Recall@K performance and drug classification task (the only task it supports given its limited context encoding). In the image→context (image→compound) retrieval, CP-CLIP outperforms CLOOME, achieving an R@1 of 68.92 versus 59.86. In the drug classification benchmark, CP-CLIP achieves a macro-average F1 score of 0.891, substantially outperforming CLOOME. We have updated this comparison in **Table 2**  and **Appendix T** in the revised manuscript. These results show that CP-CLIP leverages contextual metadata for richer semantic reasoning and also outperforms compound-only contrastive methods in phenotypic tasks.
>
>
> #### Table: Performance of CLOOME on Retrieval Tasks
>
> | Model | R@1 | R@5 | R@10 | R@20 | R@50 | MRR | R@1 | R@5 | R@10 | R@20 | R@50 | MRR |
> |-----------|---------|---------|----------|----------|----------|---------|---------|---------|----------|----------|----------|---------|
> |           | Molecule → Image | | | | | | Image → Molecule | | | | | |
> | CLOOME | 95.58  | 99.94   | 99.94    | 99.94    | 99.94    | 0.9754  | 59.86  | 59.98   | 60.52    | 65.59    | 71.89    | 0.6063  |
>
> ---
>
> #### Table: Performance of CP-CLIP on Retrieval Tasks
>
> | Model   | R@1 | R@5 | R@10 | R@20 | R@50 | MRR | R@1 | R@5 | R@10 | R@20 | R@50 | MRR |
> |-------------|---------|---------|----------|----------|----------|---------|---------|---------|----------|----------|----------|---------|
> |             | Context → Image | | | | | | Image → Context | | | | | |
> | CP-CLIP | 77.09  | 94.69   | 97.87    | 99.21    | 99.74    | 0.8479  | 68.92  | 87.77   | 92.14    | 95.56    | 98.55    | 0.7716  |

---

> > ### Author Response · Authors · 2025-11-21
> > **Response to Method, Point 1&2; Limited evaluation and non-standard evaluation metrics, Point 1**
> >
> > **Method Point 1**: We thank the reviewer for raising this important point. To quantitatively evaluate the reports generated from CP-Agent, we have conducted a structured expert survey, as described in **Section 3.5** (CP-Agent Reports) and detailed in **Appendix P**. Specifically, we invited 12 domain experts to assess reports generated by CP-Agent across 10 different screening cases, rating each report on 10 rubric-based dimensions related to language quality and biological reasoning quality (**Appendix P, Q**). As shown in **Figure 16**, all LLMs achieved average scores above 5.0 out of 7.0 across most dimensions, indicating that CP-Agent, guided by CP-CLIP, generates reports that are biologically meaningful, a capability that cannot be achieved by general MLLMs (upper part in **Table 2**).
> >
> > To further quantitatively evaluate the interpretability and consistency of CP-Agent, we focus on two LLM-powered modules:  (i) the *FeatRank* Agent, which ranks CellProfiler-extracted morphological features;  (ii) the *ReportGen* Agent, which generates natural language reports based on the ranked features and contextual metadata.
> >
> > To ensure determinism in the feature selection step, we fixed the temperature of the *FeatRank* Agent at 0.0. We conducted five experimental settings where the *ReportGen* Agent’s temperature varied from 0.0 to 1.0, while keeping the *FeatRank* Agent fixed. The details of experimental setting is added in **Appendix R**. In **Appendix R, Table 15**, the selected features were highly consistent across five runs, demonstrating robust feature selection. To evaluate corpus-level consistency of the reports generated by *ReportGen*, we performed a systematic evaluation using GPT-5 as a judgment model along three dimensions: thematic consistency, terminological consistency, and structural consistency. As shown in **Appendix R, Table 16**, the reports maintained stable scores (7.67–8.00 out of 10) across different temperature, suggesting that the generation process is robust and coherent to sampling variability. We have clarified these points in the revised **Section 3.5** and moved the new quantitative faithfulness/consistency analyses into **Appendix R** so that our interpretability claims are supported by both expert evaluation and explicit quantitative measures.
> >
> > **Method Point 2 & Limited evaluation and non-standard evaluation metrics, Point 1:** We thank the reviewer for raising this point and would like to provide further clarification on the number of unique experimental conditions in our dataset. While the training set includes ~500 distinct compounds, this does not equate to only 500 unique experimental scenarios. In fact, each compound is tested under diverse conditions, including variations in dosage, treatment time, cell line, imaging magnifications, and other covariates. These combinations result in 20,448 unique experimental contexts. Each of these contexts is associated with a distinct textual description and reflects subtle yet meaningful differences in cellular morphology. On average, each experimental context is associated with only ~50–100 images, depending on repeat frequency and plate design.
> >
> > Furthermore, to mitigate class imbalance and prevent overfitting, we applied data augmentation techniques (e.g., random crops, flips...) and a more balanced sampling strategy during training. The diversity and granularity of the experimental conditions offer a richer alignment signal than only providing the perturbation compound information.
> >
> > Moreover, the selection of these compounds was deliberate. We focused on compounds with well-characterized mechanisms of action (MOA) that are publicly accessible through curated databases such as ChEMBL, for which structured chemical representations (e.g., SMILES strings) and corresponding MOA annotations are available. Specifically, to retrieve MOA information, we used the **ChEMBL web resource client** in combination with **RDKit**. Specifically, we validated SMILES strings using **Chem.MolFromSmiles**, performed exact structure matching via **similarity.filter(smiles, similarity=100)**, and queried **mechanism.filter()** to obtain associated targets and mechanisms of action (**Section 2.1**).
> >
> > This enables the extraction of reliable descriptors and enhances interpretability and traceability of the learned representations. Such information-rich design would be helpful when researchers apply the tool to new drugs, as it can offer clear semantic, chemical, and experimental grounding for downstream analysis or hypothesis generation. We have also uploaded the script to our GitHub repository to support reproducibility.

---

> > > ### Author Response · Authors · 2025-11-21
> > > **Response to Limited evaluation and non-standard evaluation metrics, Point 2**
> > >
> > > We appreciate the reviewer’s concern regarding the alignment metric. To address this point, we have conducted additional evaluations on a held-out validation dataset comprising 9,395 image-text pairs, using more standard retrieval metrics: Recall@K (R@1, R@5, R@10, etc.) and Mean Reciprocal Rank (MRR). The results are summarized in the table below:
> > >
> > > #### Table: Context-to-Image and Image-to-Context retrieval performance using Recall@K
> > >
> > > | Model                 | R@1 | R@5 | R@10 | R@20 | R@50 | MRR | R@1 | R@5 | R@10 | R@20 | R@50 | MRR |
> > > |---------------------------|--------:|--------:|---------:|---------:|---------:|--------:|--------:|--------:|---------:|---------:|---------:|--------:|
> > > |                           | Context-Image                      |         |          |          |          |         | Image-Context                     |         |          |          |          |         |
> > > | CLIP ViT-B/16             | 66.80   | 90.80   | 95.40    | 97.89    | 99.38    | 0.7719  | 58.55   | 80.67   | 86.54    | 91.71    | 96.85    | 0.6820  |
> > > | SigLIP-ViT-B/16           | 43.85   | 67.38   | 77.44    | 85.75    | 93.19    | 0.5457  | 38.65   | 56.15   | 63.86    | 71.56    | 81.92    | 0.4700  |
> > > | SigLIP-ViT-B/16 (D)       | 25.93   | 52.21   | 61.37    | 71.45    | 85.38    | 0.3842  | 20.71   | 39.87   | 48.83    | 59.65    | 74.33    | 0.3015  |
> > > | CP-CLIP ViT-B/16 (fp)     | 72.97   | 93.96   | 97.47    | 99.10    | 99.86    | 0.8213  | 64.20   | 86.55   | 91.99    | 95.73    | 98.75    | 0.7385  |
> > > | CP-CLIP ViT-B/16 (D)      | 77.09   | 94.69   | 97.87    | 99.21    | 99.74    | 0.8479  | 68.92   | 87.77   | 92.14    | 95.56    | 98.55    | 0.7716  |
> > > | CP-CLIP ViT-L/16 (D)      | 73.83   | 92.93   | 96.44    | 98.49    | 99.61    | 0.8215  | 64.77   | 85.02   | 90.13    | 93.83    | 97.52    | 0.7351  |
> > >
> > > These findings confirm that the performance gains of CP-CLIP are not limited to cosine similarity scores but also show improvements in standard Recall@K metrics. We have added these results to **Appendix T** in the revised manuscript.

---

> > > > ### Author Response · Authors · 2025-11-21
> > > > **Response to Limited evaluation and non-standard evaluation metrics, Point 3**
> > > >
> > > > We thank the reviewer for this insightful comment and agree that, in principle, a domain-matched VLM baseline trained on the same Cell Painting datasets would provide the fairest comparison. However, to the best of our knowledge, there is currently no off-the-shelf VLM that has been pretrained specifically on Cell Painting data and can be directly used as such a baseline.
> > > >
> > > > We would like to clarify that **CP-CLIP** itself is a Cell Painting–specific VLM, and that **CP-Agent** is built on top of CP-CLIP by adding agentic orchestration and multi-step reasoning. Our comparison to general-purpose MLLMs (**GPT-5**, **Gemini-2.5-Pro**, **Claude-4-Sonnet**) is therefore intended to answer a complementary question: given access to domain data and tools, can a specialized Cell Painting VLM + agent outperform state-of-the-art, general-purpose MLLMs that have not been pretrained on Cell Painting? This evaluation setup is consistent with recent AI for Healthcare/Biomedicine works that use open-ended VQA-style tasks to probe the visual understanding capabilities of general-purpose MLLMs in biomedical imaging settings [1,2].
> > > >
> > > > At the current stage, we believe our experiments already demonstrated that a domain-specific VLM (**CP-CLIP**) combined with an agentic framework (**CP-Agent**) provides clear advantages over general-purpose MLLMs on Cell Painting tasks, even without finetuning the latter on Cell Painting data.
> > > >
> > > > #### References
> > > >
> > > > [1] Lozano A, Nirschl J, Burgess J, et al. *μ-bench: A vision-language benchmark for microscopy understanding*. arXiv preprint arXiv:2407.01791, 2024.
> > > >
> > > > [2] Burgess J, Nirschl J J, Bravo-Sánchez L, et al. *Microvqa: A multimodal reasoning benchmark for microscopy-based scientific research*, 2025. [arXiv:2503.13399](https://arxiv.org/abs/2503.13399)

---

> ### Author Response · Authors · 2025-11-21
> **Response to Limited evaluation and non-standard evaluation metrics, Point 4**
>
> Thanks for raising this important point. In our CLIP-style architecture, classification is performed via retrieval-based inference, consistent with the standard CLIP paradigm. Specifically, during inference, the model computes cosine similarity between the image representation and a set of candidate textual prompts, and ranks them accordingly, with no additional classifier or task-specific head trained. For example, for the cell line classification task, we formulate prompts in the form:  *“The cell line is { }...., ....”*  and replace the placeholder with candidate cell lines such as *MCF7*, *A549*, and others from the database. The model then selects the most likely label based on similarity scores.
>
> Although the input context includes both textual and numerical variables, these are integrated into the token sequence alongside natural language tokens. This unified token sequence is then processed by the shared text encoder, enabling the model to align image features with rich context. Therefore, CP-CLIP supports **zero-shot classification** and **prompt-based querying**, where downstream tasks can be formulated as retrieval problems over a set of textual hypotheses.
>
> This clarification has been added to **Section 3.2 (Lines 317–321)** of the revised manuscript.

---

> ### Author Response · Authors · 2025-11-21
> **Response to Missing Ablation Studies of Modules in CP-Agent**
>
> We thank the reviewer for raising this important point. Our modules are designed to work together in an **end-to-end manner**, with each providing a distinct and non-redundant capability. Completely removing several of these modules would make it impossible to run the system end-to-end.
>
> For example, there will be no way to translate from raw images to text if *ReportGen* is removed. Similarly, without *StatSynth*, we won't be able to provide any cell-level statistics. Removing *FeatRank* or *StatSynth* would make it infeasible to process the high-dimensional CellProfiler features effectively. A single image pair (control vs perturbation) can yield over 10,000 features, which will exceed token limits and introduce noise when fed directly to LLMs. *FeatRank* selects context-relevant features, while *StatSynth* aggregates per-cell values into compact statistical summaries. Both are essential to produce reliable and interpretable inputs for final report generation (as described in **Section 2.5**).
>
> Nonetheless, we recognize that the reviewer may not only be asking for “remove-this-module” experiments, but for evidence about which components are most influential for performance and interpretability. We already provided a strong implicit ablation for *CPContext* in the main text. As shown in **Table 2** (top half), removing this module and relying solely on LLMs (e.g., Grok, GPT, Claude, Gemini) results in performance close to random guessing. So *CPContext*, the image-to-text grounding module based on CP-CLIP, is essential for retrieving context-aware descriptions of cell states.
>
> We have made this connection explicit in the revision: **Section 3.2** now states that these results constitute a “no-CPContext” baseline (**Table 2**) and demonstrate that CPContext is necessary for any meaningful interpretation of Cell Painting images by an LLM.
>
> In summary, **CP-Agent** is inherently modular, and several modules are functionally indispensable for end-to-end operation. However, we agree with the reviewer that it is important to quantify their contributions. We already provide a strong implicit ablation for *CPContext* via “no-vision” MLLM baselines (**Table 2**), and in the revision we will add targeted ablations for *FeatRank*, to make the influence of each component more explicit.

---

> ### Author Response · Authors · 2025-11-21
> **Response to Unclear dataset and evaluation setup**
>
> 1).  Thank you for pointing this out. We agree that the description of data splits should be made clearer, and we have revised **Section 2.1** accordingly. Our training, validation, and test sets are constructed by combining data from the three publicly available Cell Painting datasets (JUMP, RxRx, and BBBC021), which differ in imaging conditions as well as in compound perturbation designs (**Section 2.1, Table 1**). This unified setup facilitates cross-dataset generalization under diverse imaging and perturbation conditions. The training set contains 1,846,436 image-text pairs. The validation set includes 9,395 pairs. The compounds listed in **Table 3** are used for evaluating zero-shot unseen drug generalization. These zero-shot test compounds were randomly selected and removed from the entire dataset before training to ensure they were completely unseen during training. Each of these compounds represents a distinct MOA to ensure diversity. The source of each compound is provided in the table below：
>
> #### Table: Compounds used in zero-shot evaluation
>
> | **CPJUMP**                                | **RxRx3**                                 | **BBBC021**       |
> |-------------------------------------------|-------------------------------------------|-------------------|
> | Buparlisib, Dexamethasone, Nimodipine     | Regorafenib, Sacubitril, Nilotinib        | AZ258, MG-132     |
>
> ----
>
> 2).  Regarding the omission of dataset [4] *(Bray et al., Nature Protocols 2016)*:
> We have carefully reviewed the referenced protocol paper. While it is an important and foundational work in the Cell Painting community, its primary purpose is to demonstrate the experimental pipeline for morphological profiling using **RNAi perturbations in U2OS cells**.
>
> RNAi is a technique that **silences gene expression by degrading target mRNA**, enabling gene-specific functional studies. In contrast, **our work focuses on small molecular perturbations**, with an emphasis on generalization across diverse drug experimental settings. Given this difference in perturbation modality and study scope, we believe reference [4] is not directly applicable to our current work.

---

> ### Author Response · Authors · 2025-11-21
> **Response to Presentation and Clarity Issues**
>
> We sincerely thank the reviewer for pointing this out. Owing to page constraints, we had to shorten the figure captions in the first submission. We apologize for the lack of clarity and any confusion this may have caused. We have now revised the manuscript to provide full and detailed captions for all figures, which we believe will improve clarity.
>
> Specifically for **Figure 1**, we have clarified the role of the tokenizer and the definition of *"raw text"* below in the revised manuscript (**lines 169–178**):
>
> > *"The “raw text” refers to structured experimental metadata such as cell line, culture medium, imaging parameters, compound identity, dosage, time, and other cultural information if available. These contextual descriptions are first composed into a natural language-style sentence and tokenized into input IDs using the standard GPT-2 tokenizer. ... The special placeholder tokens are directly inserted into the text sequence and registered into the tokenizer’s vocabulary. During tokenization, they are automatically recognized as atomic units and their positions are preserved without being split or altered."*

---

> ### Comment · Reviewer_uvt6 · 2025-11-28
>
> Thank you to the authors for addressing several of my earlier concerns. I appreciate the revisions and additional experiments. A few important questions remain:
>
> **Evaluation for CP-CLIP**
>
> * While I understand that CP-CLIP is intended primarily as a metadata retriever for CP-agent, the experimental setup and results differ substantially from previously reported findings [1,2,3], and further clarification would strengthen the paper. Given that the held-out set consists of unseen compounds, it is unexpected that even fine-tuned CLIP and CLOOME achieve nearly 100% retrieval accuracy in some settings (Table 23 & updated table). More details on how these compounds were selected would be needed, particularly why only 9 out of 476 compounds (<2%) were included. This evaluation set is considerably smaller than prior work (e.g., 2.1k unseen compounds in [1,3], 6.5k in [2]).
> * To improve comparability and demonstrate that the proposed method generalizes beyond the current evaluation setting, I encourage the authors to evaluate retrieval performance on Bray et al. [4], which includes ~10k small-molecule perturbations and is widely used in related work. (Prior studies report Recall@1 of only <10% in retrieval with unseen compounds [1,3].)
> * Finally, the advantage of incorporating extensive metadata (e.g., time, concentration, etc) remains unclear, as downstream tasks such as classification and report generation appear to depend strongly on drug name. It is not obvious whether performance would differ if only compound names were provided.
>
> **Evaluation for CP-Agent**
>
> * Comparing against a vanilla VLM is not particularly informative, as it has no exposure to Cell Painting data and, unsurprisingly, achieves near-zero accuracy. These baselines, therefore, do not meaningfully evaluate the proposed method. Since the primary comparison in the paper is between CP-agent and MLLMs on classification, a more suitable and informative baseline would be a LoRA-fine-tuned VLM trained on Cell Painting images and metadata, with drug name prediction as the target. Such a baseline would provide a fairer assessment of the proposed method’s contribution and better isolate the value added by CP-Agent.
>
>
> ----------
> * [1] CLOOME: contrastive learning unlocks bioimaging databases for queries with chemical structures Sanchez-Fernandez et al Nature com 2023
> * [2] How Molecules Impact Cells: Unlocking Contrastive PhenoMolecular Retrieval Fradkin et al, NeurIPS 2024
> * [3] CellCLIP – Learning Perturbation Effects in Cell Painting via Text-Guided Contrastive Learning Lu et al NeurIPS 2025
> * [4] A dataset of images and morphological profiles of 30,000 small-molecule treatments using the Cell Painting assay

---

> ### Author Response · Authors · 2025-12-03
> **Response to Evaluation for CP-CLIP, Point 1 – Round 2**
>
> We thank the reviewer for the thoughtful comments and the opportunity to clarify our experimental design and evaluation setup. As introduced in the manuscript, our primary goal is not to perform compound-only retrieval/classification as in prior works like CLOOME[1], MolPhenix[2], and CellCLIP[3]; rather, we aim to develop a unified, interpretable, context-aware metadata retriever that aligns Cell Painting images with structured experimental annotations (e.g., compound, cell line, fluorescence channel, and other context fields). This module, CP-CLIP, serves as the perception backbone of our broader CP-Agent framework, which supports biological reasoning and agentic decision-making for drug screening.
>
> **Our design differs substantially from prior work in:**
> **Architecture:** CP-CLIP aligns images with structured experimental metadata via token-injected multimodal contrastive learning, rather than using chemical fingerprints[1] or free-text perturbation descriptions[3].
> **Task formulation:** We treat metadata retrieval as a structured, multi-field classification problem (compound, cell line, channel), not only as molecule-image retrieval.
> **Data usage:** While previous studies focus on molecule-level retrieval, our dataset included over ~1.9 million image–metadata pairs covering diverse experimental conditions. Despite lower compound diversity, the overall scale and diverse contents make it well-suited for CLIP-style contrastive learning:
>
> | **Method**     | **Data Scale**                      |
> |----------------|-------------------------------------|
> | CLOOME[1]      | ~759k image-compound pairs          |
> | MolPhenix[2]   | ~1.3M image-compound pairs          |
> | CellCLIP[3]    | ~284k image-text pairs              |
> | Ours           | ~1.9M image–metadata pairs          |
>
>
> Therefore, direct comparison of compound retrieval scores between the four methods is unfair, as the task scope, data design, and evaluation objectives differ.
>
> The reviewer raised concerns about high accuracy in some Tables and the small number of compounds in the evaluation. We clarify as follows:
> In **Table 2**, we not only reported compound classification F1 scores, but also cell line and fluorescence channel classification metrics, where the numbers of classes were small (4 for cell lines, 7 for channels), and the visual phenotypes are highly distinctive. These classification tasks are based on similarity scores, so high accuracy is expected and consistent across models. This design is also intentional, as performing open-set retrieval over ~500 compounds is infeasible in a fair LLM benchmark setup, where each compound query would require not only a well-defined candidate scope but also curated background knowledge, e.g., `nuclear_dominant`, `filamentous` (detailed in **Section M**). This ensures that the LLM has access to sufficient context to reason meaningfully. Furthermore, we also conducted few-shot querying experiments (**Section M.4 Table 12**) to provide a more comprehensive evaluation under realistic LLM usage scenarios. Therefore, we evaluate LLMs and CP-CLIP under controlled multi-class settings to ensure fair comparison. Metrics reported are F1 scores, not similarity-based ranking scores. We also provide retrieval-style similarity scores in **Appendix I Table 7** under the same compound subset settings as a reference.
>
> Regarding **Table 23**, we would like to clarify that each unique context (i.e., metadata entry) is associated with 50–100 images spanning various concentrations, time points and other conditions. The many-to-one structure between context and images makes the context-to-image retrieval task easier, leading to high Recall@50. However, the Recall@1 remains below 80%, and the image-to-context direction shows relatively lower performance. We believe this setup realistically reflects the many-to-one nature of dataset, and showing the model can resolve morphological variability across replicates. For these results, We have open-sourced all code to ensure transparency and reproducibility.

---

> ### Author Response · Authors · 2025-12-03
> **Response to Evaluation for CP-CLIP, Point 2 – Round 2**
>
> We thank the reviewer for this valuable suggestion. We agree that evaluating on widely-used benchmarks such as Bray et al. [1] is useful for assessing generalization. However, after careful investigation, we found that the Bray et al. dataset, while large in total perturbations (~30,000), includes only 2,500 known bioactive compounds, primarily derived from BBBC022-v1 [2]. Related information in Bray et al. [1] can be found in the corresponding data website:
> https://www.cellimagelibrary.org/pages/project_20269:
>
> > "......This image-based assay provides an unbiased approach to characterize compound- and disease-associated cell states to support future probe discovery. Using the cell-painting assay, the Broad Institute has assembled a reference dataset of profiles for U2OS osteosarcoma cells treated with ~30,000 compounds. The compound collection includes DOS-derived compounds (20,000), as well as chemically diverse MLI compounds with biologically diverse performance identified through analysis of PubChem (10,000), and **known bioactive compounds to serve as landmarks (2,500)**......"
>
> In our work, as explained in our response to **Limited evaluation, point 1**, we only included compounds with well-defined and traceable mechanisms of action (MOAs). This is essential so that all compounds included in the evaluation are supported by well-defined mechanistic annotations, thereby enabling evidence-based downstream analysis and hypothesis generation. By applying the same procedure (extract MOA via ChEMBL with RDKit) to the BBBC022-v1 compound set, we were only able to extract annotations for 209 compounds as well.
>
> Furthermore, BBBC022-v1 **lacks dose-response and time-series variation**, which are key considerations when we select datasets. Our chosen datasets included dosage series and multiple time points for nuanced phenotypic modeling, and CP-CLIP is specifically designed to handle this complexity. For this reason, BBBC022-v1 was considered but not included when we curated the dataset collection. As described in [2], all compounds were applied at a single concentration (10 µM) and single time point (48h post-treatment) with no dose-response or temporal variation:
>
> > "......Briefly, U2OS cells were plated in quadruplicate in 384-well plates, incubated for 24 h to allow cells to adhere and resume growth, and then **treated with compounds for 48 h** (typical **concentration 10 µM**)...... "
>
> Based on all these aspects: lack of dose-response or temporal variation, and the practical constraints of the rebuttal timeline, we consider that extending our approach to the suggested dataset is not a priority at this stage. That said, we appreciate the suggestion and are interested in exploring this direction in future work.
>
> ---
> [1] Bray, Mark-Anthony, et al. "A dataset of images and morphological profiles of 30,000 small-molecule treatments using the Cell Painting assay." *Gigascience* 6.12 (2017): giw014.
> [2] Gustafsdottir, Sigrun M., et al. "Multiplex cytological profiling assay to measure diverse cellular states." *PloS One* 8(12) (2013): e80999.

---

> ### Author Response · Authors · 2025-12-03
> **Response to Evaluation for CP-CLIP, Point3 – Round 2**
>
> We appreciate the reviewer’s thoughtful question, and we offer two complementary perspectives to clarify the role of extensive metadata in our framework (CP-CLIP and CP-agent).
>
> 1. CP-CLIP: Metadata Ablation
> To directly address the concern that performance might be trivially driven by metadata correlations (e.g., compound name), we conducted a series of controlled ablation experiments, where we systematically masked specific textual fields in the input prompts—namely:(1) Compound Name + MOA, (2) Concentration, and (3) Time. As shown in **Section S, Table 17** (included in our response to Reviewer k4QM, under Weakness 2, Question 1 (1/2)), masking the compound name and MOA, as expected, leads to a big drop in retrieval performance. Simultaneously, masking concentration or time, while keeping the compound name and MOA visible, also results in substantial degradation (e.g., R@1 drops to 57.00 when masking concentration), suggesting that these fields also played substantial roles in model performance. It is evident from this experiment that the model is not merely relying on compound names (and MOA). Those non-compound metadata (e.g., concentration or time) also contributed to fine-grained alignment and semantic disambiguation.
>
> 2. CP-Agent: Metadata Enables Self-Reflective Reasoning in Report Generation
> Beyond retrieval, these metadata fields are critical for report generation in CP-agent, where the model is prompted to reason mechanistically about observed morphological features. In our LLM evaluation (e.g., Q8 and Q13 from the survey, **Section P**), the automatically generated reports regularly invoke metadata-aware reflections, such as:
>
> >*"......Caveats include the contradictory area increase, potentially due to low dose (24 µM) or time (24 hours) not fully inducing contraction, and the low cell count suggesting cytotoxicity; suggest dose-response experiments, time-course imaging, and orthogonal microtubule markers to verify..."*
>
> >*"......The weak or non-significant changes in cell shape features suggest the 24-hour timepoint primarily captures cytoskeletal reorganization rather than secondary morphological consequences, though longer exposure or higher concentrations might be needed to observe more pronounced cellular rounding...."*
>
> The generation of such self-reflection was intentionally designed in our report generation prompt template (**Section N**), where the model is able to reflect on possible inconsistencies between expected and observed morphological trends. This ability is important because Cell Painting images inherently contain biological and technical variability—due to factors like heterogeneous cell states, batch effects, subtle timing differences etc. In such settings, metadata like dose and time can affect whether a morphological phenotype is fully manifested, partially expressed, or confounded by off-target or cytotoxic effects. When such discrepancies occur, the model's confidence in those meaningful features (via *FeatRank*) may decrease, triggering self-reflective caveats in the report.
>
> Thus, metadata fields serve not only as descriptive variables, but also as anchors for mechanistic reasoning. They provide CP-agent with additional context to formulate plausible hypotheses and able to interpret unexpected phenotypic outcomes.
>
> More example reports and prompts can be found in our open-sourced codebase, which demonstrate how metadata interacts with model reasoning in practice.

---

> ### Author Response · Authors · 2025-12-03
> **Response to Evaluation for CP-Agent – Round 2**
>
> We appreciate the reviewer’s suggestion. In the revised manuscript, we now explicitly clarify our motivation and evaluation protocol in **Section 3.1**. To address concerns about the informativeness of zero-shot comparisons, we have conducted additional few-shot evaluations where each general-purpose MLLM is provided with a small visual memory bank before prediction. These results are included in **Appendix M.4, Table 12**. Across both zero-shot and few-shot settings, CP-Agent continues to outperform all tested MLLMs, supporting the robustness of our findings.
>
> To further strengthen the comparison, we also implemented a LoRA-fine-tuned baseline following the reviewer’s suggestion. Specifically, we fine-tuned Qwen-VL-2B-Instruct and Qwen-VL-7B-Instruct using our curated Cell Painting dataset. The training formulation follows a drug name prediction task, with each input structured as an instruction-style prompt, as shown one example below:
>
> ```json
> {
>   "id": "train_00000458",
>   "image": "bbbc021_compound_clean/Week9_090907_B05_s3_w48B6E79CE-F3C2-4AD9-98FA-DE19ABA66B79_3.png",
>   "conversations": [
>     {
>       "from": "human",
>       "value": "<image>\nCell line is MCF7; Cells cultured in RPMI 1640 medium with 10% fetal bovine serum, 1% GlutaMAX, and 900 ug/mL G418, maintained at 37C with 5% CO2; Image channel is Actin; The imaging objective is 20X; The concentration is 10.0; The observation time is 24 h; The perturbation compound is <mask>.\nWhat is the perturbation compound?"
>     },
>     {
>       "from": "gpt",
>       "value": "floxuridine"
>     }
>   ]
> }
> ```
>
> This prompt format, where compound names and corresponding MoA are masked, is consistent with our query formulation used in the retrieval-based classification tasks. The LoRA module was applied on both the vision and language components of Qwen-VL, while freezing the base vision tower, LLM, and merger, and the model was trained for 5 epochs. The CP-CLIP retrieval accuracy in the table below is not equivalent to results in Table 2, which evaluates classification over a predefined subset of compounds. In contrast, this retrieval task operates over the entire context bank, reflecting a more realistic open-world setting. As Qwen is a VLM that does not natively support restricting its output space during inference, we include this comparison to ensure a fairer evaluation under the same open-ended conditions. We found that LoRA-fine-tuning a large VLM like Qwen-VL doesn’t always perform better than training a smaller model from scratch, especially for such specific domain.
>
> Table: Benchmark with LoRA-finetuned VLM
>
> | Method                                | Flindokalner | Racecadotril | AZM-475271 | Misoprostol | Trazodone | Orantinib | Rufinamide | Lumiracoxib | BIRB-796 | Methoxsalen | Average  |
> |-------------------------------------------|------------------|------------------|----------------|-----------------|---------------|----------------|----------------|------------------|--------------|------------------|------------------|
> | CP-CLIP ViT-B/16 (shuffle Compound)       | 0.72             | 0.72             | 0.58           | 0.70            | 0.42          | 0.62           | 0.42           | 0.46             | 0.78         | 0.44             | 0.59 |
> | Qwen-VL-7B-Instruct (predict Compound)    | 0.18             | 0.20             | 0.22           | 0.10            | 0.46          | 0.22           | 0.26           | 0.26             | 0.18         | 0.42             |  0.25

---

### Official Review · Reviewer_k4QM · 2025-11-07

**Soundness:** 3
**Presentation:** 3
**Contribution:** 4
**Rating:** 6
**Confidence:** 3

**Summary:**

The paper introduces CP-Agent, an agentic multimodal large language model (MLLM) system designed to analyze Cell Painting assay data to study how chemical perturbations affect cell morphology. The core of CP-Agent is CP-CLIP, pretrained with cell paint images and experimental metadata pairs by contrastive learning. Then CP-Agent leverages the context inferred from CP-CLIP to help generate structured report including experimental design and hypothesis refinement.

**Strengths:**

1. The paper is original in explicitly treating experimental context as signal and injecting it into the text encoder for semantically image–text alignment with CLIP style training. The pretraining corpus including 1.9M image–context pairs, which is reasonable scale for Cell Painting experiments and well-suited to learning robust, context-aware representations.

2. The system is presented as a clean, step-wise workflow with structured JSON outputs and rich case studies that connect reasoning steps, making the reasoning traceable for practitioners.

3. CP-CLIP achieves robust MoA/treatment discrimination. The framework is portable to other imaging modalities and broader phenotypic screening use cases.

**Weaknesses:**

1. More ablation study is needed for the proposed method. The ablation study could include: 1) remove each context field (<CMPD>, <CONC>, <TIME>) separately, and then retrain the model, to show how the model performance change 2) remove control images from the image embedding, use only the perturbation tile, to verify the contribution of control embedding

2. Table 2 results with high performance on compound recovery, may be results from the fact that the CP-CLIP is training on the meta data text, and the model can recover compound from the text context. Additional experiment to test performance change: image-only, text-only, and shuffled-context controls (time and dosage).

**Questions:**

1. Can you report  image-only, text-only baselines and counterfactual context (swap dose/time ) to test the performance change?

2. Are agent outputs fully deterministic or do responses vary with randomness? How reproducible are the reasoning outputs across runs?

---

> ### Author Response · Authors · 2025-11-21
> **Response to Weakness 1**
>
> We appreciate the reviewer’s suggestion to perform additional ablation studies by removing each context field individually. We would like to clarify that the inclusion of all context fields is crucial for the reproducibility and interpretability of the experimental results. These fields are not arbitrary metadata, but essential conditions of the experimental setup that directly influence the observed cellular phenotypes.
>
> Specifically, **Cell culture conditions** (such as medium composition, cell line background) can significantly affect cell morphology and drug responses. For example, [1] demonstrated that different culture media led to notable changes in both the growth and phenotype of A549 and HepG2 cells when treated with selenium compounds. **Optical settings** (e.g., magnification, illumination) affect image quality. We retained only magnification due to its primary influence on image detail, while augmentation mitigates contrast-related variability. Within the **drug condition fields**, not only the compound identity but also the dosage and treatment time substantially influence cellular responses, both at the morphological and molecular levels. For example, [2] investigated the dose-dependent effects of three anticancer drugs — cisplatin (0–100 µM), vorinostat (0–5 µM), and erlotinib (0–20 µM) — on two breast cancer cell lines (MDA-MB-231 and MCF-7). Their results showed that different cell lines exhibited different levels of dose-dependent cytotoxicity among the three chemical compounds. Additional examples are provided in **Appendix B.2**.
>
> Therefore, we believe that performing ablation studies by retraining based on individual context fields may not yield interpretable or actionable insights, as these fields are fundamental to the experimental design and tightly coupled with cellular phenotypes. To evaluate the contribution of individual context fields, we performed an **inference-time ablation** in which selected fields in the prompt (e.g., compound name + MOA, concentration, or treatment time) were replaced with a [MASK] token, keeping the rest of the metadata intact. The results, summarized in **Appendix S (Table 17)** of the revised manuscript and referenced in W2/Q1, demonstrate that omitting any of these fields leads to notable (but difference level) performance degradation.
>
> Regarding (2). In our design, we stack the control and perturbation images as a dual-channel input to the image encoder for two key reasons: 1. **Anchoring within batch context**: All control–perturbation pairs come from the same batch, encouraging the model to learn the intra-pair visual shift caused by the perturbation. 2. **Explicit quantification**: This setup enables us to explicitly quantify the morphological feature changes between the control and perturbation images in the CP-agent pipeline, linking visual differences to textual descriptions of the experimental conditions. We conducted an ablation experiment by replacing the control image with a duplicate of the perturbation image, thereby removing the meaningful perturbation-induced visual differences.
>
> The results are shown in the table below:
>
> #### Retrieval Performance With and Without Control Image
>
> | Image Input                 | R@1  | R@5   | R@10  | R@20  | R@50  | MRR    | R@1  | R@5   | R@10  | R@20  | R@50  | MRR    |
> |----------------------------|------|-------|-------|-------|-------|--------|------|-------|-------|-------|-------|--------|
> |                                      | Context → Image         |                       |      |       |       |        | Image → Context          |       |       |       |       |        |
> | perturbation, perturbation | 58.4 | 81.48 | 87.56 | 92.4  | 96.36 | 0.668   | 46.89| 67.05 | 74.18 | 81.35 | 89.29 | 0.5605 |
> | control, perturbation      | 77.09| 94.69 | 97.87 | 99.21 | 99.74 | 0.8479  | 68.92| 87.77 | 92.14 | 95.56 | 98.55 | 0.7716 |
>
> ---
> #### References
>
> [1] Arodin Selenius L, Wallenberg Lundgren M, Jawad R, et al. *The cell culture medium affects growth, phenotype expression and the response to selenium cytotoxicity in A549 and HepG2 cells*. Antioxidants, 2019, 8(5): 130.
>
> [2] Domura R, Sasaki R, Ishikawa Y, et al. *Cellular morphology-mediated proliferation and drug sensitivity of breast cancer cells*. Journal of Functional Biomaterials, 2017, 8(2): 18.

---

> ### Author Response · Authors · 2025-11-21
> **Response to Weakness 2, Question 1 (1/2)**
>
> We thank the reviewer for raising this important point, which inspired us to perform a more detailed evaluation. Firstly, we would like to clarify the classification approach used in the main text. This classification does not involve training a separate classifier on top of the image and text embeddings. Inference is performed via masked or prompted text templates, and similarity ranking is calculated. This formulation avoids the need for training task-specific linear heads for various classification tasks, and allows us to flexibly handle different classification scenarios within the shared embedding space.
>
> Unlike general-purpose VLMs that process richly descriptive text, many fields in our structured text prompts are semantically repetitive because those data originate from the same experimental batch. So some prompt fields are shared across multiple compounds and are unable to provide sufficient information to distinguish compounds. This redundancy is exemplified in **Table 1**, which shows that under the same cell line, channel, time, and other contextual conditions, there can still be dozens to hundreds of distinct compounds.
>
> To address the concern that the high retrieval performance may be due to metadata correlations, we conducted controlled ablation experiments to isolate and evaluate the extent to which different textual components contribute to model performance. We adopt an inference-time masking strategy, where specific fields in the text prompt (e.g., compound name + MOA, concentration, or time) are masked, while all other metadata is kept unchanged. This allows us to isolate the contribution of individual textual components to retrieval performance without modifying the model. As shown in **Appendix S, Table 17** (summarized below), masking the compound information results in a **catastrophic performance drop** (e.g., text-to-image R@1 drops from 98.70 to 3.50; MRR drops by nearly 90%), indicating that the model relies heavily on compound-specific textual information. In contrast, masking concentration or time results in moderate performance degradation (e.g., R@1 drops to 3.50 when masking compound, and only to 93.00 when masking time). This pattern suggests that the model encodes and leverages different fields of the metadata (e.g., compound identity, concentration) independently and meaningfully.
>
> | Experiment | Modality        | Recall@1 Δ Absolute |
> |------------|------------------|-------------|
> | Mask Compound + MOA | T-I | -95.20      |
> |            | I-T    | -95.40      |
> | Mask Concentration  | T-I   | -41.70      |
> |            | I-T   | -43.50      |
> | Mask Time            | T-I | -5.70       |
> |            | I-T     | -4.05          |
>
> ---
>
> We also conducted **counterfactual context experiments** on dosage and time variables in two ways: 1. **Clean Prompt**: Masking the target field, and leaving unrelated metadata intact.  2. **Disturbed Prompt (Shuffled Context)**: Masking the target field, while also shuffling unrelated metadata (e.g., time, channel, magnification), keeping only the correct compound identity. This tests the model’s robustness to misleading contextual cues. Detailed results are now presented in **Appendix S, Table 18–21** in revised manuscript.  This shows that CP-CLIP is more robust to noisy or misleading metadata context, while the standard CLIP collapses.
>
> Summary of Concentration Classification Performance — CP-CLIP vs CLIP under clean vs. disturbed prompts*
>
> | Compound            | CLIP Δ (Disturbed-Clean)        | CP-CLIP Δ (Disturbed-Clean)         |
> |--------------------|----------|----------|
> | firocoxib          | -0.1474  | -0.0987  |
> | opicapone          | -0.1413  | -0.0675  |
> | cinoxacin         | -0.1087  | -0.0269  |
> | neratinib          | -0.2256  | -0.0319  |
> | hydroflumethiazide | -0.1644  | +0.0300  |
> | acetaminophen      | -0.1150  | +0.0082  |
> | primidone          | -0.1200  | -0.0032  |
>
> **CP-CLIP ↑ over CLIP (Clean):** +7.83% Accuracy
> **CP-CLIP ↑ over CLIP (Disturbed):** +54.05% Accuracy / +50.55% F1-Score
>
> Summary of Time Classification Performance — CP-CLIP vs CLIP under clean vs. disturbed prompts
>
> | Compound        | CLIP Δ (Disturbed-Clean)        | CP-CLIP Δ (Disturbed-Clean)         |
> |------------------|---------|---------|
> | ixabepilone      | -0.2400 | -0.0375 |
> | methoxsalen      | -0.3575 | -0.0425 |
> | sulfinpyrazone   | -0.3650 | -0.0825 |
> | triamterene      | -0.2900 | -0.1150 |
> | miconazole       | -0.3375 | -0.0675 |
> | ceritinib        | -0.3300 | -0.0650 |
> | acetohexamide    | -0.2350 | -0.0650 |
>
> **CP-CLIP ↑ over CLIP (Clean):** +0.08% Accuracy / +0.08% F1-Score
> **CP-CLIP ↑ over CLIP (Disturbed):** +34.91% Accuracy / +35.23% F1-Score
>
> Together, these results confirm that CP-CLIP does not rely on contextual shortcuts. Instead, it learns semantically grounded, robust representations that align morphology with textual semantics in a way that generalizes across both compounds and experimental conditions.

---

> > ### Author Response · Authors · 2025-11-21
> > **Response to Question 2**
> >
> > Thank you for the thoughtful question. The reproducibility of CP-Agent’s reasoning outputs is primarily influenced by the **temperature parameter** in the LLMs since the perturbation condition is inferred from CP-CLIP, which makes the conclusion of the report deterministic. In our pipeline, following the initial pretrained CLIP model, there are two LLM-powered modules, **FeatRank** and **ReportGen**. *FeatRank* ranks morphology features extracted by CellProfiler, and *ReportGen* generates natural language reports based on the ranked features and contextual information. To ensure that the feature ranking step is as deterministic as possible, we set the temperature of *FeatRank* to 0. To systematically evaluate the reproducibility, we designed five experiments with varying temperature settings, *ReportGen*'s temperature was set to values ranging from 0.0 to 1.0, while keeping *FeatRank*'s temperature fixed at 0.0. In each setting, we repeated the pipeline 30 times on the same input samples and analyzed the consistency of the selected features and generated reports.
> >
> > In the five independent runs (fixed temperature for *FeatRank*). The number of selected features per run was consistent around 18, with 14 high-frequency features commonly identified out of 396 feature candidates by *FeatRank*. The top five stable features remained the same across the five experiments, with results now shown in **Appendix R, Table 15** (shown below), confirming the output’s robustness in identifying key features despite some minor variability.
> >
> > #### Table: FeatRank Agent's Repeatability
> >
> > | FeatRank's Temp 1 | Features Number Avg |Features Number std | Top 5 Stable Features |
> > |-----------|--------------------------|--------------------------|----------------------------|
> > | 0.0       | 18.37                    | 2.37                     | AreaShape_Area, AreaShape_Eccentricity, Texture_Contrast_5_02_256, Granularity_2, Granularity_3 |
> > | 0.0       | 18.20                    | 2.26                     | AreaShape_Area, AreaShape_Eccentricity, Texture_Contrast_5_02_256, Granularity_2, Granularity_3 |
> > | 0.0       | 18.27                    | 2.31                     | AreaShape_Area, AreaShape_Eccentricity, Texture_Contrast_5_02_256, Granularity_2, Granularity_3 |
> > | 0.0       | 18.87                    | 1.96                     | AreaShape_Area, AreaShape_Eccentricity, Texture_Contrast_5_02_256, Granularity_2, Granularity_3 |
> > | 0.0       | 17.17                    | 2.28                     | AreaShape_Area, AreaShape_Eccentricity, Texture_Contrast_5_02_256, Granularity_2, Granularity_3 |
> >
> > To evaluate the consistency of the summary reports, we conducted a **corpus-level evaluation** using GPT-5 as the judgment model. Three dimensions are set: Thematic Consistency, Terminological Consistency, and Structural Consistency. The judgmental model is asked to give score from 0-10 to each batch of reports, and a detailed scoring scheme also given in prompts (**Appendix R, prompt**). The evaluation revealed a stable corpus-level consistency across different temperature settings with minimal variation, suggesting that the generated reports maintain the internal coherence, terminology, and structure.
> >
> > #### Table: Corpus score comparison across different temperature setting.
> >
> > | FeatRank's Temp/ReportGen's Temp | Thematic Consistency | Terminological Consistency | Structural Consistency| Averaged Corpus Score |
> > |----------------|---------------------------|-------------------------------|-----------------------------|----------------------------|
> > | 0.0/0.0        | 8.00                      | 7.00                          | 9.00                        | 8.00                       |
> > | 0.0/0.1        | 8.00                      | 7.00                          | 8.00                        | 7.67                       |
> > | 0.0/0.2        | 8.00                      | 7.00                          | 8.00                        | 7.67                       |
> > | 0.0/0.5        | 8.00                      | 7.00                          | 8.00                        | 7.67                       |
> > | 0.0/1.0        | 8.00                      | 7.00                          | 8.00                        | 7.67                       |
> >
> > We have added a new section in the revised manuscript (**Appendix R**: “Robustness Evaluation: FeatRank Agent and ReportGen Agent”) detailing these experiments, along with comprehensive results and evaluation prompts to ensure clarity and reproducibility.

---

> > ### Author Response · Authors · 2025-11-23
> > **Response to Weakness 2, Question 1 (2/2)**
> >
> > To further assess the contribution of microscopy images while avoiding spurious correlations between textual fields, we also conducted an **image-free experiments** using only textual metadata and compound identity. Specifically, we constructed a text-only input that describes the experimental setup context, while masking the compound-related information (compound identity and MOA). The resulting input, compound-masked metadata context is fed into a text encoder, whose output is then passed to a classifier trained to predict the compound label (from 450 unique compounds).
> >
> > To ensure a fair evaluation, the text dataset was deduplicated and then split into training and validation sets (10:1 ratio), such that no instance in the validation set overlaps with those in the training set. A simple classifier (LayerNorm + MLP) on top of the text encoder is used to predict compound labels.
> >
> > The results of this experiment are shown in the table below. Across both XGBoost and MLP classifier settings, and under both pure text (original CLIP's text encoder) and hybrid token injection modes (CP-CLIP's text encoder), the model performance remained near-random. For reference, random guessing would yield an expected accuracy of approximately **0.22%** (1/450), and all observed accuracies fall at or below this level.
> >
> > #### Table: Metadata–Compound Classification Accuracy
> >
> > | Model (Input Mode)          | Accuracy | Random Guess |
> > |---------------------------------|--------------|------------------|
> > | XGBoost (Pure Text)             | 0.050%       | 0.222%           |
> > | XGBoost (Hybrid Text)           | 0.110%       | 0.222%           |
> > | MLP Classifier (Pure Text)      | 0.368%       | 0.222%           |
> > | MLP Classifier (Hybrid Text)    | 0.343%       | 0.222%           |
> >
> > These findings indicate that the model also relies on meaningful visual signals in the microscopy images related to compound identity, rather than relying on metadata or experimental context leakage.

---

### Author Response · Authors · 2025-12-03
**General Response -- Summary**

For ease of reference, we refer to reviewers k4QM, uvt6, and 3gDr as R1, R2, and R3 in what follows.

---
We would like to express our sincere appreciation to all reviewers for their insightful and constructive comments, which have helped us further strengthen the completeness, clarity, and rigor of our work. In particular, we appreciate the reviewers' positive remarks on:
- **Reframing experimental context as meaningful signal for semantic alignment:**  Experimental context (e.g., compound, concentration, time) is treated as meaningful signal and fused via token projections, supporting semantically aligned contrastive learning. (**R1, R3**)
- **Conceptual novelty of introducing agentic framework:**  The introduction of agentic framework tailored for Cell Painting is conceptually original. It represents a step toward structured, interpretable automation in cellular image analysis and drug discovery. (**R2**)
- **Structured system design with traceable reasoning:**  The system is presented as a clean, step-wise workflow with structured JSON outputs and case studies connecting each reasoning step. (**R1, R3**)
- **Robust performance with scalable and transferable design:** Strong performance in MoA/treatment discrimination and is designed with principled dataset integration, making it scalable to new datasets and portable to other imaging modalities and phenotypic screening workflows. (**R1, R3**)
---
We have carefully addressed the major concerns raised by the reviewers and incorporated new experiments and clarifications in the manuscript. Below, we summarize the key concerns and our corresponding updates:
- **Ablation Studies of CP-CLIP:**  More rigorous ablation studies needed for CP-CLIP, specifically on individual metadata fields, control image embeddings, image-free baseline. (**R1, R3**)
  -- Addressed in Response to Weakness 1 & 2 (R1), Response to Question 3 (R3)
  -- **Appendix S, Table 17–21 and Additional Ablation Studies** conducted validity checks to ensure the model learns meaningful morphological signals. These include:
  (i) Inference-time ablations of context fields (compound/dosage/time) and counterfactual (shuffled) prompt experiments.
  (ii) Image-free baselines (masked compound) to rule out metadata leakage.
  (iii) Control-image ablations to verify the necessity of the dual-channel input design.

- **Ablation Studies of CP-Agent:**  Need to assess each module's contribution and the robustness of report generating. (**R1, R2**)
  -- Addressed in **Appendix R**, where we evaluated the robustness and reproducibility of FeatRank and ReportGen agents by varying the temperature to verify output determinism.

- **CP-CLIP evaluation metrics** (**R2**)
  -- Addressed in **Appendix T, Table 23**, where we evaluated CP-CLIP by additional retrieval metrics (Recall@K and MRR) on the held-out validation set.
- **Classification method unclear** (**R2**)
  -- Addressed in **Section 3.2** and **Figure 1**, where we clarified the retrieval-based classification and provided a detailed explanation of the "raw text" tokenization process.
  -- **Section 2.1**, where we described the composition of training, validation, and test sets across the JUMP, RxRx, and BBBC021 datasets.
- **VLM Baseline Comparisons:** Need more strong baselines for drug name prediction. (**R2, R3**)
  -- **Section 3, Table 2**: Added a direct quantitative comparison with CLOOME to benchmark CP-CLIP against compound-only contrastive learning frameworks.
  -- **Appendix M.4, Table 12 and fine-tuned VLM**: Expanded model comparisons to ensure fair assessment, including:
  (i) Few-shot classification benchmark for general-purpose MLLMs.
  (ii) LoRA fine-tuned VLM baseline (Qwen-VL).
- **Presentation and clarity issues** (**R2**)   -- Corrected
- **Advantage of extensive metadata for downstream tasks** (**R2**)
  -- Addressed in **Appendix S, Table 17**, where we provided inference-time ablations of different context fields.
  -- Generated report examples
- **Datascope:**  Use Bray et al Nature Protocol 2016, Bray et al GigaScience 2017 for assessment (**R2**)
  -- Clarified in response (R2) regarding the two datasets
- **Related Work:**  Omitted discussion of several relevant studies (e.g., CLOOME, CellCLIP, MolPhenix). (**R2**)
  -- In **Section 1**, we expanded the literature review to include the related works and clarified how CP-CLIP distinguishes itself.
- **Expert Evaluation Design:**  The evaluation lacks inter-rater reliability analysis. (**R3**)
  -- Addressed in **Appendix Q.2, Table 12–13**, where we evaluated the inter-rater consistency of the expert evaluation.
- **Definition and Justification needed for “Agentic” framing** (**R3**)
  -- Clarification added in **Section 2.5**: we refined the definition of “agentic” to better characterize CP-Agent as a procedurally autonomous system, rather than an RL-based planner.

---

### Meta-Review · Area_Chair_7AGJ · 2026-01-18

**Summary:**

The initial reviews raised concerns about missing ablations, unclear evaluation protocols, limited baseline comparisons, etc.. After the rebuttal, the authors added substantial new evaluations, addressing most technical concerns. Some concerns remain, but overall, the contribution is largely strengthened.

Since the rebuttal largely improves the experimental completeness and addresses the key technical concerns, and the remaining concerns are primarily about a broader generalization/evaluation scale rather than correctness, I recommend acceptance.

**Reviewer Concerns:**

Most technical concerns about the paper are addressed by the abundant experiments provided in the rebuttals. The evaluations for CP-CLIP and CP-Agent remain a concern for Reviewer uvt6

**Reviewer Scores:**

The two positive scores are likely to remain the same, while the negative score might raise to around the borderline.

---

### Decision · Program_Chairs · 2026-01-26

Accept (Poster)